# Counterfactual Identifiability via Dynamic Optimal Transport

**Fabio De Sousa Ribeiro**[†]      **Ainkaran Santhirasekaram**      **Ben Glocker**

Imperial College London, UK

## Abstract

We address the open question of counterfactual identification for high-dimensional multivariate outcomes from observational data. Pearl (2000) argues that counterfactuals must be identifiable (i.e., recoverable from the observed data distribution) to justify causal claims. A recent line of work on counterfactual inference shows promising results but lacks identification, undermining the causal validity of its estimates. To address this, we establish a foundation for multivariate counterfactual identification using continuous-time flows, including non-Markovian settings under standard criteria. We characterise the conditions under which flow matching yields a unique, monotone, and rank-preserving counterfactual transport map with tools from dynamic optimal transport, ensuring consistent inference. Building on this, we validate the theory in controlled scenarios with counterfactual ground-truth and demonstrate improvements in axiomatic counterfactual soundness on real images.

## 1 Introduction

It has been argued that shortcomings in today's deep models reveal an overreliance on statistical associations and a lack of *causal* understanding (Schölkopf et al., 2021; Bareinboim et al., 2022). This view has sparked interest in causality within machine learning (Peters et al., 2017; Castro et al., 2020; Schölkopf, 2022). Causal reasoning is the process of drawing conclusions from a causal model, which represents our assumptions about the data-generating process. Thus, causality and generative modelling are inextricably linked. Pearl (2009) formalised causal inference using the Structural Causal Model (SCM) framework, in which variables are generated by functional assignments and dependencies are represented by a graph. This framework (with *do*-calculus (Pearl, 1995)) expresses causal queries and, under stated assumptions, *identifies* causal effects that can be estimated from data.

Why is *identification* essential? Without identification it is impossible to make a precise causal claim: there can exist observationally equivalent models that yield different answers, and providing such guarantees is what causal methods are designed for (Bareinboim et al., 2022; Hyvärinen et al., 2024). Indeed, *do*-calculus (Pearl, 1995) is an identification tool that tells us whether a causal effect can be determined from observed data and a set of assumptions encoded in a causal graph. Further, Pearl (2000) insists that any counterfactual query be bound by an identifiability requirement, that is, that there exists a unique mapping from the observed data distribution to the counterfactual of interest.

This work focuses on the identification of high-dimensional counterfactuals from observational data. Counterfactuals are hypothetical scenarios given observed evidence, for example, one may query *"What would $Y$ have been, had $X$ been $x$"*. Counterfactuals have broad scientific utility, supporting the evaluation of interventions (Kusner et al., 2017; Tsirtsis and Rodriguez, 2023), the characterisation of causal relationships (Karimi et al., 2020; Budhathoki et al., 2022), and the generation of targeted synthetic data for downstream tasks (Pitis et al., 2022; Roschewitz et al., 2024; Mehta et al., 2025).

---

[†]Email: f.de-sousa-ribeiro@imperial.ac.uk

39th Conference on Neural Information Processing Systems (NeurIPS 2025).

Deep generative models are increasingly being used to parameterise SCMs (Pawlowski et al., 2020; Sanchez and Tsaftaris, 2021; Ribeiro et al., 2023; Komanduri et al., 2024; Kumar et al., 2025; Rasal et al., 2025; Xia et al., 2025). While the inferred counterfactuals are promising, the lack of identification precludes warranted causal interpretations of the estimates beyond statistical predictions. We aim to course-correct this practice. Counterfactual identification is arguably the most ambitious goal in causal analysis, as counterfactuals sit at the top of the causal hierarchy and subsume all other causal queries (Bareinboim et al., 2022). Thus, existing identification results are limited, with classical symbolic methods (Tian and Pearl, 2000; Pearl, 2001; Shpitser and Pearl, 2007) not having been developed for high-dimensional variables (Xia et al., 2023; Nasr-Esfahany and Kiciman, 2023).

Nasr-Esfahany et al. (2023) established counterfactual identification of bijective mechanisms, proposing a flow-based estimation method similar to Pawlowski et al. (2020); Khemakhem et al. (2021). However, their identification results only extend to multi-dimensional variables under the Backdoor Criterion (BC) and sufficient *variability* (Hyvarinen et al., 2019), stating that: "it is not clear how to generalise the monotonicity condition to multi-dimensional variables" in Markovian causal structures. This is a well-known issue with close connections to ICA (Hyvärinen and Pajunen, 1999) and disentanglement (Locatello et al., 2019). In this work, we characterise a multivariate generalisation of monotonicity to provably solve this problem, without imposing an arbitrary coordinate order.

We establish a foundation for counterfactual identification of high-dimensional, multivariate outcomes (e.g. images) from observational data, built on continuous-time flows. We characterise the conditions under which flow matching yields a unique, rank-preserving counterfactual transport map using tools from dynamic optimal transport. Our results address causal validity issues in prior work and extend to non-Markovian settings under standard criteria. We validate our theory in controlled scenarios with counterfactual ground-truth, and improve axiomatic counterfactual soundness on real images.

## 2   Related Work

Pearl's Causal Hierarchy (PCH) (Pearl and Mackenzie, 2018) defines three levels of causal abstraction: *associational* ($\mathcal{L}_1$); *interventional* ($\mathcal{L}_2$), and *counterfactual* ($\mathcal{L}_3$). The PCH Theorem (Bareinboim et al., 2022) states that cross-layer inference of $\mathcal{L}_i$ quantities using $\mathcal{L}_{<i}$ data is generally not possible without further assumptions. Pearl (1995)'s *do*-calculus is a sound and complete procedure for determining the identifiability of an $\mathcal{L}_2$ causal quantity from $\mathcal{L}_1$ data, given a causal graph. Counterfactual identification ($\mathcal{L}_3$) represents the most ambitious goal in causal analysis as it sits at the top of the PCH and therefore subsumes all other causal queries (Pearl, 2009). Subsequently, known counterfactual identification results are comparatively limited in number and scope to date, and existing constraints needed for $\mathcal{L}_3$ identification can be broadly categorised as either graphical or functional in nature.

Classical identification techniques involve symbolic methods and assumptions about the causal graph to identify counterfactual queries from both $\mathcal{L}_1$ and $\mathcal{L}_2$ data (Tian and Pearl, 2000; Pearl, 2001; Shpitser and Pearl, 2007). Recently, graph-based criteria have been extended to nested counterfactuals and fairness-aware queries (Zhang and Bareinboim, 2018; Correa et al., 2021), whereas functional identification has moved towards neural and nonparametric schemes that approximate or bound counterfactuals (Hartford et al., 2017; Xia et al., 2023; Melnychuk et al., 2023; Geffner et al., 2024; Pan and Bareinboim, 2024; Wu et al., 2025), even in non-identifiable settings (Gresele et al., 2022). Identification results predicated on monotonicity (Lu et al., 2020; Vlontzos et al., 2023; Javaloy et al., 2023) and bijectivity (Nasr-Esfahany et al., 2023) constraints are of primary interest to our work, since they generalise a broad range of model classes with known identifiability results, such as linear and nonlinear additive noise models (Shimizu et al., 2006; Hoyer et al., 2008; Peters et al., 2014), post-nonlinear models (Zhang and Hyvärinen, 2009), and location-scale models (Immer et al., 2023).

Establishing counterfactual identification of high-dimensional Markovian SCMs is a timely contribution, as many recent works rely on such assumptions (Pawlowski et al., 2020; Sanchez and Tsaftaris, 2021; Sánchez-Martin et al., 2022; Ribeiro et al., 2023; Chen et al., 2024; Rasal et al., 2025). However, these methods lack an identification strategy and cannot support causal claims, posing operational risk. Of particular relevance to our work is optimal transport (Brenier, 1991; Caffarelli, 1992; Benamou and Brenier, 2000; Villani et al., 2008; Santambrogio, 2015; Peyré et al., 2019), and modern continuous-time flow models (Chen et al., 2018; Lipman et al., 2023; Liu et al., 2023; Albergo and Vanden-Eijnden, 2023; Pooladian et al., 2023; Tong et al., 2024), which in combination form the basis of our theoretical results and practical counterfactual inference model prescription.

# 3 Preliminaries

## 3.1 Structural Causal Models

**Definition 3.1** (Structural Causal Model (SCM) (Pearl, 2009)). An SCM $\mathfrak{C} = (\mathbf{U}, \mathbf{X}, \mathcal{F})$ is a mathematical tool designed to express and infer causal quantities. It consists of: (i) a set of exogenous variables $\mathbf{U} = \{U_1, \ldots, U_n\}$ determined by factors outside of $\mathfrak{C}$, and distributed according to $P_{\mathbf{U}}$; (ii) a set of endogenous variables $\mathbf{X} = \{X_1, \ldots, X_n\}$ determined by other variables in $\mathfrak{C}$; and (iii) a set of functions $\mathcal{F} = \{f_1, \ldots, f_n\}$ specifying the causal generative process mapping $\mathbf{U}$ to $\mathbf{X}$:

$$X_i := f_i(\mathbf{PA}_i, U_i), \qquad \mathbf{PA}_i \subseteq \mathbf{X} \setminus \{X_i\}, \qquad \text{for } i = 1, \ldots, n, \tag{1}$$

where $\mathbf{PA}_i$ is the subset of endogenous variables that directly cause $X_i$, called its *parents*. If the causal generative process is acyclic, an SCM can be represented by a Directed Acyclic Graph (DAG).

**Counterfactual Inference.** A counterfactual is a claim about what would have happened if some fact were different, all else being equal. SCMs can express and answer counterfactual queries of the form: *"Given that we observed $\mathbf{X} = \mathbf{x}$, what would $\mathbf{X}$ have been had $X_i$ been set to $x^*$"*. An intervention sets chosen variables to specified values, e.g. $do(X_i := x^*)$ or $do(\mathbf{X}_S := \mathbf{x}_S^*)$, where $S \subseteq \{1, \ldots, n\}$ indexes the intervened variables in the SCM, and the components of $\mathbf{x}_S^*$ may differ[1]. Counterfactual inference proceeds in three steps: (i) **Abduction**: infer the posterior distribution over the exogenous variables $P_{\mathbf{U}|\mathbf{X}=\mathbf{x}}$, given observed evidence $\mathbf{X} = \mathbf{x}$; (ii) **Action**: intervene, e.g., apply $do(X_i := x^*)$, to obtain a modified SCM $\mathfrak{C}_{x^*}$; (iii) **Prediction**: use the SCM $\mathfrak{C}_{x^*}$ and the posterior over the exogenous variables $P_{\mathbf{U}|\mathbf{X}=\mathbf{x}}$ to compute the counterfactual distribution $P_{\mathbf{X}}^{\mathfrak{C}_{x^*}|\mathbf{X}=\mathbf{x}}$.

Without loss of generality, in this work, we focus on the counterfactual identification ($\mathcal{L}_3$) of causal mechanisms $f_i$ of multi-dimensional ($d > 1$) variables $X_i \in \mathbf{X}$, given only observational data ($\mathcal{L}_1$).

## 3.2 Optimal Transport: Static & Dynamic

Optimal Transport (OT) (Villani et al., 2008; Santambrogio, 2015; Peyré et al., 2019) is a suite of techniques for learning an optimal map between measures that minimises a transport cost.

**Monge and Kantorovich Formulation.** Let $\mathcal{P}(\Omega)$ be the set of probability measures on $\Omega \subset \mathbb{R}^d$. The Monge formulation of OT seeks a map $T : \Omega \to \Omega$ between two distributions $\mu, \nu \in \mathcal{P}(\Omega)$ that pushes $\mu$ forward to $\nu$ (i.e., $T_\sharp \mu = \nu$) while minimising a transport cost function $c : \Omega \times \Omega \to \mathbb{R}$:

$$W_c(\mu, \nu) := \inf_{T_\sharp \mu = \nu} \int_\Omega c(x, T(x)) \, \mathrm{d}\mu(x), \quad \text{e.g.} \quad c(x, T(x)) = \|x - T(x)\|^2. \tag{2}$$

However, a transport map $T$ may fail to exist, for instance when $\mu$ is discrete and $\nu$ is continuous. The Kantorovich formulation of OT relaxes the Monge problem and seeks an optimal coupling $\pi \in \Pi(\mu, \nu)$, where $\Pi(\mu, \nu) \subset \mathcal{P}(\Omega \times \Omega)$ denotes the set of couplings with marginals $\mu$ and $\nu$[2]:

$$\pi^\star = \arg\min_{\pi \in \Pi(\mu, \nu)} \int_{\Omega \times \Omega} c(x, y) \, \mathrm{d}\pi(x, y). \tag{3}$$

An optimal $\pi^\star$ always exists, and when a Monge map $T$ also exists, both formulations coincide. Brenier (1991)'s theorem states that, under fairly general conditions, there is a unique and monotone optimal transport map $T = \nabla \phi$, where $\phi : \mathbb{R}^d \to \mathbb{R}$ is a convex function. In this work, we show that this result has far-reaching implications for the counterfactual identification of causal mechanisms.

**Benamou-Brenier Formulation.** Benamou and Brenier (2000) showed that the above formulation, known as *static* OT with quadratic cost, can be equivalently expressed using a *dynamic* formulation. In simplified terms, one seeks a time-dependent velocity field $v : \mathbb{R}^d \times [0, 1] \to \mathbb{R}^d$, for $t \in [0, 1]$, that transports $p_0$ to $p_1$ along a flow defined by an Ordinary Differential Equation (ODE):

$$v^\star = \arg\min_{v \in \mathcal{V}} \left\{ \int_0^1 \mathbb{E}\left[\|v(X_t, t)\|^2\right] \mathrm{d}t \; : \; \mathrm{d}X_t = v(X_t, t)\,\mathrm{d}t, \; X_0 \sim p_0, \; (X_1)_\sharp p_0 = p_1 \right\}, \tag{4}$$

where $\mathcal{V}$ is the set of admissible velocity fields, i.e., fields that are measurable, ensure a well-posed flow with a unique ODE solution, and yield finite kinetic energy while transporting $p_0$ to $p_1$.

---

[1] There are other types of interventions, such as replacing $f_i$ by some new mechanism $\widetilde{f}_i$.

[2] That is, for any measurable sets $A, B \subset \Omega$, we have that $\pi(A \times \Omega) = \mu(A)$ and $\pi(\Omega \times B) = \nu(B)$.

### 3.3 Neural ODEs and Flow Matching

Continuous Normalizing Flows (CNFs) (Chen et al., 2018) seek to learn a mapping from a simple base distribution $X_0 \sim p_0$ to the data distribution $X_1 \sim p_{\text{data}}$ using an ODE, whose time-dependent vector field $v : \mathbb{R}^d \times [0, 1] \to \mathbb{R}^d$ is parameterised by a neural network with parameters $\theta$:

$$\forall t \in [0, 1], \qquad \mathrm{d}X_t = v_t(X_t; \theta)\,\mathrm{d}t, \qquad p_t := (X_t)_\sharp p_0. \qquad (5)$$

Flow Matching (FM) (Lipman et al., 2023; Liu et al., 2023; Albergo and Vanden-Eijnden, 2023) trains CNFs simulation-free by regressing a parameterised vector field onto a known target field:

$$\min_\theta \int_0^1 \mathbb{E}_{X_1 \sim p_{\text{data}}, X_t \sim p_t} \left[ \| v_t(X_t; \theta) - v_t^\star(X_t \mid X_1) \|^2 \right] \mathrm{d}t, \quad X_t = (1 - t)X_0 + tX_1, \qquad (6)$$

where the target vector field is defined to be $\mathrm{d}X_t = v_t^\star(X_t \mid X_1)\,\mathrm{d}t = (X_1 - X_0)\,\mathrm{d}t$. The simplicity of $v_t^\star$ is a result of choosing a simple *linear* interpolation path (McCann, 1997) between $p_0$ and $p_1$. OT has recently been used to augment neural ODEs by straightening the sample paths of flows, and speeding up simulations at inference time (Liu et al., 2023; Pooladian et al., 2023; Tong et al., 2024).

## 4 Counterfactual Identifiability: Theoretical Analysis

We begin by defining the scope of our theoretical analysis in the context of existing counterfactual identifiability results. Detailed proofs for all our theoretical results are provided in Appendix B.

### 4.1 Problem Statement

Without loss of generality, we focus on the counterfactual identification of the $i^{\text{th}}$ causal mechanism $f$ for a multi-dimensional variable $X$ within an SCM $\mathfrak{C}$, where $\dim(X) = \dim(U) = d > 1$, given only observational data. Further, we are interested in the *Markovian* case where there is no unobserved confounding, that is, $U$ is independent of $X$'s parents $U \perp\!\!\!\perp \mathbf{PA}_X$ as detailed in Definition 4.1 below.

**Definition 4.1** (Markovian SCM). An SCM is said to be Markovian if $U_i \perp\!\!\!\perp U_j$ whenever $i \neq j$. In other words, the exogenous variables in the model are statistically independent of each other, and their joint distribution factorises $P_{\mathbf{U}}(\mathbf{u}) = \prod_{i=1}^n P_{U_i}(u_i)$, where $\mathbf{u}$ is a realisation of $\mathbf{U}$. Thus, Markovian SCMs induce a unique, factored joint observational distribution: $P_{\mathbf{X}}^{\mathfrak{C}}(\mathbf{x}) = \prod_{i=1}^n P_{X_i \mid \mathbf{PA}_i}(x_i \mid \mathbf{pa}_i)$.

Counterfactual identifiability ($\mathcal{L}_3$, cf. Definition 4.2) results using only observational ($\mathcal{L}_1$) data exist for the Markovian case when $d = 1$, by using a strict monotonicity constraint on $f$. However, results for the $d > 1$ case are still missing, as generalising the monotonicity condition with multi-dimensional $(X, U)$ variables is non-trivial (Nasr-Esfahany et al., 2023), and remains unexplored in counterfactual identifiability literature. In the presence of unobserved confounding (i.e. non-Markovian SCMs), Nasr-Esfahany et al. (2023) established counterfactual identifiability from observational ($\mathcal{L}_1$) data for: (i) $d = 1$ given a set of Instrumental Variables (IVs)[3], $\mathbf{I}$ (Imbens and Angrist, 1994); and (ii) $d \geq 1$ given a set of variables $\mathbf{Z}$ that satisfy the Backdoor Criterion (BC)[4] (Pearl, 2009), w.r.t. the pair $(X, \mathbf{PA}_X)$.

**Definition 4.2** (Counterfactual Identifiability (Pearl, 2009)). A counterfactual query $Q$ is identifiable if for any pair of SCMs $\mathfrak{C}^{(1)}$ and $\mathfrak{C}^{(2)}$ we have that $Q(\mathfrak{C}^{(1)}) = Q(\mathfrak{C}^{(2)})$, whenever $P_{\mathbf{X}}^{\mathfrak{C}^{(1)}} = P_{\mathbf{X}}^{\mathfrak{C}^{(2)}}$.

Nasr-Esfahany et al. (2023)'s counterfactual identifiability results rely on the bijectivity and monotonicity of the causal mechanism $f$, except for the BC case where bijectivity alone is shown to be sufficient for $d \geq 1$ given that a *variability* assumption holds. This assumption requires that $\mathbf{Z}$ influence $U$ strongly enough for $P_{U \mid \mathbf{Z}}$ and its gradient to vary across different values of $\mathbf{Z}$. In Markovian settings, the independence of $U$ and $\mathbf{PA}_X$ alone is not sufficient for counterfactual identifiability of $f$. In addition, bijectivity constraints on $f$ are also insufficient when $d > 1$, due to well-known rotational symmetries of the prior on $U$ (Hyvärinen and Pajunen, 1999; Hyvärinen et al., 2024).

In the following, we characterise the set of constraints and assumptions that permit counterfactual identifiability for multi-dimensional variables in Markovian SCMs from observational data. For this, we connect ideas from (dynamic) optimal transport (Benamou and Brenier, 2000; Villani et al., 2008; Santambrogio, 2015) and graphical causality (Pearl, 2009) to build monotone mechanisms for $d > 1$.

---

[3]IVs must be $\mathbf{I} \perp\!\!\!\perp \mathbf{U}$ and only influence $X$ through $\mathbf{PA}_X$.

[4]If $\mathbf{Z}$ blocks all backdoor paths from $\mathbf{PA}_X$ to $X$; non-descendancy of $\mathbf{Z}$ is only required for adjustment.

## 4.2 Dynamic Optimal Transport for Counterfactual Identification

A counterfactual query $X_{\mathbf{pa}^*} | \{X = x, \mathbf{PA} = \mathbf{pa}\}$ can be answered by the deterministic map[5]:

$$f : \mathbb{R}^k \times \mathbb{R}^d \to \mathbb{R}^d, \quad \forall x, \mathbf{pa}, \mathbf{pa}^* : x^* = f(\mathbf{pa}^*, u), \quad \text{where} \quad u = f^{-1}(\mathbf{pa}, x), \qquad (7)$$

with $\mathbf{pa}^*$ denoting the counterfactual parents of $x$, and $f$ is invertible w.r.t. the exogenous noise $u$. Therefore, a counterfactual transport map that implicitly abducts the exogenous noise exists:

$$T^* : \mathbb{R}^{2k} \times \mathbb{R}^d \to \mathbb{R}^d, \quad \forall x, \mathbf{pa}, \mathbf{pa}^* : T^*(\mathbf{pa}^*, \mathbf{pa}, x) = f(\mathbf{pa}^*, f^{-1}(\mathbf{pa}, x)), \qquad (8)$$

directly pushing the observational distribution forward to the counterfactual $T^*_\sharp P^{\mathfrak{C}}_{X|\mathbf{PA}=\mathbf{pa}}$.

**Definition 4.3** (Monotone Operator)**.** A mapping $f : \mathbb{R}^k \times \mathbb{R}^d \to \mathbb{R}^d$ is monotone in $u$ if:

$$\langle f(\mathbf{pa}, u_1) - f(\mathbf{pa}, u_2), u_1 - u_2 \rangle \geq 0, \qquad \forall u_1, u_2 \in \mathbb{R}^d, \mathbf{pa} \in \mathbb{R}^k. \qquad (9)$$

**Proposition 4.4** (Monotone Counterfactual Transport Map)**.** *If a mechanism $f(\mathbf{pa}, u)$ is monotone in $u$ (Def. 4.3), then the respective counterfactual transport map $T^*(\mathbf{pa}^*, \mathbf{pa}, x)$ is monotone in $x$.*

*Remark* 4.5. In Appendix A we provide a gentle motivating example in $(d = 1)$-dimension of what monotonicity of $T^*$ entails and why it is important for consistent counterfactual inferences. The subsequent proof provides a generalisation to vectors (i.e., $d > 1$) in the sense of monotone operators.

Monotonicity of the counterfactual transport map $T^*$ w.r.t. $x$ guarantees that, for a given intervention on $\mathbf{pa}$, the rank order of factual outcomes is preserved in the counterfactuals. This is important (e.g. for *fairness*) as it prevents rank inversions across individuals under a given intervention. However, learning the counterfactual transport map directly from data is challenging, since we (almost) never have access to paired or unpaired samples from both the observational and counterfactual distributions.

As we will show, the dynamic optimal transport specialisation of $T^*$ generalises quantile and rank functions for $d > 1$, ensuring multivariate rank preservation and consistent counterfactual inferences.

**Lemma 4.6** (Unique and Monotone Dynamic OT Mechanism)**.** *Let $\dim(X) = \dim(U) = d > 1$, and consider a Markovian setting $U \perp\!\!\!\perp \mathbf{PA}$. Assume $P_U$ and $P^{\mathfrak{C}}_{X|\mathbf{PA}}$ are absolutely continuous w.r.t. the Lebesgue measure with strictly positive and bounded densities on bounded, open, convex domains. Let $T : \mathbb{R}^d \times \mathbb{R}^k \to \mathbb{R}^d$ be the time-1 map of a dynamic optimal transport flow:*

$$\{T_t : t \in [0, 1]\}, \qquad \mathrm{d}T_t(u; \mathbf{pa}) = v_t(T_t(u; \mathbf{pa})) \, \mathrm{d}t, \qquad T_\sharp P_U = P^{\mathfrak{C}}_{X|\mathbf{PA}}. \qquad (10)$$

*Then, there exists a convex function $\phi : \mathbb{R}^d \to \mathbb{R}$ such that: $T(u; \mathbf{pa}) = \nabla_u \phi(u; \mathbf{pa})$, for $P_U(u)$-a.e. $u$, where $T$ is monotone, bijective a.e., and uniquely determined by the pair $(P_U, P^{\mathfrak{C}}_{X|\mathbf{PA}})$.*

*Remark* 4.7. We draw on Brenier (1991)'s theorem to show that the causal mechanism is monotone. However, Brenier's map does not guarantee bijectivity by default. For this, we use Caffarelli (1992)'s standard regularity results to show the OT map is locally diffeomorphic. In practice, the bijectivity condition can be satisfied by smoothing empirical target distributions by adding mild continuous noise (e.g. uniform or Gaussian) to ensure the resulting distribution admits a smooth density.

Since dynamic OT maps are invertible when the velocity field $v_t$ is well-defined (e.g. Lipschitz continuous in space), we can recover the counterfactual transport map by composition (cf. Eq. (8)). By Proposition 4.4, if $T(u; \mathbf{pa})$ is monotone in $u$ then $T^*(\mathbf{pa}^*, \mathbf{pa}, x)$ is monotone in $x$. In the following section, we discuss the indeterminacy induced by the choice of prior on the exogenous noise distribution $P_U$ and characterise the necessary conditions for strict monotonicity of $T^*$ in $x$.

### 4.2.1 Exogenous Prior Indeterminacy

Since the true prior distribution over the exogenous variables $P_U$ is typically unknown, one must either rely on domain expertise to define a prior, choose one for mathematical convenience, or try to learn it directly from data. This choice of prior induces an indeterminacy in counterfactual outcomes.

Next, we show that regardless of the prior $P_U$ we choose, under standard regularity conditions, there exists a unique and optimal function $g^\star : \mathbb{R}^d \to \mathbb{R}^d$ that maps to any other $P_U$ we would have chosen.

---

[5]We omit the subscripts for simplicity, i.e., $\mathbf{PA} := \mathbf{PA}_X$ and similarly for their realisations: $\mathbf{pa} := \mathbf{pa}_x$.

**Definition 4.8** (Transport $\mathcal{L}_3$-Equivalence). Let $T^{(1)}, T^{(2)} : \mathbb{R}^d \times \mathbb{R}^k \to \mathbb{R}^d$ be transport maps with exogenous priors $P_U^{(1)}, P_U^{(2)}$ on $\mathbb{R}^d$, and parent domain $\mathbb{R}^k$. We say they are counterfactually equivalent $\sim_{\mathcal{L}_3}$, if and only if there exists a bijection $g : \mathbb{R}^d \to \mathbb{R}^d$ such that $g_\sharp P_U^{(2)} = P_U^{(1)}$, that is:

$$T^{(1)} \sim_{\mathcal{L}_3} T^{(2)} \iff \exists g \; : \; T^{(1)}(u; \mathbf{pa}) = T^{(2)}(g^{-1}(u); \mathbf{pa}) \quad \text{for } P_U^{(1)}\text{-a.e. } u, \; \forall \mathbf{pa}. \tag{11}$$

**Proposition 4.9** (Nasr-Esfahany et al. (2023)). *Transport maps $T^{(1)}, T^{(2)} : \mathbb{R}^d \times \mathbb{R}^k \to \mathbb{R}^d$ produce the same counterfactuals if and only if they are $\mathcal{L}_3$-equivalent in the sense of Definition 4.8:*

$$\forall x, \mathbf{pa}, \mathbf{pa}^* \; : \; x^* = T^{*(1)}(\mathbf{pa}^*, \mathbf{pa}, x) = T^{*(2)}(\mathbf{pa}^*, \mathbf{pa}, x) \iff T^{(1)} \sim_{\mathcal{L}_3} T^{(2)}. \tag{12}$$

*Remark* 4.10. Given only observational data ($\mathcal{L}_1$), we seek a transport map $T$ that is counterfactually equivalent ($\sim_{\mathcal{L}_3}$) to the true transport map $T^\star$ underlying the data-generating process.

**Lemma 4.11** (Existence of the Prior Transition Map). *Let $P_U^{(1)}, P_U^{(2)}$ be probability measures on $\mathbb{R}^d$ with finite second moments, both absolutely continuous w.r.t. the Lebesgue measure. Then, there exists a transport map $g : \mathbb{R}^d \to \mathbb{R}^d$ that is unique $P_U^{(1)}$-a.e., monotone, and a.e. bijective.*

We are now equipped to prove our main counterfactual identifiability result in the sense of Def. 4.8.

**Theorem 4.12** (Counterfactual Identifiability in Markovian SCMs). *Let $\dim(X) = \dim(U) = d > 1$, and $U \perp\!\!\!\perp \mathbf{PA}$. Assume $P_U$ is the continuous uniform measure on $[0,1]^d$. Let $T : \mathbb{R}^d \times \mathbb{R}^k \to \mathbb{R}^d$ be the time-1 dynamic OT map described in Lemma 4.6, which pushes $P_U$ forward to $P_{X|\mathbf{PA}}^{\mathfrak{C}}$. Then, the induced time-1 counterfactual dynamic OT map $T^* : \mathbb{R}^{2k} \times \mathbb{R}^d \to \mathbb{R}^d$ is strictly monotone in $x$:*

$$\langle T^*(\mathbf{pa}^*, \mathbf{pa}, x_1) - T^*(\mathbf{pa}^*, \mathbf{pa}, x_2), x_1 - x_2 \rangle > 0, \quad \forall x_1, x_2 \in \mathbb{R}^d, \mathbf{pa}, \mathbf{pa}^* \in \mathbb{R}^k. \tag{13}$$

*Remark* 4.13. This shows a dynamic OT flow mechanism (Lemma 4.6) yields a counterfactual optimal transport map $T^*$ that is strictly monotone in $x$ (i.e., monotonic in the vector sense), thereby extending the classical monotone-quantile notion to multivariate ($d > 1$) settings and, under stated Caffarelli (1992) regularity conditions, guaranteeing $\sim_{\mathcal{L}_3}$ identifiability from observational data ($\mathcal{L}_1$).

### 4.3 Non-Markovian Counterfactual Identifiability

Figure 1: **Four canonical causal graphs**. From left to right: (i) *Markovian*, (ii) *Instrumental Variable*, (iii) *Backdoor*, and (iv) *Frontdoor*. Dashed bidirected arcs denote unobserved confounding.

To ensure wide applicability, we extend counterfactual identification from observational data ($\mathcal{L}_1$) to *non-Markovian* multivariate ($d > 1$) settings under the following common criteria (Figure 1): (i) Instrumental Variable (IV), (ii) Backdoor Criterion (BC), and (iii) Frontdoor Criterion (FC). Appendix B.3 contains the proofs. Nasr-Esfahany et al. (2023) provided results for $d > 1$ only when BC applies. We expand the IV results to $d > 1$ thanks to the mechanism proposed in Lemma 4.6 being monotone, bijective and uniquely determinable under regularity conditions (Caffarelli, 1992). We also prove $\sim_{\mathcal{L}_3}$ identifiability for FC under similar conditions to BC (Nasr-Esfahany et al., 2023).

## 5 Counterfactual Transport Maps via Flow Matching

Although bijectivity and *variability* are sufficient for counterfactual identification under BC settings (see Appendix B.3), Markovian/IV settings require a monotone counterfactual OT map to obtain equivalent guarantees (Theorem 4.12). For the former, we prescribe standard continuous-time flows trained via flow matching, and for the latter, we approximate dynamic OT with these flows.

**Counterfactual Inference.** Computing counterfactuals with a continuous-time flow model is simple. Using *abduction-action-prediction* (Pearl, 2009), we first solve the associated ODE backwards in time for *abduction* $u = T^{-1}(x; \mathbf{pa})$, then forwards in time for *prediction* $x^* = T(u; \mathbf{pa}^*)$, as follows:

$$u = x - \int_0^1 v_t(x_t; \mathbf{pa}, \theta) \, \mathrm{d}t, \qquad \Rightarrow \qquad x^* = u + \int_0^1 v_t(x_t; \mathbf{pa}^*, \theta) \, \mathrm{d}t, \tag{14}$$

where $\mathbf{pa}^*$ is obtained from an upstream intervention (i.e. *action*) on one or more nodes in the associated SCM. The Picard-Lindelöf theorem states that both initial value problems admit unique solutions if $v_t(x_t; \mathbf{pa}, \theta)$ is Lipschitz continuous, a condition that is readily satisfiable in practice.

**Markovian OT Coupling.**   Although our theory for Markovian and IV settings is not tied to any particular OT approximation, it remains nontrivial to operationalise at scale with current techniques. To validate our theoretical claims, we offer a flow matching approach with a *bespoke* Markovian Batch-OT coupling that resolves counterfactual consistency issues in the standard formulation. Batch-OT flow matching (Pooladian et al., 2023; Tong et al., 2024) solves the OT problem on batches of source and target samples to form better pairs, using GPU-enabled solvers (Flamary et al., 2021). This asymptotically recovers the global OT map, but requires large batches in practice, especially in high dimensions (Klein et al., 2025). Notably, Mousavi-Hosseini et al. (2025) amortise OT pairing via semidiscrete couplings, reporting improved sample quality with substantially lower pairing cost.

As prescribed by Lemma 4.6, the correct way to formulate OT for Markovian structures must ensure that the marginal $P_U$ remains independent of $\mathbf{PA}$ upon coupling $P^{\mathfrak{C}}_{X|\mathbf{PA}}$. A naive application of Batch-OT flow matching to a Markovian setting implicitly entangles the parent variables $\mathbf{PA}$ and the exogenous noise $U$, which violates the independence requirement $U \perp\!\!\!\perp \mathbf{PA}$. This happens because upon sampling a random batch of exogenous noise $\{u^{(i)}\}_{i=1}^m \sim P_U$, and observations $\{(x^{(j)}, \mathbf{pa}^{(j)})\}_{j=1}^m \sim P^{\mathfrak{C}}_{X,\mathbf{PA}}$, solving the OT problem then reassigns each $u^{(i)}$ to a pair $(x^{(j)}, \mathbf{pa}^{(j)})$, but implicitly couples $u^{(i)}$ with $\mathbf{pa}^{(j)}$ by association with $x^{(j)}$. This invalidates the abduction process in Markovian settings as $U$ is no longer strictly exogenous, and the counterfactuals will be incorrect.

To fix this issue, we can instead sample batched data from the *conditional* distribution:

$$\{u^{(i)}\}_{i=1}^m \sim P_U, \qquad \mathbf{pa} \sim P^{\mathfrak{C}}_{\mathbf{PA}}, \qquad \{x^{(j)}\}_{j=1}^m \sim P^{\mathfrak{C}}_{X|\mathbf{PA}=\mathbf{pa}}, \qquad (15)$$

then solve the Batch-OT problem for each fixed value of $\mathbf{pa}$ in turn. Simply put, the parent values must match across each element in the batch to satisfy the Markovian constraint that $U \perp\!\!\!\perp \mathbf{PA}$.

The objective we optimise can be understood as solving a *family* of OT flow couplings:

$$\min_\theta \int_{\mathbb{R}^k} \mathbb{E}_{t\sim\mathcal{U}(0,1),(u,x)\sim\pi^{(m)}(\cdot|\mathbf{pa})} \left[ \left\| v_t(x_t; \mathbf{pa}, \theta) - v_t^\star(x_t \mid x, \mathbf{pa}) \right\|^2 \right] \mathrm{d}P^{\mathfrak{C}}_{\mathbf{PA}}(\mathbf{pa}), \qquad (16)$$

where the flow is $x_t = (1-t)u + tx$. Here $\pi^{(m)}$ denotes the implicit conditional joint distribution:

$$\pi^{(m)}(u, x \mid \mathbf{pa}) = \frac{1}{m} \sum_{i=1}^m \sum_{j=1}^m \pi^\star_{\mathbf{pa}}(i,j)\delta_{u^{(i)}}(u)\delta_{x^{(j)}}(x), \qquad (17)$$

induced by the OT coupling $\pi^\star_{\mathbf{pa}} \in \mathbb{R}^m \times \mathbb{R}^m$, for $m$ i.i.d. samples drawn according to Eq. (15).

# 6   Experiments

We conduct two sets of experiments: (i) a constructed scenario where the ground-truth counterfactuals are known and we can, in principle, verify our identifiability claims; (ii) a real-world medical imaging dataset widely used for counterfactual inference. When counterfactual ground truth is not available, we follow Monteiro et al. (2023); Ribeiro et al. (2023) and measure the counterfactual soundness axioms of *composition*, *effectiveness* and *reversibility*. While useful, these metrics alone do not imply identification and should not be construed as evidence of causal validity. For details, see Appendix D.

## 6.1   Counterfactual Ellipse Generation

For this study, we adapt the counterfactual ellipse generation setup by Nasr-Esfahany et al. (2023). Let $U \in \mathbb{R}^2$ be the semi-major and -minor parameters of an ellipse, $\mathrm{PA} \in (0, 2\pi)$ be an angle specifying a single point on the ellipse, and $X \in \mathbb{R}^2$ be its cartesian coordinates. The data-generating process is: $z \sim P_Z$, $u \sim P_{U|Z=z}$, $\mathrm{pa} \sim P_{\mathrm{PA}|Z=z}$, and $x := f(\mathrm{pa}, u)$. By construction, $U \perp\!\!\!\perp \mathrm{PA} \mid Z$, and $Z$ satisfies the backdoor criterion (BC) w.r.t. $\mathrm{PA} \to X$. To induce a Markovian setting, we randomise $\mathrm{PA}$, yielding $U \perp\!\!\!\perp \mathrm{PA}$. The goal is to learn a map $T : (0, 2\pi) \times \mathbb{R}^2 \to \mathbb{R}^2$ that can infer the set of counterfactual points $\mathcal{X}^* := \{x^* = T(\mathrm{pa}^*, T^{-1}(\mathrm{pa}, x)) \mid \mathrm{pa}^* \in (0, 2\pi)\}$, which draws the entire ellipse each observed $(\mathrm{pa}, x)$ belongs to. Importantly, since we always know the ground truth $\mathcal{X}^*$, we can evaluate our counterfactual estimates exactly using the mean average percentage error ($\mu_{\mathrm{APE}}$).

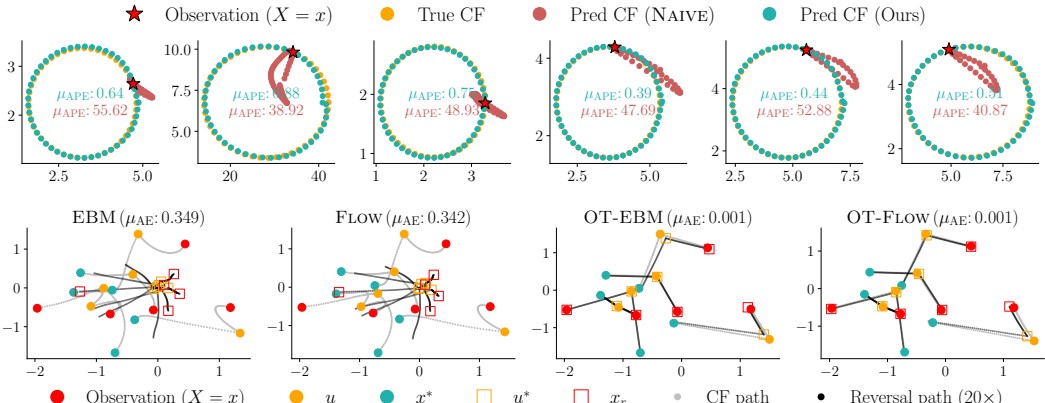

Figure 2: **Counterfactual ellipse generation**. (Top) Comparing our OT coupling flow to the naive approach (Section 5). (Bottom) OT maps exhibit greater counterfactual *reversibility*. A counterfactual $x^* = T_{\text{pa}^*} \circ T_{\text{pa}}^{-1}(x)$, is reversed by $x_r = T_{\text{pa}} \circ T_{\text{pa}^*}^{-1}(x^*)$, and a perfect reversal squares the circle.

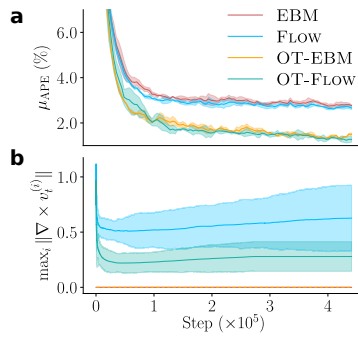

Figure 3: **(a)** CF error; **(b)** *curl* of the vector field during training.

Table 1: Counterfactual ellipse evaluation. As predicted by our theory, monotone flows are consistent in the Markovian setting, and bijectivity alone is sufficient under the front-door criterion.

|  |  | Baselines (Nasr-Esfahany et al., 2023) | | | |
|---|---|---|---|---|---|
| SCHEME |  | COND$_{\text{PA}}$ | COND$_{\text{PA},Z}$ | Markovian | BC$_Z$ |
| $\mu_{\text{APE}}$ (%) ↓ |  | 6607 | 6582 | 607 | 1.0 |
| (**Ours**) | NFE | EBM | FLOW | OT-EBM | OT-FLOW |
| Front-door | 2 | 19.7±.03 | 19.9±.30 | 1.64±.19 | 1.60±.01 |
|  | 10 | 5.79±.11 | 5.80±.08 | 1.42±.23 | 1.37±.14 |
|  | 50 | 1.79±.08 | 1.67±.10 | 1.42±.22 | 1.35±.18 |
| Markovian | 2 | 34.0±.90 | 33.6±.23 | 1.21±.04 | 1.06±.02 |
|  | 10 | 9.41±.24 | 9.20±.16 | 0.95±.02 | 0.78±.01 |
|  | 50 | 2.32±.01 | 2.30±.02 | 0.93±.02 | **0.76**±.01 |

We build four flow variants: (i) an energy-based model flow (EBM), which is curl-free by design; (ii) a continuous-time flow (Lipman et al., 2023) (FLOW)[6]; (iii) an EBM using the family of OT couplings described in Section 5 (OT-EBM); and (iv) same as (iii) but using a FLOW (denoted OT-FLOW). To show our OT coupling is necessary for correct counterfactual inferences, we compare with the 'naive' (NAIVE) version of a Batch-OT flow, which violates Markovianity (cf. Section 5). For all other details regarding architectures, datasets and experimental setup please refer to Appendix E.

Nasr-Esfahany et al. (2023) reports that their spline flow-based model *failed* in the Markovian case, with $\mu_{\text{APE}}=607\%$. It only succeeded using the BC$_Z$ scheme (Back-door) with a $\mu_{\text{APE}}$ of 1% (reproduced at .98%). Learning $P(X \mid \text{PA})$ or $P(X \mid \text{PA}, Z)$ also failed when $Z$ is a confounder (cf. COND$_{\text{PA}}$, COND$_{\text{PA},Z}$). Table 1 shows our flows produce near-exact ground-truth counterfactuals, and using just two function evaluations with OT. Counterfactual *reversibility* (Monteiro et al., 2023) is also improved by straighter paths (Figure 2). Further, we experiment with a non-Markovian setting with non-linear *unobserved* confounding where the front-door criterion applies. The results validate our theory in that bijectivity alone is sufficient for consistent counterfactual inference in this case.

## 6.2 Case Study: Chest X-ray Imaging Counterfactuals

To extend our study to high-dimensional settings, we conduct experiments on MIMIC-CXR (Johnson et al., 2019), a widely used dataset for counterfactual inference. Our assumed causal graph follows the baselines (Ribeiro et al., 2023; Xia et al., 2024), and includes SEX ($S$), RACE ($R$), AGE ($A$) and DISEASE ($D$) variables, where $A \rightarrow D$, and $\mathbf{PA}_X = \{S, R, A, D\}$ cause the X-ray image $X$. To parameterise our flow models, we use a streamlined version of Dhariwal and Nichol (2021)'s UNet

---

[6]Equivalent to Sanchez and Tsaftaris (2021) if the source distribution is Gaussian (Gao et al., 2025).

Table 2: Counterfactual *effectiveness* on MIMIC Chest X-ray (192×192). $|\Delta_{\text{AUC}}|$ denotes the absolute difference in ROCAUC of counterfactuals relative to the observed data baseline. For each variable, our results (blue shade) appear on the right, and baseline results (Ribeiro et al., 2023) are on the left. Note the large improvements for e.g. RACE. For more comparisons and ablations, see Appendix F.

| BASELINE | SEX $(S)$ AUC (%) ↑ | | RACE $(R)$ AUC (%) ↑ | | AGE $(A)$ MAE (yr) ↓ | | DISEASE $(D)$ AUC (%) ↑ | |
|---|---|---|---|---|---|---|---|---|
| Observed data | 99.63 | | 95.34 | | 6.197 | | 94.41 | |
| INTERVENTION | $|\Delta_{\text{AUC}}|$ (%) ↓ | | $|\Delta_{\text{AUC}}|$ (%) ↓ | | $\Delta_{\text{MAE}}$ (yr) ↓ | | $|\Delta_{\text{AUC}}|$ (%) ↓ | |
| $do(S=s)$ | 0.370 | $0.173_{\pm.02}$ | 11.44 | $0.583_{\pm.15}$ | 0.288 | $0.333_{\pm.06}$ | 2.490 | $0.023_{\pm.05}$ |
| $do(R=r)$ | 0.070 | $0.180_{\pm.01}$ | 8.640 | $0.050_{\pm.07}$ | 0.144 | $0.394_{\pm.10}$ | 7.010 | $0.310_{\pm.19}$ |
| $do(A=a)$ | 0.070 | $0.187_{\pm.03}$ | 14.64 | $1.197_{\pm.15}$ | 0.446 | $0.836_{\pm.08}$ | 2.810 | $0.347_{\pm.09}$ |
| $do(D=d)$ | 0.070 | $0.067_{\pm.00}$ | 16.04 | $0.627_{\pm.18}$ | 0.371 | $0.435_{\pm.05}$ | 3.790 | $2.280_{\pm.37}$ |
| $do(\text{rand})$ | 0.170 | $0.150_{\pm.02}$ | 12.54 | $0.730_{\pm.22}$ | 0.300 | $0.510_{\pm.05}$ | 0.590 | $0.640_{\pm.17}$ |

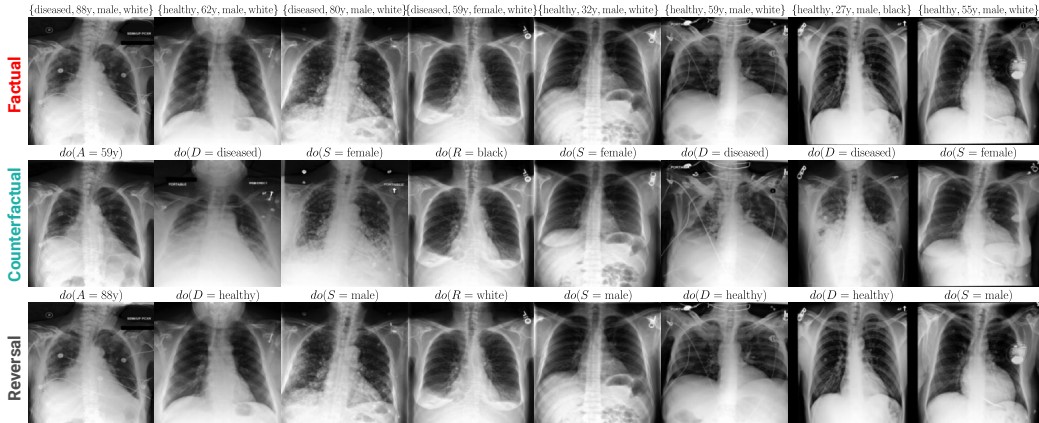

Figure 4: Qualitative counterfactual inference results on MIMIC Chest X-ray. We observe faithful, reversible interventions without requiring counterfactual fine-tuning, or classifier(-free) guidance.

architecture (see Appendix F for details). Table 2 and Figure 4 report our main results. We observe substantial improvements over baselines (Ribeiro et al., 2023; Xia et al., 2024) using our flows, across all three counterfactual soundness axioms, and without requiring any costly counterfactual fine-tuning or classifier(-free) guidance. That said, this alone does not imply causal validity. In Appendix F, we report additional comparisons and ablations. We observe performance trade-offs: for instance, OT-FLOW (which assumes Markovianity) outperforms on race interventions but underperforms FLOW on disease interventions, suggesting non-Markovian interaction effects or a subpar OT approximation. Notably, our Markovian OT coupling substantially improves over the NAIVE OT flow baseline.

## 7 Conclusion

Causal claims are credible only insofar as their identification is defensible, because observationally equivalent models can imply different answers. This work establishes a foundation for counterfactual identification of high-dimensional, multivariate outcomes (e.g. images) from observational data. We clarify the conditions under which flow matching yields identified counterfactuals for common causal structures, stating all assumptions and constraints explicitly to invite scrutiny and relaxation. Our results address causal validity concerns in prior work and extend to non-Markovian settings under standard criteria. For Markovian and instrumental variable settings, we characterise a continuous flow that is provably unique and monotone, yielding a rank-preserving dynamic OT map and consistent counterfactuals. Using continuous flows, we demonstrate near-exact ground-truth counterfactuals in controlled scenarios and improved axiomatic counterfactual soundness on real images relative to prior methods. Scaling OT to large problems remains a challenge. We conclude by urging practitioners to assess the causal validity of their estimates when using generative models for causal inference.

## Acknowledgments and Disclosure of Funding

We thank Aapo Hyvärinen and Charles Jones for helpful insights and discussions. F.R. and B.G. acknowledge the support of the UKRI AI programme, and the EPSRC, for CHAI-EPSRC Causality in Healthcare AI Hub (grant no. EP/Y028856/1). B.G. received support from the Royal Academy of Engineering as part of his Kheiron/RAEng Research Chair.

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

# Appendices

## Table of Contents

## A  Motivating Example: Monotonicity Requirement

We start with a simple instructive example in one dimension ($d = 1$). Suppose we have two causal mechanisms $T^{(1)}$ and $T^{(2)}$ defined as follows:

$$X = T^{(1)}(\mathbf{PA}, U) = \mathbf{PA} + U, \tag{18}$$

$$X = T^{(2)}(\mathbf{PA}, U) = \mathbf{PA} + 1 - U, \tag{19}$$

where $\mathbf{PA} \sim \mathcal{B}(0.5)$ is Bernoulli distributed, and $U \sim \mathcal{U}(0, 1)$ is continuous uniform. One can verify that both $T^{(1)}$ and $T^{(2)}$ induce the same conditional observational distribution:

$$P(X \mid \mathbf{PA} = 0) = \mathcal{U}(0, 1), \qquad P(X \mid \mathbf{PA} = 1) = \mathcal{U}(1, 2). \tag{20}$$

Without loss of generality, consider the counterfactual query

$$X_{\mathbf{pa}^* := 1} \mid \{\mathbf{PA} = 0, X = 0.8\}. \tag{21}$$

In the abduction step, $T^{(1)}$ and $T^{(2)}$ infer *different* exogenous noise values:

$$u^{(1)} = (T^{(1)})^{-1}(\mathbf{pa}, x) = x - \mathbf{pa} = 0.8 - 0 = 0.8, \tag{22}$$

$$u^{(2)} = (T^{(2)})^{-1}(\mathbf{pa}, x) = \mathbf{pa} + 1 - x = 0 + 1 - 0.8 = 0.2. \tag{23}$$

Nonetheless, both mechanisms produce the same counterfactuals:

$$x^{*(1)} = T^{(1)}(\mathbf{pa}^*, u^{(1)}) = \mathbf{pa}^* + u^{(1)} = 1 + 0.8 = 1.8, \tag{24}$$

$$x^{*(2)} = T^{(2)}(\mathbf{pa}^*, u^{(2)}) = \mathbf{pa}^* + 1 - u^{(2)} = 1 + 1 - 0.2 = 1.8. \tag{25}$$

Even if $u^{(1)} \neq u^{(2)}$, both $T^{(1)}$ and $T^{(2)}$ can return the same counterfactual outcomes because they assign the same **rank** $u$ (i.e., cumulative probability or quantile level) to the observed value $x$.

Concretely, since $T^{(1)}, T^{(2)}$ define the same observational distribution $P(X \mid \mathbf{PA})$, their conditional CDFs $F_{X|\mathbf{PA}} : \mathbb{R} \to [0, 1]$ match, and we can use them to perform abduction:

$$u = F_{X|\mathbf{PA}=0}(x) \tag{26}$$

$$= P(X \leq x \mid \mathbf{PA} = 0) \tag{27}$$

$$= \frac{x - a}{b - a} = \frac{0.8 - 0}{1 - 0} = 0.8, \tag{28}$$

where $(a, b) = (0, 1)$ since $P(X \mid \mathbf{PA} = 0) = \mathcal{U}(0, 1)$.

Then, we compute the counterfactual using the quantile function $F_{X|\mathbf{PA}=\mathbf{pa}^*}^{-1} : (0, 1) \to \mathbb{R}$, that is, the inverse conditional CDF under intervention:

$$x^* = F_{X|\mathbf{PA}=\mathbf{pa}^*}^{-1}(u) = a + u(b - a) = 1 + 0.8(1) = 1.8, \tag{29}$$

where $(a, b) = (1, 2)$, since we set $\mathbf{pa}^* := 1$ and $P(X \mid \mathbf{PA} = 1) = \mathcal{U}(1, 2)$.

The key here is the **strict monotonicity** of $T^{(1)}, T^{(2)}$ in $U$, that is for any $u^{(1)} < u^{(2)} \in \mathbb{R}$, we have

$$T^{(1)}(\cdot, u^{(1)}) < T^{(1)}(\cdot, u^{(2)}) \qquad \text{and} \qquad T^{(2)}(\cdot, u^{(1)}) > T^{(2)}(\cdot, u^{(2)}). \tag{30}$$

To illustrate the importance of this property, consider a third mechanism $T^{(3)}$ that induces the same conditional observational distribution:

$$X = T^{(3)}(\mathbf{PA}, U) = \begin{cases} U & \text{if } \mathbf{PA} = 0 \\ 2 - U & \text{if } \mathbf{PA} = 1 \end{cases}, \qquad U \sim \mathcal{U}(0, 1). \tag{31}$$

However, we can see that $T^{(3)}$ produces a different counterfactual for the same query used before:

$$u^{(3)} = (T^{(3)})^{-1}(\mathbf{pa}, x) = x = 0.8, \tag{32}$$

$$x^{*(3)} = T^{(3)}(\mathbf{pa}^*, u^{(3)}) = 2 - u^{(3)} = 1.2. \tag{33}$$

One can verify that, unlike for $T^{(1)}, T^{(2)}$, the rank is not being preserved under intervention:

$$F_{X|\mathbf{PA}=0}(x) = 0.8 > F_{X|\mathbf{PA}=\mathbf{pa}^*}(x^{*(3)}) = 0.2. \tag{34}$$

For a consistent counterfactual inference model, we would expect the inferred rank $u$ (i.e., the cumulative probability or quantile level) of an observed outcome $x$ given its parents $\mathbf{pa}$ to map to a counterfactual outcome $x^*$ that lies at the same rank in the counterfactual distribution:

$$x^* = F_{X|\mathbf{PA}=\mathbf{pa}^*}^{-1}(F_{X|\mathbf{PA}=\mathbf{pa}}(x)). \tag{35}$$

The mismatch observed previously occurs because $T^{(3)}$ is not monotonic in $U$ across different values of the parents $\mathbf{PA}$[7]. As a result, $T^{(3)}$ fails to preserve the rank order of outcomes across different interventions, thereby breaking counterfactual consistency and identifiability.

This rank-preservation condition can be viewed as the continuous analogue of the *monotone treatment response* (Manski, 1997; Angrist and Imbens, 1995) used in epidemiological studies, which rules out so-called *'defiers'*: units for whom exposure to disease risk factors would actually *reduce* disease risk. In both cases, monotonicity ensures that counterfactuals respect the domain knowledge-based intuition that exposure should not make disease less likely. As is often the case in causal analysis, the appropriateness of this condition is contingent on the particular context in which it is applied.

With the above $d = 1$ example in mind, in the following sections, we characterise the set of constraints on the map $T$ which permit (point-wise) counterfactual identification of causal mechanisms for multi-dimensional variables (i.e. $d > 1$); a key gap in existing literature. To achieve this, we connect ideas from dynamic optimal transport theory and graphical causality in ways previously unexplored.

---

[7]Note that $T^{(3)}$ is monotonic in $U$ for each fixed value of $\mathbf{PA}$, but not globally across both settings.

# B  Proofs: Counterfactual Identifiability

## B.1  Optimal Transport for Counterfactual Identification

**Proposition B.1** (Monotone Counterfactual Transport Map). *If a mechanism $x := f(\mathbf{pa}, u)$ is monotone in $u$, then the respective counterfactual transport map $T^*(\mathbf{pa}^*, \mathbf{pa}, x)$ is monotone in $x$.*

*Proof.* If $f(\mathbf{pa}, u_1) < f(\mathbf{pa}, u_2)$ for all $u_1 < u_2$, then its inverse must be monotonic in $x$ given $\mathbf{pa}$:

$$\forall x_1, x_2 \ : \ x_1 < x_2 \Rightarrow f^{-1}(\mathbf{pa}, x_1) < f^{-1}(\mathbf{pa}, x_2). \tag{36}$$

Applying the same logic given $\mathbf{pa}^*$, we have that their composition is also monotonic in $x$:

$$\forall x_1, x_2 \ : \ x_1 < x_2 \Rightarrow T^*(\mathbf{pa}^*, \mathbf{pa}, x_1) < T^*(\mathbf{pa}^*, \mathbf{pa}, x_2). \tag{37}$$

To generalise to multiple dimensions ($d > 1$), assume $f : \mathbb{R}^k \times \mathbb{R}^d \to \mathbb{R}^d$ is monotone in $u$:

$$\forall u_1, u_2 \ : \ \langle f(\mathbf{pa}, u_1) - f(\mathbf{pa}, u_2), u_1 - u_2 \rangle \geq 0, \tag{38}$$

and let

$$x_1 - x_2 = f(\mathbf{pa}, u_1) - f(\mathbf{pa}, u_2), \qquad u_1 - u_2 = f^{-1}(\mathbf{pa}, x_1) - f^{-1}(\mathbf{pa}, x_2). \tag{39}$$

By substituting in the above, we have that:

$$\langle f(\mathbf{pa}, u_1) - f(\mathbf{pa}, u_2), u_1 - u_2 \rangle = \langle x_1 - x_2, f^{-1}(\mathbf{pa}, x_1) - f^{-1}(\mathbf{pa}, x_2) \rangle \geq 0, \tag{40}$$

which by symmetry of the dot product, shows $f^{-1}$ is monotone in $x$. Lastly, to show when the counterfactual transport map $T^*(\mathbf{pa}^*, \mathbf{pa}, x) = f(\mathbf{pa}^*, f^{-1}(\mathbf{pa}, x))$ is also monotone in $x$ (for $d > 1$), suppose that the causal mechanism $f$ is equal to the gradient of a convex function:

$$\phi : \mathbb{R}^d \times \mathbb{R}^k \to \mathbb{R}, \qquad \forall u, \mathbf{pa} \ : \ f(\mathbf{pa}, u) = \nabla_u \phi(u; \mathbf{pa}). \tag{41}$$

Recall that gradients of convex functions are monotone:

$$\forall u_1, u_2, \mathbf{pa} \ : \ \langle \nabla_u \phi(u_1; \mathbf{pa}) - \nabla_u \phi(u_2; \mathbf{pa}), u_1 - u_2 \rangle \geq 0, \tag{42}$$

and therefore, the following must hold:

$$\forall x, \mathbf{pa}, \mathbf{pa}^* \ : \ T^*(\mathbf{pa}^*, \mathbf{pa}, x) = \nabla_u \phi(u; \mathbf{pa}^*)\big|_{u = f^{-1}(\mathbf{pa}, x)}, \tag{43}$$

$$\implies \forall x_1, x_2 \ : \ \langle T^*(\mathbf{pa}^*, \mathbf{pa}, x_1) - T^*(\mathbf{pa}^*, \mathbf{pa}, x_2), x_1 - x_2 \rangle_x \geq 0, \tag{44}$$

where $\langle v, w \rangle_x := v^\top \partial_x f^{-1}(\mathbf{pa}, x) w$, concluding the proof. $\qquad \square$

*Remark* B.2. Monotonicity of the counterfactual transport map $T^*$ w.r.t. $x$ guarantees that, for a given intervention, the rank order of factual outcomes is preserved in the counterfactuals. This is important for, e.g. *fairness*, as it prevents rank inversions across individuals under intervention. Euclidean monotonicity of $T^*$ (for $d > 1$) is guaranteed if the symmetric part of its Jacobian $\partial_x T^* = \partial_u f \partial_x f^{-1}$ is positive semi-definite; this holds, for example, if the two Jacobian components commute.

Next, we present Brenier's theorem, a celebrated result in optimal transport theory that we draw from in our counterfactual identifiability analysis of dynamic OT-based causal mechanisms.

**Theorem B.3** (Brenier (1991)). *Let $\mu$, $\nu$ be probability measures on $\Omega \subset \mathbb{R}^d$ with finite second moments, and suppose $\mu$ is absolutely continuous w.r.t. Lebesgue measure. Then, for the quadratic cost function $c(x, T(x)) = \|x - T(x)\|^2$, the Monge problem admits a unique optimal transport map $T : \Omega \to \Omega$ pushing $\mu$ forward to $\nu$, i.e., $T_\sharp \mu = \nu$. Moreover, this optimal map is the gradient of a convex function, $T = \nabla \phi$, and is monotone in the sense that:*

$$\forall x_1, x_2 \ : \ \langle T(x_1) - T(x_2), x_1 - x_2 \rangle \geq 0. \tag{45}$$

*Remark* B.4. This result provides a powerful geometric insight: under regularity conditions, the most efficient way of transporting mass from a source to a target distribution is by pushing it along the gradient of a convex function. The gradient structure ensures that the map does not 'fold', and the monotonicity condition reflects directional consistency in the sense that points which are initially close tend to remain close under the map, thereby preserving order and preventing overlaps.

**Lemma B.5** (Unique and Monotone Dynamic OT Mechanism). *Let* $\dim(X) = \dim(U) = d > 1$, *and consider the Markovian setting* $U \perp\!\!\!\perp \mathbf{PA}$. *Assume* $P_U$ *and* $P_{X|\mathbf{PA}}^{\mathfrak{C}}$ *are absolutely continuous w.r.t. the Lebesgue measure with strictly positive and bounded densities on bounded, open, convex domains. Let* $T : \mathbb{R}^d \times \mathbb{R}^k \to \mathbb{R}^d$ *be the time-1 map of a dynamic OT flow satisfying:*

$$\{T_t : t \in [0, 1]\}, \qquad \mathrm{d}T_t(u; \mathbf{pa}) = v_t(T_t(u; \mathbf{pa})) \, \mathrm{d}t, \qquad T_\sharp P_U = P_{X|\mathbf{PA}}^{\mathfrak{C}}. \tag{46}$$

*Then, there exists a convex function* $\phi : \mathbb{R}^d \to \mathbb{R}$ *such that:* $T(u; \mathbf{pa}) = \nabla_u \phi(u; \mathbf{pa})$, *for* $P_U(u)$-*a.e.* $u$, *where* $T$ *is monotone, bijective a.e., and uniquely determined by the pair* $(P_U, P_{X|\mathbf{PA}}^{\mathfrak{C}})$.

*Proof.* First, we invoke standard Caffarelli (1992) regularity conditions for OT (see Theorem 12.50 in Villani et al. (2008) for a helpful summary). Let $\Omega, \Lambda \subset \mathbb{R}^d$ be bounded, open, convex sets. The source $P_U$ and target $P_{X|\mathbf{PA}}^{\mathfrak{C}}$ are absolutely continuous w.r.t. Lebesgue measure, with densities $\rho_U$ and $\rho_{X|\mathbf{PA}}^{\mathfrak{C}}$ bounded above and below by positive constants on $\overline{\Omega}$ and $\overline{\Lambda}$, respectively. These conditions cover the strictest version of our results; weaker conclusions follow under standard relaxations.

According to Benamou and Brenier (2000)'s dynamic formulation of optimal transport, the time-1 map $T$ of a flow $\{T_t : t \in [0, 1]\}$ minimising the kinetic energy

$$\inf_{(\rho_t, v_t)} \int_0^1 \int_{\mathbb{R}^d \times \mathbb{R}^k} \|v_t(u; \mathbf{pa})\|^2 \rho_t(u, \mathbf{pa}) \, \mathrm{d}u \, \mathrm{d}\mathbf{pa} \, \mathrm{d}t \tag{47}$$

subject to the continuity equation

$$\partial_t \rho_t + \mathrm{div}(\rho_t v_t) = 0, \qquad \rho_0 = \rho_U, \qquad \rho_1 = \rho_{X|\mathbf{PA}}^{\mathfrak{C}}, \tag{48}$$

solves the Monge problem:

$$T^\star = \underset{T(\cdot;\mathbf{PA})_\sharp P_U = P_{X|\mathbf{PA}}^{\mathfrak{C}}}{\arg\min} \int \|u - T(u; \mathbf{pa})\|^2 \, \mathrm{d}P_U(u). \tag{49}$$

Since $P_U \ll \mathcal{L}^d$ (i.e. absolutely continuous w.r.t. the Lebesgue measure), Brenier (1991)'s theorem (cf. Theorem B.3) applies, and implies that the optimal transport map $T^\star$ is of the form:

$$T^\star(u; \mathbf{pa}) = \nabla_u \phi(u; \mathbf{pa}), \tag{50}$$

where $\phi$ is convex on $\Omega$. Further, this gradient map is well-defined $P_U$-almost everywhere and is uniquely determined by $P_U$ and $P_{X|\mathbf{PA}}^{\mathfrak{C}}$. Moreover, since $\nabla\phi$ pushes $P_U$ forward to $P_{X|\mathbf{PA}}^{\mathfrak{C}}$, this implies that for almost every $u \in \Omega$, it satisfies the Monge-Ampère equation:

$$\rho_U(u) = \rho_{X|\mathbf{PA}}^{\mathfrak{C}} \left( \nabla_u \phi(u; \mathbf{pa}) \right) \det D_u^2 \phi(u; \mathbf{pa}). \tag{51}$$

If $\phi$ is differentiable and strictly convex, then $\nabla\phi$ is strictly monotone and therefore injective:

$$\left\langle \nabla_u \phi(u^{(1)}; \mathbf{pa}) - \nabla_u \phi(u^{(2)}; \mathbf{pa}), u^{(1)} - u^{(2)} \right\rangle > 0 \quad \text{for} \quad u^{(1)} \neq u^{(2)}. \tag{52}$$

In particular, $\nabla_u \phi(u^{(1)}; \mathbf{pa}) = \nabla_u \phi(u^{(2)}; \mathbf{pa})$ implies $u^{(1)} = u^{(2)}$, and thus $T$ is injective almost everywhere. Moreover, since $T_\sharp P_U = P_{X|\mathbf{PA}}^{\mathfrak{C}}$, no Borel set $A \subset \Lambda$ of positive $P_{X|\mathbf{PA}}^{\mathfrak{C}}$-measure can be missed by $T$ and must satisfy $P_U(T^{-1}(A)) > 0$, hence $T$ is onto $\Lambda$ up to null sets.

The regularity of $\phi$ now follows from the theory of elliptic partial differential equations, specifically from Caffarelli (1992)'s well-known interior and global regularity results for convex solutions to the Monge-Ampère equation. If $\rho_U$ and $\rho_{X|\mathbf{PA}}^{\mathfrak{C}}$ are bounded above and below by positive constants on $\overline{\Omega}, \overline{\Lambda}$ and are $C^\alpha$, then by Caffarelli (1992)'s theory the convex potential $\phi \in C^{2,\alpha}(\overline{\Omega})$; in particular, $D_u^2 \phi(u; \mathbf{pa})$ is almost everywhere positive definite and $\nabla\phi$ is locally a diffeomorphism. Hence, we conclude that $T(u; \mathbf{pa}) = \nabla_u \phi(u; \mathbf{pa})$, $P_U$-almost everywhere, and that $T$ is monotone, almost everywhere bijective, and uniquely determined by the pair $(P_U, P_{X|\mathbf{PA}}^{\mathfrak{C}})$. $\square$

*Remark B.6.* We draw on Brenier (1991)'s theorem to show that the causal mechanism is monotone. However, Brenier's map does not guarantee bijectivity by default. For this, we must invoke Caffarelli (1992)'s standard regularity results for OT, to ensure the map is locally diffeomorphic. In practice, this condition is often satisfied by smoothing empirical target distributions by adding mild continuous noise (e.g. uniform or Gaussian) to ensure the resulting distribution admits a density.

## B.2 Exogenous Prior Indeterminacy

Given only observational data ($\mathcal{L}_1$), our goal is to find a transport map $T$ that is counterfactually equivalent ($\sim_{\mathcal{L}_3}$) to the true transport map $T^\star$ underlying the data-generating process.

**Definition B.7** (Transport $\mathcal{L}_3$-Equivalence). Let $T^{(1)}, T^{(2)} : \mathbb{R}^d \times \mathbb{R}^k \to \mathbb{R}^d$ be transport maps with exogenous priors $P_U^{(1)}, P_U^{(2)}$ on $\mathbb{R}^d$, and parent domain $\mathbb{R}^k$. We say they are counterfactually equivalent $\sim_{\mathcal{L}_3}$, if and only if there exists a bijection $g : \mathbb{R}^d \to \mathbb{R}^d$ such that $g_\sharp P_U^{(2)} = P_U^{(1)}$, that is:

$$T^{(1)} \sim_{\mathcal{L}_3} T^{(2)} \iff \exists g : T^{(1)}(u; \mathbf{pa}) = T^{(2)}(g^{-1}(u); \mathbf{pa}) \quad \text{for } P_U^{(1)}\text{-a.e. } u, \forall \mathbf{pa}. \tag{53}$$

**Proposition B.8** (Nasr-Esfahany et al. (2023)). *Transport maps $T^{(1)}, T^{(2)} : \mathbb{R}^d \times \mathbb{R}^k \to \mathbb{R}^d$ produce the same counterfactuals if and only if they are $\mathcal{L}_3$-equivalent in the sense of Definition B.7:*

$$\forall x, \mathbf{pa}, \mathbf{pa}^* : x^* = T^{*(1)}(\mathbf{pa}^*, \mathbf{pa}, x) = T^{*(2)}(\mathbf{pa}^*, \mathbf{pa}, x) \iff T^{(1)} \sim_{\mathcal{L}_3} T^{(2)}. \tag{54}$$

*Proof.* We provide a proof here for completeness. An arbitrary counterfactual query $X_{\mathbf{pa}^*} \mid \{X = x, \mathbf{PA} = \mathbf{pa}\}$, where $\{X = x, \mathbf{PA} = \mathbf{pa}\}$ is the observed evidence, is answerable by any two equivalent transport maps $T^{(1)}, T^{(2)} : \mathbb{R}^d \times \mathbb{R}^k \to \mathbb{R}^d$ via abduction-action-prediction.

We first prove $\Leftarrow$.

In the abduction step, both maps use the observed evidence to infer the exogenous noise:

$$u^{(1)} = \left(T^{(1)}\right)^{-1}(x; \mathbf{pa}), \qquad u^{(2)} = (T^{(2)})^{-1}(x; \mathbf{pa}), \tag{55}$$

and their equivalence (cf. Definition B.7) implies that:

$$\exists g : u^{(1)} = g(u^{(2)}). \tag{56}$$

In the action-prediction step, for any intervention on the parents, we have that:

$$T^{(1)}(u^{(1)}; \mathbf{pa}^*) = T^{(1)}(g(u^{(2)}); \mathbf{pa}^*) = T^{(2)}(g^{-1}(u^{(1)}); \mathbf{pa}^*) = T^{(2)}(u^{(2)}; \mathbf{pa}^*), \tag{57}$$

thus both maps produce the same counterfactual outcomes:

$$\forall x, \mathbf{pa}, \mathbf{pa}^* : x^* = T^{(1)}(\mathbf{pa}^*, \mathbf{pa}, x) = T^{(2)}(\mathbf{pa}^*, \mathbf{pa}, x). \tag{58}$$

We now prove $\Rightarrow$.

Suppose, for the sake of contradiction, that the function $g$ depends on the parents $\mathbf{pa}$, then:

$$\forall u^{(1)}, \mathbf{pa} : T^{(1)}(u^{(1)}; \mathbf{pa}) = T^{(2)}(g^{-1}(u^{(1)}; \mathbf{pa}); \mathbf{pa}). \tag{59}$$

Since $T^{(1)}, T^{(2)}$ produce identical counterfactuals, they must agree for any fixed $\mathbf{pa}^*$:

$$T^{(1)}(u^{(1)}; \mathbf{pa}^*) = T^{(2)}(g^{-1}(u^{(1)}; \mathbf{pa}); \mathbf{pa}^*), \tag{60}$$

where the abduction step is given by:

$$g^{-1}(u^{(1)}; \mathbf{pa}) = u^{(2)} = (T^{(2)})^{-1}(x; \mathbf{pa}). \tag{61}$$

Since Eq. (59) also holds when $\mathbf{pa} := \mathbf{pa}^*$, we have that:

$$T^{(1)}(u^{(1)}; \mathbf{pa}^*) = T^{(2)}(g^{-1}(u^{(1)}; \mathbf{pa}^*); \mathbf{pa}^*) \tag{62}$$

$$\implies T^{(2)}(g^{-1}(u^{(1)}; \mathbf{pa}); \mathbf{pa}^*) = T^{(2)}(g^{-1}(u^{(1)}; \mathbf{pa}^*); \mathbf{pa}^*). \tag{63}$$

Since $T^{(2)}$ is injective (i.e. bijective in $u^{(2)}$ by construction), the above implies that:

$$\forall \mathbf{pa}, \mathbf{pa}^* : g^{-1}(u^{(1)}; \mathbf{pa}) = g^{-1}(u^{(1)}; \mathbf{pa}^*), \tag{64}$$

$$\implies \forall \mathbf{pa} : g^{-1}(u^{(1)}; \mathbf{pa}) = g^{-1}(u^{(1)}). \tag{65}$$

This contradicts the assumption that $g$ depends on $\mathbf{pa}$, completing the proof. $\qquad\square$

**Lemma B.9** (Existence of the Prior Transition Map). *Let $P_U^{(1)}, P_U^{(2)}$ be probability measures on $\mathbb{R}^d$ with finite second moments, both absolutely continuous w.r.t. the Lebesgue measure. Then, there exists a transport map $g : \mathbb{R}^d \to \mathbb{R}^d$ that is unique $P_U^{(1)}$-a.e., monotone, and a.e. bijective:*

$$\inf_{g_\sharp P_U^{(1)} = P_U^{(2)}} \int_{\mathbb{R}^d} c(u, g(u)) \, dP_U^{(1)}(u), \qquad c(u, g(u)) := \|u - g(u)\|^2. \tag{66}$$

*Proof.* If $P_U^{(1)}, P_U^{(2)}$ have finite second moments and $P_U^{(1)} \ll \mathcal{L}^d$, by Brenier (1991) (Theorem B.3), there exists a unique $P_U^{(1)}$-a.e., monotone, map $g = \nabla\phi$. Since $P_U^{(2)} \ll \mathcal{L}^d$, the map is a.e. bijective. Under standard Caffarelli (1992) regularity conditions (bounded densities and convex domains), we can upgrade a.e. invertibility to a homeomorphism/diffeomorphism between supports.

To fix a common reference, we invoke the higher-dimensional Probability Integral Transform (PIT), known as the Rosenblatt transform. It states that any absolutely continuous probability measure $P_U$ on $\mathbb{R}^d$ (with $d > 1$) whose density is strictly positive almost everywhere can be represented as the push-forward of the uniform $\mathcal{U}([0,1]^d)$ via a measurable and almost everywhere invertible map.

More concretely, first consider the one-dimensional case ($d = 1$). For any random variable $X \in \mathbb{R}$, with cumulative distribution function (CDF) $F_X : \mathbb{R} \to [0, 1]$, the PIT states that:

$$U = F_X(X), \qquad\qquad X = F_X^{-1}(U), \tag{67}$$

where $U$ is uniformly distributed on $[0, 1]$. To extend this to higher dimensions ($d > 1$), we use a recursive component-wise construction known as the Rosenblatt transform, explained below.

Let $U \sim \mathcal{U}([0,1]^d)$, and define a map $T : [0,1]^d \to \mathbb{R}^d$ component-wise in a triangular manner. Further, write $(u_1, \ldots, u_d)$ for coordinates in $[0,1]^d$, and let $X = (X_1, \ldots, X_d)$ be a random vector distributed according to $P_U$. Now set $T_1(u_1) = F_{X_1}^{-1}(u_1)$.

For each $k = 2, \ldots, d$, let $F_{X_k|X_1,\ldots,X_{k-1}}$ be the conditional CDF of $X_k$ given $(X_1, \ldots, X_{k-1})$, then define:

$$T_k(u_1, \ldots, u_k) = \left[ F_{X_k|X_1,\ldots,X_{k-1}} \left( \cdot \mid T_1(u_1), \ldots, T_{k-1}(u_1, \ldots, u_{k-1}) \right) \right]^{-1} (u_k). \tag{68}$$

Under mild continuity assumptions on each conditional CDF (which follow from strict positivity of the density of $P_U$), this construction is well-defined for almost every $(u_1, \ldots, u_d) \in [0,1]^d$.

The resulting map

$$T(u_1, \ldots, u_d) = (T_1(u_1), T_2(u_1, u_2), \ldots, T_d(u_1, \ldots, u_d)) \tag{69}$$

is measurable and invertible almost everywhere. Since $U \sim \mathcal{U}([0,1]^d)$, one can verify that $T(U)$ has law $P_U$, or equivalently that $T_\sharp \mathcal{U}([0,1]^d) = P_U$. Finally, for any two distinct absolutely continuous probability measures $P_U^{(1)}$ and $P_U^{(2)}$ on $\mathbb{R}^d$, one may similarly construct maps $T^{(1)}, T^{(2)} : [0,1]^d \to \mathbb{R}^d$ such that:

$$T_\sharp^{(1)} \mathcal{U}([0,1]^d) = P_U^{(1)}, \qquad\qquad T_\sharp^{(2)} \mathcal{U}([0,1]^d) = P_U^{(2)}. \tag{70}$$

With an almost-everywhere inverse $(T^{(2)})^{-1}$ (i.e. $(T^{(2)})^{-1} \circ T^{(2)} = \text{id}$ a.e.), it follows that the transport map

$$g = \left( T^{(2)} \right)^{-1} \circ T^{(1)}, \qquad \text{satisfies} \qquad g_\sharp P_U^{(1)} = P_U^{(2)}. \tag{71}$$

Thus, both measures $P_U^{(1)}$ and $P_U^{(2)}$ can be considered push-forwards of the same uniform measure on $[0,1]^d$, via different invertible maps, showing that any particular choice of $P_U$ is essentially arbitrary. Without loss of generality, we may therefore fix the exogenous prior as $P_U = \mathcal{U}([0,1]^d)$, and interpret all subsequent distributions as push-forwards thereof, concluding the proof. $\qquad\square$

*Remark* B.10. We use the Rosenblatt transform only to fix a common reference for the result; any absolutely continuous law can be written as $T_\sharp \mathcal{U}([0,1]^d)$ for a measurable, a.e.-invertible triangular $T$. This does not imply that $T$ is the quadratic-cost OT (Brenier) map. Note that the Rosenblatt transform depends on the chosen coordinate order, meaning that different permutations yield different triangular maps (each still pushes $\mathcal{U}([0,1]^d)$ to the same target law). This indicates that rank-preserving continuous-time flows without OT are realisable by, for example, assuming a coordinate order and parameterising the flow with a component-wise autoregressive model. By contrast, the Brenier OT map is order-invariant and is determined solely by the two measures and the quadratic cost.

**Theorem B.11** (Counterfactual Identifiability in Markovian SCMs). *Let* $\dim(X) = \dim(U) = d > 1$, *and* $U \perp\!\!\!\perp \mathbf{PA}$. *Assume* $P_U$ *is the continuous uniform measure on* $[0,1]^d$. *Let* $T : \mathbb{R}^d \times \mathbb{R}^k \to \mathbb{R}^d$ *be the time-1 dynamic OT map described in Lemma 4.6, which pushes* $P_U$ *forward to* $P^{\mathfrak{C}}_{X|\mathbf{PA}}$. *Then, the induced time-1 counterfactual dynamic OT map* $T^* : \mathbb{R}^{2k} \times \mathbb{R}^d \to \mathbb{R}^d$ *is strictly monotone in* $x$:

$$\langle T^*(\mathbf{pa}^*, \mathbf{pa}, x_1) - T^*(\mathbf{pa}^*, \mathbf{pa}, x_2), x_1 - x_2 \rangle > 0, \quad \forall x_1, x_2 \in \mathbb{R}^d, \mathbf{pa}, \mathbf{pa}^* \in \mathbb{R}^k. \tag{72}$$

*Proof.* We present a generalised notion of the monotonicity requirement for multi-dimensional causal mechanisms ($d > 1$), and show counterfactual identifiability can be achieved using the dynamic OT map described in Lemma B.5.

Let $\dim(X) = \dim(U) = d \geq 1$, and $U \perp\!\!\!\perp \mathbf{PA}$. Let $P_U$ be a uniform measure $\mathcal{U}([0,1]^d)$, and assume $P^{\mathfrak{C}}_{X|\mathbf{PA}}$ is absolutely continuous w.r.t. the Lebesgue measure. Let $T : \mathbb{R}^d \times \mathbb{R}^k \to \mathbb{R}^d$ be the time-1 map of a dynamic OT flow $\{T_t : t \in [0,1]\}$ described in Lemma B.5, that is:

$$\mathrm{d}T_t(u; \mathbf{pa}) = v_t(T_t(u; \mathbf{pa})) \, \mathrm{d}t, \qquad T(\cdot; \mathbf{PA})_\sharp P_U = P^{\mathfrak{C}}_{X|\mathbf{PA}}. \tag{73}$$

By Theorem B.3, there exists an optimal map that minimises quadratic transport cost:

$$T(u; \mathbf{pa}) = \nabla_u \phi(u; \mathbf{pa}), \tag{74}$$

where $\phi : \mathbb{R}^d \to \mathbb{R}$ is a convex function. Importantly, $T$ is well-defined $P_U$-almost everywhere and is uniquely determined by $P_U$ and $P^{\mathfrak{C}}_{X|\mathbf{PA}}$. This gradient map generalises the notion of one-dimensional monotonicity, as it is 'vector monotonic' in the sense that:

$$\forall u_1, u_2, \mathbf{pa} : \langle T(u_1; \mathbf{pa}) - T(u_2; \mathbf{pa}), u_1 - u_2 \rangle \geq 0. \tag{75}$$

Since $P^{\mathfrak{C}}_{X|\mathbf{PA}}$ is assumed to be absolutely continuous, under Caffarelli (1992) regularity conditions (Lemma B.5), the map $T$ is also bijective. Then, it follows that:

$$T^{-1}(T(u; \mathbf{pa}); \mathbf{pa}) = u, \qquad u \in [0,1]^d, \tag{76}$$

where the respective maps are realised by solving the associated ODE backwards in time for abduction:

$$u_0 = T^{-1}(x; \mathbf{pa}) = x - \int_0^1 v_t(T_t(u; \mathbf{pa})) \, \mathrm{d}t, \tag{77}$$

and forwards in time for prediction of $x^*$, for instance, under a chosen intervention $\mathbf{pa}^*$:

$$x^* = T(u_0; \mathbf{pa}^*) = u_0 + \int_0^1 v_t(T_t(u; \mathbf{pa}^*)) \, \mathrm{d}t. \tag{78}$$

By the Picard-Lindelöf theorem, both initial value problems admit unique solutions if the associated velocity field is Lipschitz continuous, which is readily satisfiable in practice (Chen et al., 2018).

Under Caffarelli (1992)'s standard regularity and smoothness conditions (Lemma B.5), the time-1 dynamic transport map $T$ arises as the gradient of a *strictly* convex potential $T(u; \mathbf{pa}) = \nabla_u \phi(u; \mathbf{pa})$, and its Jacobian $\partial_u T(u; \mathbf{pa}) = \nabla_u^2 \phi(u; \mathbf{pa})$ is symmetric positive-definite (SPD).

The Jacobian of the counterfactual transport map

$$T^*(\mathbf{pa}^*, \mathbf{pa}, x) := T(T^{-1}(x; \mathbf{pa}); \mathbf{pa}^*) \tag{79}$$

is thus given by the product of two SPD factors:

$$\partial_x T^*(\mathbf{pa}^*, \mathbf{pa}, x) = \partial_u T(u; \mathbf{pa}^*)\big|_{u=T^{-1}(x;\mathbf{pa})} \cdot \partial_x T^{-1}(x; \mathbf{pa}). \tag{80}$$

For two distinct points $x_1, x_2 \in \Omega$, define the straight line segment

$$\forall t \in [0,1], \qquad x_t = (1-t)x_2 + tx_1, \qquad \mathrm{d}x_t = (x_1 - x_2) \, \mathrm{d}t =: z \, \mathrm{d}t, \tag{81}$$

where $z \neq 0$. Applying the fundamental theorem of calculus, then the chain rule, we get:

$$T^*(\mathbf{pa}^*, \mathbf{pa}, x_1) - T^*(\mathbf{pa}^*, \mathbf{pa}, x_2) = \int_0^1 \frac{\mathrm{d}}{\mathrm{d}t}\Big[T^*(\mathbf{pa}^*, \mathbf{pa}, x_t)\Big] \mathrm{d}t \tag{82}$$

$$= \int_0^1 \partial_{x_t} T^*(\mathbf{pa}^*, \mathbf{pa}, x_t) z \, \mathrm{d}t. \tag{83}$$

Taking the inner product with $z$ in the form $\langle v, w \rangle_x := v^\top \partial_x T^{-1}(x; \mathbf{pa})w$ yields:

$$\langle T^*(\mathbf{pa}^*, \mathbf{pa}, x_1) - T^*(\mathbf{pa}^*, \mathbf{pa}, x_2), z \rangle = \int_0^1 \langle \partial_{x_t} T^* z, z \rangle_{x_t} \, \mathrm{d}t, \tag{84}$$

$$= \int_0^1 z^\top \left( \partial_{x_t} T^{-1} \partial_u T \partial_{x_t} T^{-1} \right) z \, \mathrm{d}t \tag{85}$$

$$= \int_0^1 \left( \partial_{x_t} T^{-1} z \right)^\top \partial_u T \left( \partial_{x_t} T^{-1} z \right) \, \mathrm{d}t > 0, \tag{86}$$

because $\partial_u T$ is SPD and $\partial_{x_t} T^{-1} z \neq 0$ for each $t \in [0, 1]$ (since $\partial_x T^{-1}$ is SPD). Hence the integral is strictly positive and the counterfactual transport map $T^*$ is strictly monotone in $x$, as required. $\square$

*Remark* B.12. An intuition for the above result is provided next. Since the time-1 transport map $T : [0, 1]^d \times \mathbb{R}^k \to \mathbb{R}^d$ exists, is uniquely optimal, and (Brenier-)monotone, it generalises the notion of a scalar quantile function to vector-valued outcomes (Carlier et al., 2016). Further, since the inverse map $T^{-1} : \mathbb{R}^d \times \mathbb{R}^k \to [0, 1]^d$ also exists and shares these properties, it serves as a vector rank function, and can be used to uniquely recover $u$ analogously to the $d = 1$ case in Appendix A.

The proof above establishes that $T^*$ is strictly monotone in $x$, and this represents the multivariate (Brenier/OT) notion of monotonicity and is sufficient for rank preservation (Carlier et al., 2016). If one additionally wants coordinate-wise (product-order) monotonicity, it holds under extra structure. For example, when $\nabla_u^2 \phi(u; \mathbf{pa})$ has nonnegative off-diagonal entries, so $T(\cdot; \mathbf{pa})$ is coordinate-wise increasing. If, for both $\mathbf{pa}$ and $\mathbf{pa}^*$, the map $T(\cdot; \mathbf{pa})$ is an order-isomorphism for the product order (hence $T^{-1}(\cdot; \mathbf{pa})$ is also coordinate-wise increasing), then the composition of the two $T^*(\mathbf{pa}^*, \mathbf{pa}, x) = T(T^{-1}(x; \mathbf{pa}); \mathbf{pa}^*)$ is coordinate-wise increasing in $x$.

Importantly, for quadratic cost with an absolutely continuous source, the OT map is unique a.e. and does not depend on a coordinate ordering; its monotonicity is in the Brenier/OT sense (i.e., it is the gradient of a convex potential). This sidesteps the non-uniqueness inherent in coordinate-wise (product) orders, or Rosenblatt transforms, which depend on an arbitrary permutation of coordinates.

## B.3 Non-Markovian Counterfactual Identifiability

In the following, we extend our theoretical analysis to non-Markovian settings. We provide counterfactual identifiability results from observational data under the following standard criteria: (i) Forward Criterion (FC); (ii) Instrumental Variable(s) (IV); (iii) Backdoor Criterion (BC). The proofs for BC ($d \geq 1$) and IV ($d = 1$) criteria are provided by Nasr-Esfahany et al. (2023). We provide a proof for FC ($d \geq 1$) and extend Nasr-Esfahany et al. (2023)'s IV result to $d > 1$.

### B.3.1 Frontdoor Criterion Setting

A setting in which counterfactual identifiability under an unobserved confounding $U$ on $(A, X)$ is possible (under stated assumptions) is through the frontdoor criterion, using a mediator variable $Z$ (Pearl, 2009). For FC to apply, the following must hold: (i) $Z$ intercepts all directed paths from $A$ to $X$; (ii) no backdoor path from $A$ to $Z$ exists; (iii) $A$ blocks all backdoor paths from $Z$ to $X$. The $\mathcal{L}_3$-equivalence result presented below also applies to the multi-dimensional ($d > 1$) setting.

**Lemma B.13** (FC $\mathcal{L}_3$-Equivalence). *Consider two FC models $T$ and $\hat{T}$ with the same observational joint law $(A, Z, X)$, where $A \to Z \to X$ and the pair $(A, X)$ is confounded by $U$, and $\hat{U}$, respectively. The bijective structural assignments are*

$$T(u; a) = \left( T_Z(a), \ T_X(u; T_Z(a)) \right), \quad \text{and} \quad \hat{T}(\hat{u}; a) = \left( \hat{T}_Z(a), \ \hat{T}_X(\hat{u}; \hat{T}_Z(a)) \right). \tag{87}$$

*The maps $T$ and $\hat{T}$ are counterfactually equivalent ($\sim_{\mathcal{L}_3}$) if*

*1. (Frontdoor Criterion):*

$$U \perp\!\!\!\perp Z \mid A \text{ and } \hat{U} \perp\!\!\!\perp Z \mid A, \quad X \perp\!\!\!\perp A \mid \{U, Z\} \text{ and } X \perp\!\!\!\perp A \mid \{\hat{U}, Z\}. \tag{88}$$

*2. (Regularity) For every $z$, the derivatives $\nabla_z |\det \mathbf{J}_{T_X^{-1}}(\cdot \, ; z)|$ and $\nabla_z |\det \mathbf{J}_{\hat{T}_X}(\cdot \, ; z)|$ exist.*

*3. (Variability) For every $u$, there exist $d+1$ points $a_1, \ldots, a_{d+1}$ such that*

$$M(u; a_1, \ldots, a_{d+1}) = \begin{bmatrix} \rho_{U|A}(u \mid a_1) & \nabla_u \rho_{U|A}(u \mid a_1) \\ \vdots & \vdots \\ \rho_{U|A}(u \mid a_{d+1}) & \nabla_u \rho_{U|A}(u \mid a_{d+1}) \end{bmatrix} \tag{89}$$

*is full rank, i.e.* $\det M \neq 0$.

*Proof.* If $Z \mid A$ is *deterministic* in both models, then matching the joint law of $(A, Z)$ forces:

$$Z = T_Z(A), \qquad Z = \hat{T}_Z(A) \quad \Rightarrow \quad T_Z(A) = \hat{T}_Z(A) \text{ a.s.} \tag{90}$$

If $A$ is discrete with $P(A = a) > 0$, or has density strictly positive on an open $\mathcal{A}$ and $T_Z, \hat{T}_Z$ are continuous on $\mathcal{A}$, then $T_Z(a) = \hat{T}_Z(a)$ for all $a$. If $Z \mid A$ is instead *stochastic*, with bijective maps w.r.t. noise variables $W, \hat{W}$, counterfactual equivalence (Definition B.7) states that:

$$T_Z \sim_{\mathcal{L}_3} \hat{T}_Z \iff \exists h, \forall w, a \; : \; T_Z(w; a) = \hat{T}_Z(h^{-1}(w); a), \tag{91}$$

where $h$ is a bijection. By Lemma B.9, $h$ exists, therefore, $T_Z \sim_{\mathcal{L}_3} \hat{T}_Z$ holds.

Next, we prove $T_X \sim_{\mathcal{L}_3} \hat{T}_X$ under similar conditions to [Nasr-Esfahany et al. (2023)](https://)'s BC result. First, define a link function of both transport maps on the exogenous noise channel:

$$g(\cdot; z) := T_X^{-1}(\cdot; z) \circ \hat{T}_X(\cdot; z), \qquad u = g(\hat{u}; z), \tag{92}$$

which, by the change-of-variables formula:

$$\rho_{\hat{U}|A,Z}(\hat{u} \mid a, z) = \rho_{U|A,Z}(g(\hat{u}; z) \mid a, z) \left| \det \mathbf{J}_g(\hat{u}; z) \right|. \tag{93}$$

Under FC (condition 1), we have that $U \perp\!\!\!\perp Z \mid A$ and $\hat{U} \perp\!\!\!\perp Z \mid A$, respectively:

$$\rho_{\hat{U}|A}(\hat{u} \mid a) = \rho_{U|A}(g(\hat{u}; z) \mid a) \left| \det \mathbf{J}_g(\hat{u}; z) \right|. \tag{94}$$

Given *regularity* (condition 2), differentiate both sides of Eq. (94) w.r.t. $z_i$:

$$0 = \left( \nabla_u \rho_{U|A}(u \mid a) \right)^\top \frac{\partial g(\hat{u}; z)}{\partial z_i} \left| \det \mathbf{J}_g(\hat{u}; z) \right| + \rho_{U|A}(u \mid a) \frac{\partial}{\partial z_i} \left| \det \mathbf{J}_g(\hat{u}; z) \right|, \tag{95}$$

and write in block form:

$$\begin{bmatrix} \rho_{U|A}(u \mid a) & \left( \nabla_u \rho_{U|A}(u \mid a) \right)^\top \end{bmatrix} \begin{bmatrix} \frac{\partial}{\partial z_i} \left| \det \mathbf{J}_g(\hat{u}; z) \right| \\ \frac{\partial g(\hat{u}; z)}{\partial z_i} \left| \det \mathbf{J}_g(\hat{u}; z) \right| \end{bmatrix} = 0. \tag{96}$$

Stacking the respective equations for $a_1, a_2, \ldots, a_{d+1}$ then yields:

$$M(u; a_1, \ldots, a_{d+1}) \Big|_{u=g(\hat{u}; z)} \begin{bmatrix} \frac{\partial}{\partial z_i} \left| \det \mathbf{J}_g(\hat{u}; z) \right| \\ \frac{\partial g(\hat{u}; z)}{\partial z_i} \left| \det \mathbf{J}_g(\hat{u}; z) \right| \end{bmatrix} = \mathbf{0}, \tag{97}$$

because $M$ must be full rank by *variability* (condition 3) we have that, for all indices $i \in \{1, \ldots, d\}$, $\partial g(\hat{u}; z)/\partial z_i = 0$ and $\partial/\partial z_i \left| \det \mathbf{J}_g(\hat{u}; z) \right| = 0$. Therefore, $g(\hat{u}; z)$ does not depend on $z$, and can be written as $g(\hat{u}; z) = g(\hat{u})$, proving $T_X$ and $\hat{T}_X$ differ only by a bijection of the exogenous noise. Finally, combining the above $\sim_{\mathcal{L}_3}$-equivalence results, there exist bijections $g$ and $h$ such that:

$$\forall u, w, a \; : \; T_X(u; T_Z(w; a)) = \hat{T}_X(g^{-1}(u); \hat{T}_Z(h^{-1}(w); a)), \tag{98}$$

and for the deterministic $Z \mid A$ case we have

$$\forall u, a \; : \; T_X(u; T_Z(a)) = \hat{T}_X(g^{-1}(u); \hat{T}_Z(a)), \tag{99}$$

hence $T$ and $\hat{T}$ produce the same counterfactuals, concluding the proof. $\square$

### B.3.2 Instrumental Variable Setting

Even when an unobserved confounder is present, counterfactual identifiability can still be recovered from purely observational ($\mathcal{L}_1$) data, provided we can locate suitable instrumental variables. Here, an instrumental variable is any variable (or set of variables) that is: (i) statistically independent of the latent noise $U$; and (ii) influences the outcome $X$ exclusively through its effect on its parents $\mathbf{PA}$.

**Lemma B.14** (IV $\mathcal{L}_3$-Equivalence (Nasr-Esfahany et al., 2023))**.** *Let the instrument take values in the finite set $\mathbf{I} = \{i_1, \ldots, i_n\}$ and the endogenous parents of $X$ take values in $\mathbf{PA} = \{\mathbf{pa}_1, \ldots, \mathbf{pa}_n\}$. Two bijective models $T$ and $\hat{T}$ yield the same counterfactuals (i.e. are equivalent in the sense that they coincide up to a reparameterisation of the latent noise), whenever the following conditions hold:*

1. *(Instrumental Variable) $\mathbf{I} \perp\!\!\!\perp U$ and $\mathbf{I} \perp\!\!\!\perp \hat{U}$.*

2. *For all $\mathbf{pa} \in \mathbf{PA}$, the maps $T^{-1}(\cdot\,; \mathbf{pa})$ and $\hat{T}(\cdot\,; \mathbf{pa})$ are strictly monotone (either strictly increasing or strictly decreasing) and twice continuously differentiable.*

3. *The density $\rho_{\hat{U}}(\cdot)$ is continuously differentiable.*

4. *For every $i, \mathbf{pa}$ the joint density $\rho_{\mathcal{D}}(i, \mathbf{pa}, \cdot)$ is continuously differentiable in its latent coordinate.*

5. *(Positivity) $\forall u, \hat{u}$ and every $\mathbf{pa} \in \mathbf{PA}$, $\rho_{U, \mathbf{PA}}(u, \mathbf{pa}) > 0$ and $\rho_{\hat{U}, \mathbf{PA}}(\hat{u}, \mathbf{pa}) > 0$.*

6. *(Variability) For every fixed $u$, the matrix*

$$M_{\mathcal{D}}(u, \mathbf{I}) := \begin{bmatrix} \rho_{\mathcal{D}}(\mathbf{pa}_1 \mid u, i_1) & \cdots & \rho_{\mathcal{D}}(\mathbf{pa}_n \mid u, i_1) \\ \vdots & & \vdots \\ \rho_{\mathcal{D}}(\mathbf{pa}_1 \mid u, i_n) & \cdots & \rho_{\mathcal{D}}(\mathbf{pa}_n \mid u, i_n) \end{bmatrix} \tag{100}$$

*satisfies $|\det M_{\mathcal{D}}(u, \mathbf{I})| \geq c$, for some constant $c > 0$, that does not depend on $u$.*

*Proof.* The proof (for $d = 1$) is given by Nasr-Esfahany et al. (2023). Since $T$ and $\hat{T}$ are assumed to be strictly monotone only in the scalar setting, counterfactual identifiability is established for $d = 1$ only. Theorem B.11 shows (drawing on Brenier (1991)'s Theorem) that if there exists a convex function $\phi : \mathbb{R}^d \to \mathbb{R}$, where $T(u; \mathbf{pa}, i) = \nabla_u \phi(u; \mathbf{pa}, i)$, then $T, \hat{T}$ are strictly monotone (rank-preserving) in $u$ in the multi-dimensional case. Therefore, by Theorem B.11, we now generalise the above IV counterfactual identifiability result for $d > 1$, concluding the proof. $\qquad\square$

### B.3.3 Backdoor Criterion Setting

Another scenario in which counterfactual identification from observational ($\mathcal{L}_1$) data is achievable (under stated assumptions), despite the presence of confounding, is when there exists a set of variables $\mathbf{Z}$ that satisfies the backdoor criterion (BC) w.r.t. the pair $(\mathbf{PA}, X)$, where $\mathbf{PA}$ are $X$'s parents. That is, $\mathbf{Z}$ blocks all backdoor paths between $\mathbf{PA}$ and $X$, where each such path includes an arrow pointing into $\mathbf{PA}$. The role of $\mathbf{Z}$ is to account for all the spurious associations between $\mathbf{PA}$ and $U$.

**Lemma B.15** (BC $\mathcal{L}_3$-Equivalence (Nasr-Esfahany et al., 2023))**.** *Let $T$ and $\hat{T}$ be two distinct bijective models that induce identical joint distributions over the endogenous variables $\{\mathbf{PA}, X\}$. Then $T$ and $\hat{T}$ are counterfactually equivalent ($\sim_{\mathcal{L}_3}$) if the following conditions hold:*

1. *(Backdoor Criterion) $U \perp\!\!\!\perp \mathbf{PA} \mid \mathbf{Z}$ and $\hat{U} \perp\!\!\!\perp \mathbf{PA} \mid \mathbf{Z}$.*

2. *For every $\mathbf{pa}$, the derivatives $\nabla_{\mathbf{pa}} |\det \mathbf{J}_{T^{-1}}(\cdot\,; \mathbf{pa})|$ and $\nabla_{\mathbf{pa}} |\det \mathbf{J}_{\hat{T}}(\cdot\,; \mathbf{pa})|$ both exist.*

3. *(Variability) For every $u$, there exist instances $\mathbf{z}_1, \ldots, \mathbf{z}_{d+1}$ such that*

$$\left| \det M(u, \mathbf{z}_1, \ldots, \mathbf{z}_{d+1}) \right| > 0, \tag{101}$$

*where*

$$M(u, \mathbf{z}_1, \ldots, \mathbf{z}_{d+1}) := \begin{bmatrix} \rho_{U|\mathbf{Z}}(u \mid \mathbf{z}_1) & \nabla_u \rho_{U|\mathbf{Z}}(u \mid \mathbf{z}_1) \\ \vdots & \vdots \\ \rho_{U|\mathbf{Z}}(u \mid \mathbf{z}_{d+1}) & \nabla_u \rho_{U|\mathbf{Z}}(u \mid \mathbf{z}_{d+1}) \end{bmatrix}. \tag{102}$$

*Remark* B.16. The proof for Lemma B.15 is provided by Nasr-Esfahany et al. (2023) and already applies to the multi-dimensional setting ($d \geq 1$), showing that bijectivity of $T, \hat{T}$ is sufficient for counterfactual equivalence when the BC is satisfied and the *variability* assumption holds.

*Intuition.* Because BC makes the change-of-variables identity independent of the parents, differentiating w.r.t. **PA** and stacking across $d+1$ values of **Z** yields a full-rank (variability) linear system whose only solution is zero; hence the link $g(\cdot\,; \mathbf{pa}) := T^{-1}(\cdot\,; \mathbf{pa}) \circ \hat{T}(\cdot\,; \mathbf{pa})$ cannot depend on **pa**, so $g(u; \mathbf{pa}) = g(u)$ and $T$ and $\hat{T}$ are counterfactually equivalent ($\sim_{\mathcal{L}_3}$) in the sense that they produce the same counterfactuals.

## B.4 Assumptions: Summary & Plausibility

We open with Pearl and Bareinboim (2022):

> *"Assumptions are self-destructive in their honesty. The more explicit the assumption, the more criticism it invites. [...] Researchers therefore prefer to declare 'threats' in public and make assumptions in private."*

Our work counters this practice: we intend to make our assumptions and constraints explicit so they can be better understood, challenged, and ultimately relaxed in light of new evidence or insight.

**Markovian SCMs.** The Markovianity assumption is standard in causality literature and is also the default assumption in many causal representation learning frameworks (Pearl, 2009; Hyvärinen et al., 2024). The assumption that there is no unobserved confounding is strong in most real-world scenarios. However, it is widely known that Markovianity alone is insufficient for counterfactual identifiability (Xia et al., 2023; Bareinboim et al., 2022; Nasr-Esfahany and Kiciman, 2023). Thus, further assumptions and/or restrictions on the functional class are necessary, such as monotonicity in the $d = 1$ case (Nasr-Esfahany et al., 2023). This is the gap our work fills, particularly for $d > 1$.

**OT Regularity Assumptions.** We assume that $\dim(X) = \dim(U) \geq 1$ as is commonplace in flow-based generative models. This is straightforward to enforce in practice by embedding lower-dimensional latents using dummy coordinates. For our strictest results, we assume quadratic cost and that the source $P_U$ and target $P_{X|\mathbf{PA}}^{\mathfrak{C}}$ are absolutely continuous w.r.t. Lebesgue measure, with densities $\rho_U$ and $\rho_{X|\mathbf{PA}}^{\mathfrak{C}}$ bounded above and below by positive constants on bounded convex supports.

These are standard assumptions in the literature, particularly in OT, which ensure that the measures $P_U$ and $P_{X|\mathbf{PA}}$ admit well-behaved densities, guaranteeing the existence and uniqueness of an OT map (Brenier, 1991; Caffarelli, 1992). However, they may not always hold, and are often stronger than strictly necessary in practice. In practice, when working with empirical distributions, it is often advantageous to approximate them by smoothing with small continuous uniform or Gaussian noise so that the resulting distribution admits a density. Real-world data are often normalised and then modelled with smooth densities on (possibly truncated) compact supports, making these assumptions a common modelling choice. Optionally, if the densities are $C^{1,\alpha}$ and satisfy standard two-sided bounds on a convex domain, this yields improved regularity, e.g. an interior $C^{2,\alpha}$ Brenier potential.

We require that there exist a velocity field $v_t$ solving $\partial_t \rho_t + \nabla \cdot (\rho_t v_t) = 0$ where $\rho_0 = P_U$, and $\rho_1 = P_{X|\mathbf{pa}}$, with $v_t$ Lipschitz in $x$ and in $L^2([0,1] \times \Omega)$ (Benamou and Brenier, 2000). Neural ODE parameterisations with spectral normalisation or weight clipping satisfy global Lipschitz bounds, and so do classical kernels with bounded derivatives (Chen et al., 2018). Uniqueness of the associated ODE's solution is satisfied by the Lipschitz condition under Picard-Lindelöf's theorem. We note that in practice the time-dependent density $\rho_t$ is often fixed by choice of interpolant (e.g. linear (McCann, 1997)). By further optimising the interpolant, it is possible to solve Eq. (47) under mild assumptions on the Benamou-Brenier density (Albergo and Vanden-Eijnden, 2023).

For our most general result, which generalises scalar quantiles/ranks to $d > 1$, we assume that the prior $P_U$ be a uniform measure, e.g. $\mathcal{U}([0,1]^d)$. This was chosen for PIT-type reasons. However, a uniform on a $d$-dimensional cube is not strictly necessary for rank-preservation; other choices like the standard normal or the uniform on a ball are available provided regularity conditions for OT are met (Carlier et al., 2016). In any case, weaker counterfactual equivalence relations may be obtained by relaxing the conditions under which point-wise identification holds. Such relaxations are

often admissible in practice without completely sacrificing the utility of the method; this is in the spirit of many existing identifiability results (Khemakhem et al., 2021; Nasr-Esfahany et al., 2023; Hyvärinen et al., 2024). Finally, the experiments we ran to validate our theory rely on Batch-OT to recover the global OT map in practice, which Pooladian et al. (2023); Tong et al. (2024) provide convergence proofs for. However, Batch-OT is known to underperform for smaller batch sizes. Therefore, improving high-dimensional OT in terms of accuracy and efficiency remains key.

To summarise: with quadratic cost, (i) finite second moments ensure existence of an optimal coupling; (ii) if the source is absolutely continuous and $\dim(U) = \dim(X)$, the OT is (a.e.) unique and given by Brenier's gradient map; (iii) two-sided density bounds on convex domains are standard, useful regularity assumptions, but are not always strictly necessary in practice; and (iv) a Lipschitz velocity field yields a unique, invertible Benamou-Brenier flow solving the continuity equation. When embedded in a Markovian SCM with independent noise, this flow provides a unique, rank-preserving transport representation of the causal mechanism, generalising the notion of scalar quantiles to $d > 1$, and under our stated assumptions, supporting identifiability of multi-dimensional counterfactuals from observational data.

## C Energy Based Models: Curl-free Flows

**Proposition C.1** (Curl-free Flows.). *Let $(\rho_t, v_t)_{t\in[0,1]}$ be an admissible curve for the dynamic OT map in Lemma B.5, with $v_t \in C^1(\Omega; \mathbb{R}^d)$ for each fixed $\mathbf{pa} \in \mathbb{R}^k$. Write the Helmholtz-Hodge decomposition $v_t = \nabla\psi_t + s_t$, where $s_t$ is the curl component. Since $\mathrm{div}(\rho_t s_t) = 0$ by construction, the pair $(\rho_t, \nabla\psi_t)$ satisfies $\partial_t \rho_t + \mathrm{div}(\rho_t \nabla\psi_t) = 0$. Then, the Benamou-Brenier action $\mathcal{A}$ is:*

$$\mathcal{A}(\rho, v) - \mathcal{A}(\rho, \nabla\psi) = \frac{1}{2}\int_0^1 \int_\Omega \|s_t(u; \mathbf{pa})\|^2 \rho_t(u; \mathbf{pa})\, \mathrm{d}u\, \mathrm{d}t \geq 0, \tag{103}$$

*hence replacing $v_t$ by its curl-free component $\nabla\psi$ never increases the dynamic OT cost.*

*Proof.* For each $t \in [0, 1]$ the Helmholtz-Hodge decomposition provides a unique split

$$v_t = \nabla\psi_t + s_t, \quad \int_\Omega \rho_t(u, \mathbf{pa})\, s_t(u, \mathbf{pa})\cdot\nabla\eta(u, \mathbf{pa})\, \mathrm{d}u = 0, \quad \forall \eta \in H_N^1(\Omega). \tag{104}$$

Orthogonality in $L^2(\rho_t)$ then implies

$$\|v_t\|^2 = \|\nabla\psi_t\|^2 + \|s_t\|^2 \tag{105}$$

hence

$$\mathcal{A}(\rho, v) = \mathcal{A}(\rho, \nabla\psi) + \frac{1}{2}\int_0^1 \int_\Omega \|s_t(u; \mathbf{pa})\|^2 \rho_t(u, \mathbf{pa})\, \mathrm{d}u\, \mathrm{d}t. \tag{106}$$

The integral above is non-negative and vanishes iff $s_t \equiv 0$, and therefore, $\mathcal{A}(\rho, v) \geq \mathcal{A}(\rho, \nabla\psi)$, with strict inequality for every non-conservative field. $\square$

*Remark* C.2. The unique minimiser of the dynamic OT problem is curl-free, as its time-1 map is the gradient of a convex potential. Thus, the above result suggests that an energy-based model (EBM) parameterisation of $v_t$ (e.g. see De Sousa Ribeiro and Glocker (2025); Balcerak et al. (2025)) could prove to be a useful inductive bias for some problems, as it restricts the search space to curl-free vector fields. In practice, we operationalise this model by simply taking the sum of a neural network's output as the energy, then taking gradients w.r.t the input $x$, yielding a curl-free vector field. We use this model class primarily for the constructed counterfactual ellipse scenarios, as it can be quite computationally intensive for high-dimensional images, given our resources. We consider scalable curl-free flow models fertile ground for future work.

# D   Counterfactual Soundness Axioms

When counterfactual ground truth is not available, we follow prior work (Monteiro et al., 2023; Ribeiro et al., 2023) and measure counterfactual *composition*, *effectiveness* and *reversibility*, which are axiomatic soundness properties of counterfactual functions that must hold true in all causal models (Halpern, 1998; Galles and Pearl, 1998; Pearl, 2009). We provide a gentle introduction next for completeness; for a more detailed treatment, please refer to Monteiro et al. (2023).

We point out that, while certainly useful diagnostic tools, the following metrics alone do not imply identification and should not be construed as evidence of causal validity.

1. The **composition** axiom states that intervening on a variable to have a value it would have had without the intervention should leave the system unchanged. In practice, composition is often measured using reconstruction error under a null-intervention on the parents:

$$\mathbb{E}_{(x,\mathbf{pa})\sim P_{\text{data}}}\Big[\, \|x - T(\mathbf{pa}, \mathbf{pa}, x)\|_1 \,\Big]. \tag{107}$$

   We report this metric to enable fair comparisons with prior work. For a more stringent measure of composition that tests path independence of intervention sequences, we suggest:

$$\mathbb{E}_{(x,\mathbf{pa})\sim P_{\text{data}}}\Big[\, \|T(\mathbf{pa}_2^*, \mathbf{pa}_1^*, T(\mathbf{pa}_1^*, \mathbf{pa}, x)) - T(\mathbf{pa}_2^*, \mathbf{pa}, x)\|_1 \,\Big]. \tag{108}$$

   Other *composition* tests could include comparing outcomes of one-by-one sequential interventions against equivalent simultaneous ones.

2. The **effectiveness** axiom states that intervening on a variable to have a specific value will cause the variable to take on that value. Effectiveness is typically the most challenging axiom to obtain an unbiased measure of in practice, as ideally one needs access to an *oracle* to tell us when/if our interventions are faithful. In practice, pseudo-oracles in the form of parent predictor models are used to determine whether interventions are successful:

$$\mathbb{E}_{(x,\mathbf{pa})\sim P_{\text{data}}}\Big[\mathrm{d}(\mathcal{O}(T(\mathbf{pa}^*, \mathbf{pa}, x)), \mathbf{pa}^*)\Big], \tag{109}$$

   where $\mathcal{O}(\cdot)$ is a pseudo-oracle function that returns the value of the parent(s) given the observation. Here $\mathrm{d}(\cdot)$ is an appropriate distance function dependent on the parent type. It is common practice to also apply the same metric to variables that are *not* descendants of the intervened-on variable, as invariance there serves as a diagnostic measure of *minimality*.

3. The **reversibility** axiom precludes multiple solutions due to feedback loops, and follows directly from composition in recursive systems such as DAGs. This can be intuitively understood as measuring cycle consistency w.r.t. interventions on the parents:

$$\mathbb{E}_{(x,\mathbf{pa})\sim P_{\text{data}}}\Big[\, \|x - T(\mathbf{pa}, \mathbf{pa}^*, T(\mathbf{pa}^*, \mathbf{pa}, x))\|_1 \,\Big]. \tag{110}$$

   Using less cluttered notation, we measure the extent to which the reversal $x_r$ matches the observation $x$ after $n \geq 1$ intervention cycles, where each cycle is given by:

$$x^* = T_{\mathbf{pa}^*} \circ T_{\mathbf{pa}}^{-1}(x), \qquad x_r = T_{\mathbf{pa}} \circ T_{\mathbf{pa}^*}^{-1}(x^*). \tag{111}$$

Although often too costly in practice, recursive measurement of all the above metrics can yield more fine-grained insight into the faithfulness of interventions, biases of the associated counterfactual function, and incremental loss of identity w.r.t. the original observation (Monteiro et al., 2023).

# E   Counterfactual Ellipse: Extra Results

## E.1   Dataset Details

As outlined in the main text, our ellipse dataset is built on Nasr-Esfahany et al. (2023)'s setup. We reiterate here for completeness. Let $U \in \mathbb{R}^2$ be the semi-major and -minor parameters of an ellipse, PA $\in (0, 2\pi)$ be an angle specifying a single point on the ellipse, and $X \in \mathbb{R}^2$ be its cartesian coordinates. The data-generating process is defined as follows:

$$Z := \epsilon_z, \qquad\qquad \epsilon_z \sim \text{Uniform}(-0.5,\, 0.5)$$
$$\text{PA} := (1.44254843z + 0.59701923 + \epsilon_{\text{pa}}) \bmod (2\pi), \qquad\qquad \epsilon_{\text{pa}} \sim \mathcal{N}(0,1)$$
$$U_0 := \exp(1.64985274z + 0.2656131) + \epsilon_{u_0} \qquad\qquad \epsilon_{u_0} \sim \text{Beta}(1,1)$$
$$U_1 := U_0(1 + \epsilon_{u_1}\exp(1.61323358z - 0.18070237)) \qquad\qquad \epsilon_{u_1} \sim \text{Exponential}(1)$$
$$X_0 := U_0(2 + \sin(\text{PA}))$$
$$X_1 := U_1(2 + \cos(\text{PA})).$$

By construction, $U \perp\!\!\!\perp \text{PA} \mid Z$, and $Z$ satisfies the backdoor criterion (BC) w.r.t. PA $\rightarrow X$. To induce a Markovian setting, we simply randomise PA, yielding marginal independence $U \perp\!\!\!\perp \text{PA}$.

For the front-door criterion (FC) to apply, we modify the data-generating process slightly to include a mediator variable $M$, such that PA $\rightarrow M \rightarrow X$:

$$Z := \epsilon_z, \qquad\qquad \epsilon_z \sim \text{Uniform}(-0.5,\, 0.5)$$
$$\text{PA} := (1.44254843z + 0.59701923 + \epsilon_{\text{pa}}) \bmod (2\pi), \qquad\qquad \epsilon_{\text{pa}} \sim \mathcal{N}(0,1)$$
$$M_0 := \beta \cdot (\sin(\text{PA}) + \epsilon_{m_0}) \qquad\qquad \epsilon_{m_0} \sim \mathcal{N}(0, 0.01)$$
$$M_1 := \beta \cdot (\cos(\text{PA}) + \epsilon_{m_1}) \qquad\qquad \epsilon_{m_1} \sim \mathcal{N}(0, 0.01)$$
$$U_0 := \exp(1.64985274z^2 + 0.2656131) + \epsilon_{u_0} \qquad\qquad \epsilon_{u_0} \sim \text{Beta}(1,1)$$
$$U_1 := U_0(1 + \epsilon_{u_1}\exp(1.61323358z^2 - 0.18070237)) \qquad\qquad \epsilon_{u_1} \sim \text{Exponential}(1)$$
$$X_0 := U_0(2 + M_0)$$
$$X_1 := U_1(2 + M_1),$$

where $\beta = 1/\sqrt{m_0^2 + m_1^2}$ projects the noised points back onto the unit circle. Note the non-linear confounding PA $\leftarrow Z \rightarrow U \rightarrow X$. This dataset is only used for the front-door criterion experiments, where we intentionally omit the unobserved confounder $Z$ when building the model.

In all cases, we split our datasets into 70/10/20% for training, validation and testing, respectively.

## E.2   Architecture and Experimental Setup

For all our model variants, we used the following simple residual multi-layer perceptron (MLP) to parameterise the (conditional) vector field. We start with a linear projection layer which takes as input the concatenation of $x$, pa and a time index $t \in [0,1]$. The input dimension is therefore $\dim(X) + \dim(\text{PA}) + 1 = 4$. We then have 3 of the following (residual) blocks:

$$x \mapsto \text{LINEAR} \circ \text{SILU} \circ \text{LINEAR} \circ \text{LAYERNORM}(x). \tag{112}$$

The hidden dimension of these blocks is 256. The head of the MLP consists of a final layer norm and a linear projection to the output dimension $\dim(X) = 2$. The energy-based flow model variants were built by taking the mean of the MLP output, then taking gradients w.r.t. the input $x$.

We used PyTorch (Paszke et al., 2019) to train all our modes for 500k steps, under identical hyper-parameter setups. These were determined with a light sweep over the learning rate and network width. We used the AdamW optimizer with a learning rate of $10^{-4}$, weight decay of $10^{-4}$, $\beta_1 = 0.9$, $\beta_1 = 0.999$, $\epsilon = 10^{-8}$, and batch size 256. The best model was selected based on the counterfactual error ($\mu_{\text{APE}}$) on the validation set during training. Final performance is reported on the test set.

## E.3 Qualitative Results

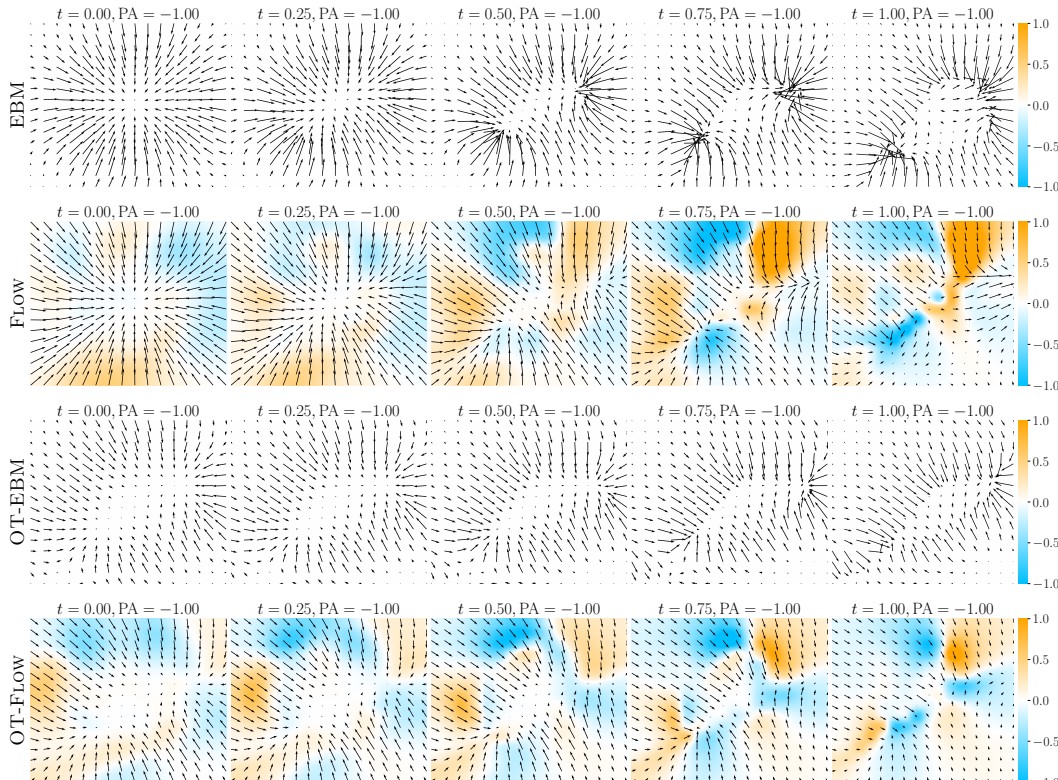

Figure 5: Visualising the curl ($\nabla \times v_t$) of the learned vector field of different models over time (scaled to $[-1, 1]$), for a given intervention on the parents PA $= -1$. We can see that the EBM variants are curl-free (irrotational) by design, as the vector field is given by the gradient of a scalar potential. We also see that the OT vector fields are smoother and OT-FLOW exhibits milder 'rotations' compared to FLOW. This is consistent with convergence to the Brenier map, which is itself curl-free.

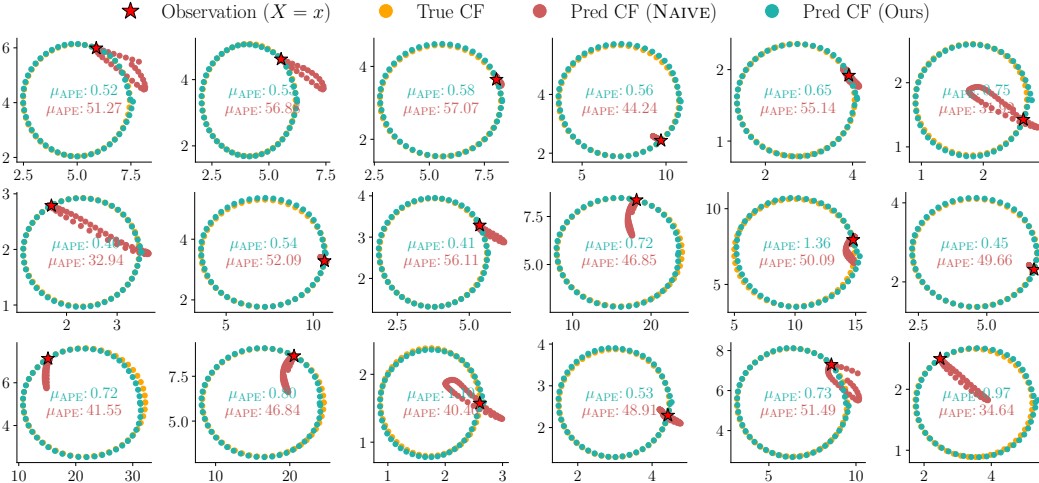

Figure 6: Inferred counterfactual (CF) ellipses using our OT coupling flow vs. the naive OT coupling approach explained in Section 5. Not cherry-picked.

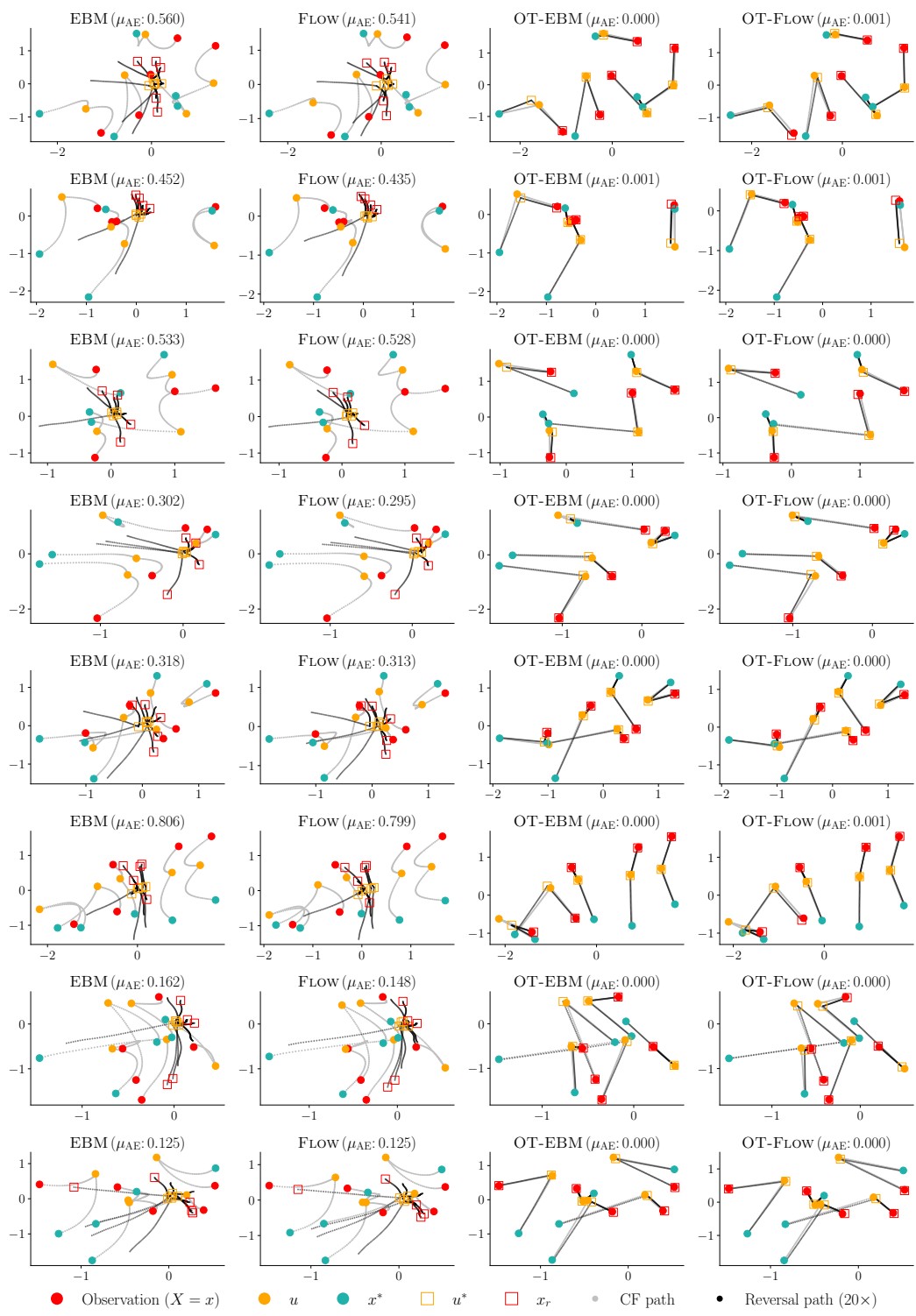

Figure 7: Extra counterfactual *reversibility* results (not cherry-picked). Our OT coupling flow exhibits near-perfect counterfactual reversibility, satisfying the soundness axiom (Monteiro et al., 2023). Conversely, standard flows (and diffusion models, which are equivalent (Gao et al., 2025)) exhibit comparatively poor reversibility upon repeated cycles, with $u^*$ often collapsing to a single point for any initial observation $x$. In simple terms, straight paths are reversible; therefore, one ought to learn straight paths induced by OT in order to satisfy axiomatic reversibility of counterfactual functions.

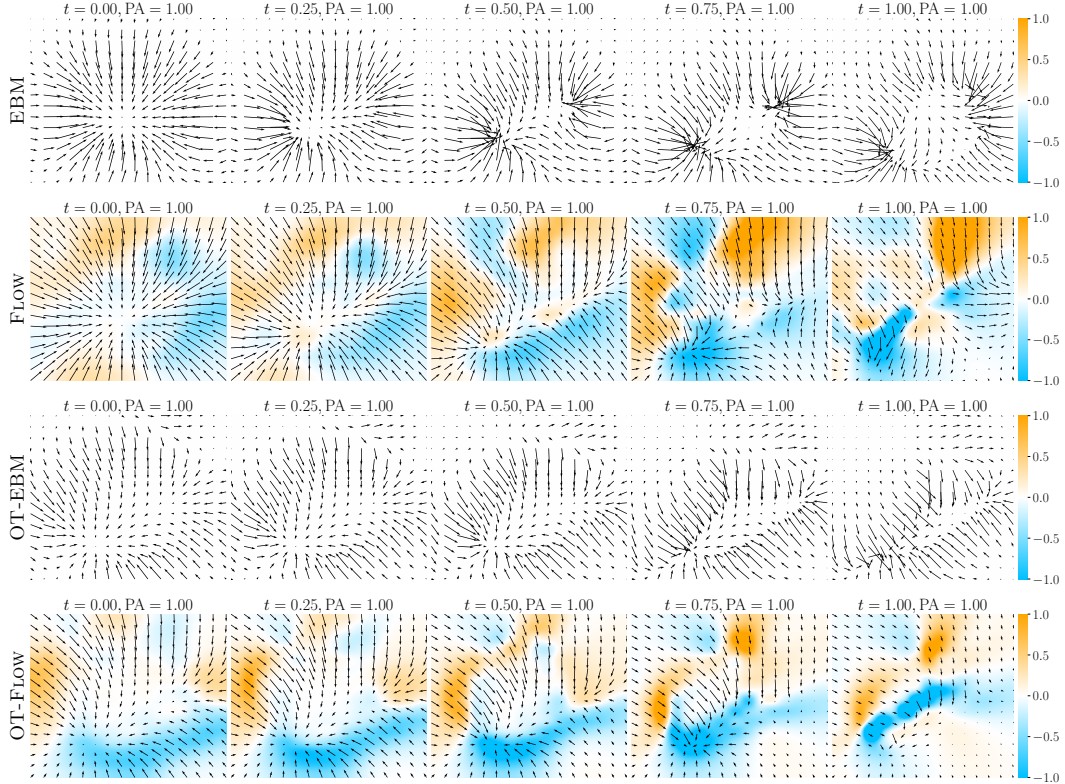

Figure 8: Visualising the curl ($\nabla \times v_t$) of the learned vector field of different models over time (scaled to $[-1, 1]$), for a given intervention on the parents $PA = 1$. We can see that the EBM variants are curl-free (irrotational) by design, as the vector field is given by the gradient of a scalar potential. We also see that the OT vector fields are smoother and OT-FLOW exhibits milder 'rotations' compared to FLOW. This is consistent with convergence to the Brenier map, which is itself curl-free.

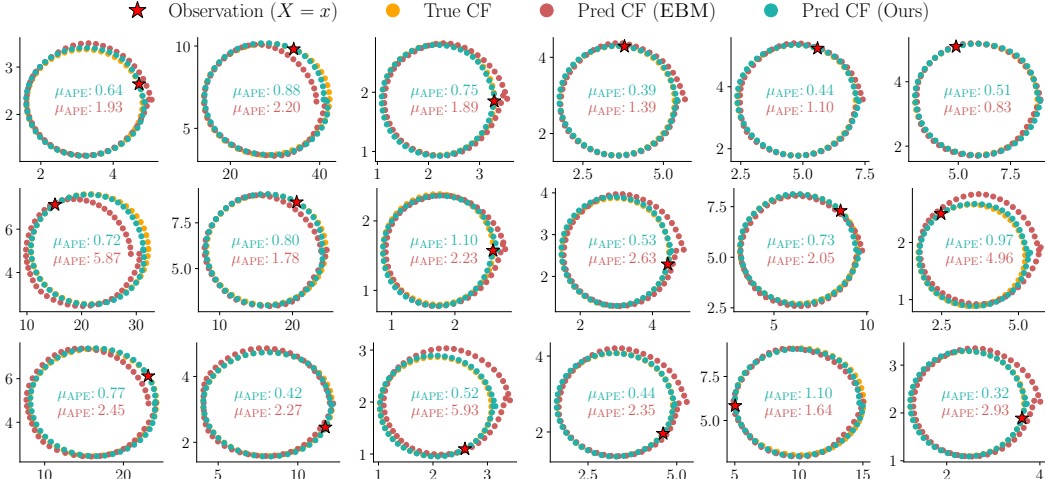

Figure 9: Inferred counterfactual (CF) ellipses using our (Markovian) OT coupling flow vs. an energy-based (curl-free) parameterisation of a flow-based model. Not cherry-picked.

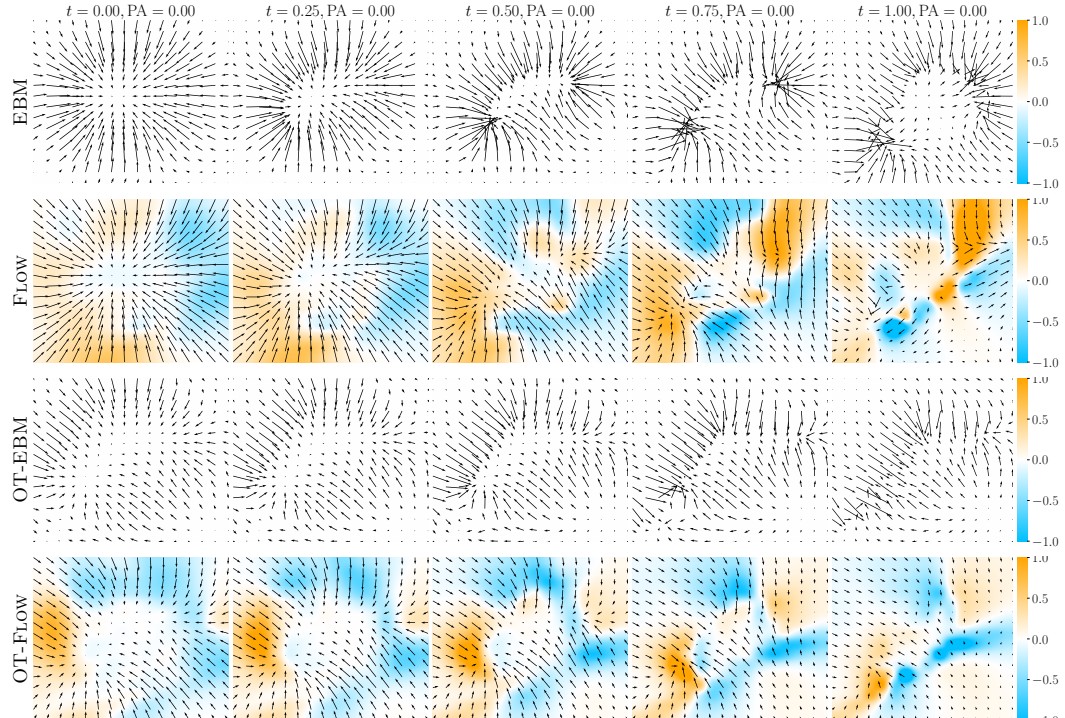

Figure 10: Visualising the *curl* ($\nabla \times v_t$) of the learned vector field of different models over time (scaled to $[-1, 1]$), for a given intervention on the parents PA $= -1$. We can see that the EBM variants are curl-free (irrotational) by design, as the vector field is given by the gradient of a scalar potential. We also see that the OT vector fields are smoother and OT-FLOW exhibits milder 'rotations' compared to FLOW. This is consistent with convergence to the Brenier map, which is itself curl-free.

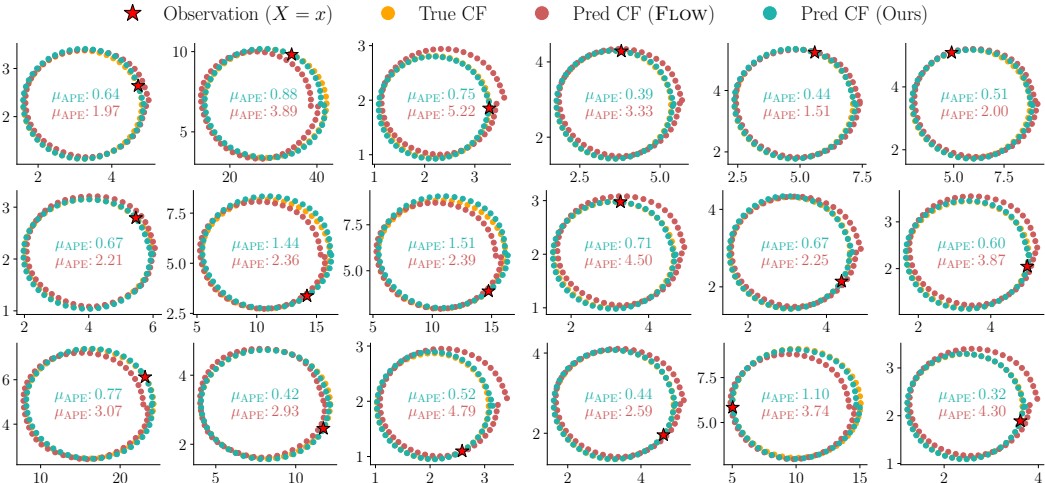

Figure 11: Inferred counterfactual (CF) ellipses using our (Markovian) OT coupling flow vs. a continuous-time flow (Lipman et al., 2023). We note that a standard continuous-time flow is equivalent to a diffusional model (Gao et al., 2025); thus, this baseline is also representative of Sanchez and Tsaftaris (2021)'s diffusion-based counterfactual inference model. Not cherry-picked.

## E.4 Ablation Studies: Priors & Assumption Violations

Table 3: Ablation of complex data-generating processes (DGP) on CF error $\mu_{\text{APE}}$ (%) $\downarrow$.

| | Counterfactual Ellipse | | | |
|---|---|---|---|---|
| DGP $P_U$ | EBM | FLOW | OT-EBM | OT-FLOW |
| Original | $2.32_{\pm.01}$ | $2.30_{\pm.02}$ | $0.93_{\pm.02}$ | $0.76_{\pm.01}$ |
| Bimodal | $5.11_{\pm.03}$ | $5.01_{\pm.01}$ | $1.73_{\pm.09}$ | $1.49_{\pm.06}$ |
| Multimodal | $7.94_{\pm.26}$ | $7.69_{\pm.20}$ | $2.29_{\pm.02}$ | $2.03_{\pm.03}$ |

**More Complex $P_U$.** Intuitively, a more complex data-generating process is expected to be harder to model. To test this hypothesis, we created new datasets with multimodal $P_U$. We use an indicator $s_i \sim \text{Bernoulli}(0.5)$ to induce a non-linear, bimodal shift into $\mu_i \in \{-2, 2\}$ into $U$'s mechanism. We also use a categorical shift $\mu_i \in \{-4, -2, 2, 4\}$, with respective shift probabilities $(0.3, 0.2, 0.2, 0.3)$, to generate another multimodal setting. As reported in Table 3, we confirm that more complex $P_U$'s can affect generation, but importantly, the relative rank of our models remains consistent. Naturally, we expect larger models to reduce performance gaps for complicated $P_U$'s.

Table 4: Ablation of monotonicity violation.

| | | Counterfactual Ellipse | | |
|---|---|---|---|---|
| MODEL | Monotone | COMPOSITION $\downarrow$ | REVERSIBILITY $\downarrow$ | $\mu_{\text{APE}}$ (%) $\downarrow$ |
| Nasr-Esfahany et al. (2023) | N | - | - | $607_{\pm\text{N/A}}$ |
| EBM | N | $27.86_{\pm.369}$ | $39.15_{\pm.349}$ | $2.319_{\pm.006}$ |
| FLOW | N | $27.99_{\pm.096}$ | $39.39_{\pm.229}$ | $2.295_{\pm.020}$ |
| OT-EBM | Y | $0.659_{\pm.020}$ | $1.056_{\pm.072}$ | $0.925_{\pm.019}$ |
| OT-FLOW | Y | $0.618_{\pm.046}$ | $1.016_{\pm.090}$ | $0.763_{\pm.014}$ |

**Monotonicity Violation.** We ran a new set of experiments to evaluate different methods under a monotonicity violation. In addition to measuring counterfactual error, we inspect *composition* and *reversibility* counterfactual soundness axioms, in order to reveal possible performance disparities. For evaluation, we use 50/250 steps for ODE/SDE solving, and 20 cycles for composition and reversibility measurements. As reported in Table 4, models which are not 'vector-monotone' exhibit substantially inferior counterfactual soundness. The results confirm the intuition that straighter paths lead to improved reversal capabilities. That said, whether this generalises to much higher dimensions is entirely dependent on the quality of the dynamic OT approximation being used.

Table 5: Ablation of bijectivity violation.

| | | Counterfactual Ellipse | | |
|---|---|---|---|---|
| MODEL | Bijective | COMPOSITION $\downarrow$ | REVERSIBILITY $\downarrow$ | $\mu_{\text{APE}}$ (%) $\downarrow$ |
| SDE Abduction | N | $2.950_{\pm.013}$ | $4.173_{\pm.019}$ | $3.064_{\pm.011}$ |
| SDE Prediction | N | $3.006_{\pm.015}$ | $4.276_{\pm.019}$ | $3.126_{\pm.007}$ |
| CFG ($w = 1.05$) | N | $0.032_{\pm.004}$ | $0.487_{\pm.010}$ | $2.174_{\pm.070}$ |
| CFG ($w = 1.15$) | N | $0.032_{\pm.004}$ | $1.499_{\pm.013}$ | $6.081_{\pm.069}$ |
| CFG ($w = 1.25$) | N | $0.032_{\pm.004}$ | $2.598_{\pm.024}$ | $9.809_{\pm.078}$ |
| OT-FLOW | Y | $0.032_{\pm.004}$ | $0.056_{\pm.009}$ | $0.763_{\pm.014}$ |

**Bijectivity Violation.** In this experiment, we use three strategies to stress-test bijectivity violations: (i) stochastic *abduction* by solving an SDE instead of an ODE; (ii) stochastic *prediction* by solving an SDE; (iii) classifier-free guidance (CFG) (Ho and Salimans, 2022). One cycle was used for measuring composition and reversibility. As reported in Table 5, bijectivity violations affect reversibility and composition substantially; this is likely due to the ground-truth mechanism being bijective.

Table 6: Ablation of Markovianity violation. The NAIVE OT coupling baseline corresponds to Batch-OT flow matching (Pooladian et al., 2023; Tong et al., 2024).

| | | Counterfactual Ellipse | | |
| OT Coupling | Markovian | COMPOSITION $\downarrow$ | REVERSIBILITY $\downarrow$ | $\mu_{\text{APE}}$ (%) $\downarrow$ |
| --- | --- | --- | --- | --- |
| NAIVE | N | $0.063_{\pm.005}$ | $0.346_{\pm.012}$ | $47.01_{\pm.016}$ |
| Ours | Y | $0.032_{\pm.004}$ | $0.056_{\pm.009}$ | $0.763_{\pm.015}$ |

**Markovianity Violation.** In this experiment, we demonstrate the importance of using the correct model specification that reflects our causal assumptions. As explained in the main text, the naive Batch-OT coupling violates the Markovianity assumption, so it should not be used for counterfactual inference as if the assumption were to hold. As reported in Table 6, we observe that composition and reversibility are reasonable for the NAIVE OT coupling version, as the map remains bijective. However, the counterfactual error is substantially higher for the NAIVE version compared to our proposed Markovian OT coupling, since the Markovianity violation leads to inconsistent counterfactuals.

# F MIMIC Chest X-ray: Extra Results

## F.1 Dataset Details

We reproduce the dataset setup used by our baselines, i.e. Ribeiro et al. (2023) and Xia et al. (2024). The chest X-ray images were resized to $192 \times 192$ resolution. From the associated metadata, we focused on four key attributes: SEX ($S$), RACE ($R$), AGE ($A$) and DISEASE ($D$). The assumed causal graph follows prior work (Ribeiro et al., 2023) in that $A \rightarrow D$ and $\mathbf{PA} = \{S, R, A, D\}$ are the causal parents of the chest X-ray image $X$. We also focus our scope to only pleural effusion for the disease, to keep comparisons fair. As a result, the final dataset includes only individuals who were either diagnoses with pleural effusion (diseased) or reported as having no findings (i.e. healthy). We then divided the dataset into 62,336 subjects for training, 9,968 for validation and 30,535 for testing, again following the exact same protocol as both Ribeiro et al. (2023) and Xia et al. (2024).

## F.2 Architecture and Experimental Setup

The causal mechanisms for all variables $\mathbf{PA} = \{S, R, A, D\}$ and $X$ were learned from observed data. For $\mathbf{PA}$, standard discrete-time flow-based modelling was used following Pawlowski et al. (2020); Ribeiro et al. (2023). For $X$, we use a scaled-up version of our most successful model analysed in the counterfactual ellipse experiments, namely OT-FLOW, which includes our specialised family of batch TO couplings to satisfy the Markovianity requirement (cf. Section 5). With that said, we recognise that using large batch sizes is best for accurate batch OT approximations in high dimensions (Klein et al., 2025). Given current resource constraints, we encourage future work to explore more accurate and efficient OT approximations at scale, possibly along the lines of Mousavi-Hosseini et al. (2025).

All models are trained in continuous time using our Markovian Batch-OT coupling. Since the AGE ($A$) attribute is continuous, we use an age binning strategy to ensure we sample from the conditional distribution $P^{\mathfrak{c}}_{X|\mathbf{PA}}$ as explained in Section 5. The effect is that each sample in each batch has the same parents, except for age, which may deviate slightly depending on the bin size. Table 12 reports an ablation study on the age bin size; we find negligible performance differences across bin sizes.

To parameterise the (conditional) vector field for all our models and ablation study variants, we used the same streamlined version of the UNet proposed by Dhariwal and Nichol (2021). To condition the model on the parent variables, we simply learn a separate embedding vector for each parent and sum them all together. This joint parent embedding is then combined with the time embedding to condition each block in the UNet, following Dhariwal and Nichol (2021). The hyperparameters are given in Table 7. The model has just 22M trainable parameters. For training, we used the AdamW optimiser with a learning rate of $10^{-4}$, weight decay of $10^{-4}$, $\beta_1 = 0.9$, $\beta_1 = 0.999$, $\epsilon = 10^{-8}$, and a batch size of 64. We use a linear learning rate warmup of 2000 steps. No extensive hyperparameter sweep was necessary to obtain sufficiently good performance. We trained our models for a maximum of 300 epochs. Since our evaluation is focused on the counterfactual soundness axioms, which are

quite costly to evaluate, we instead performed model selection using the validation set loss achieved by an exponential moving average (EMA) of the model parameters (EMA rate of 0.9999). Further improvements in performance are expected from using standard sample quality metrics for model selection. We conducted three runs with different random seeds. All our models were trained on L40 GPUs, with the full model fitting entirely on a single GPU.

### F.3   Additional Comparisons and Ablation Studies

Tables 8 and 9 report additional comparative results against the baselines (Ribeiro et al., 2023; Xia et al., 2024). We again find that our method is superior for all three measured counterfactual soundness axioms, often by large margins, without requiring any costly counterfactual fine-tuning or classifier(-free) guidance strategies. We also conduct ablation studies on our proposed modifications, and the results are reported in Tables 10, 11 and 12. In summary, we observe significant improvements from using our OT coupling compared to the NAIVE non-Markovian approach. As outlined in the main text, we find different performance trade-offs between OT-FLOW and FLOW for different interventions, suggesting either Markovianity violation or a subpar OT approximation (due to small batch size of 64). We expect further improvements from increasing model capacity and batch size significantly (Klein et al., 2025), to better approximate the global OT map (Pooladian et al., 2023).

Table 7: UNet hyperparameters.

| **MIMIC Chest X-ray** | |
|---|---|
| image shape | (1, 192, 192) |
| model channels | 32 |
| channel mult | [1, 2, 4, 6, 8] |
| residual blocks | 2 |
| attention resolutions | [-1] |
| heads | 1 |
| head channels | 64 |
| dropout | 0.1 |

Table 8: Measuring counterfactual *effectiveness* on MIMIC Chest X-ray. $|\Delta_{\mathrm{AUC}}|$ denotes the absolute difference in ROCAUC of inferred counterfactuals relative to the observed data baseline. For each variable, our results (blue shade) appear on the right, and baseline results are on the left. Unlike the baselines, which rely on a costly fine-tuning stage with classifiers/regressors for each variable, our approach does not require any form of classifier or classifier-free guidance to perform well.

| | **MIMIC Chest X-ray** $(192\times192)$ | | | | | | | |
|---|---|---|---|---|---|---|---|---|
| | SEX $(S)$ | | RACE $(R)$ | | AGE $(A)$ | | DISEASE $(D)$ | |
| BASELINE | AUC (%) ↑ | | AUC (%) ↑ | | MAE (yr) ↓ | | AUC (%) ↑ | |
| Observed data | 99.63 | | 95.34 | | 6.197 | | 94.41 | |
| | *(w/o CF fine-tuning Ribeiro et al. (2023))* | | | | | | | |
| INTERVENTION | $|\Delta_{\mathrm{AUC}}|$ (%) ↓ | | $|\Delta_{\mathrm{AUC}}|$ (%) ↓ | | $\Delta_{\mathrm{MAE}}$ (yr) ↓ | | $|\Delta_{\mathrm{AUC}}|$ (%) ↓ | |
| $do(S=s)$ | 7.430 | **0.173**±.02 | 20.84 | **0.583**±.15 | 0.486 | **0.333**±.06 | 1.310 | **0.023**±.05 |
| $do(R=r)$ | **0.130** | 0.180±.01 | 36.94 | **0.050**±.07 | **0.229** | 0.394±.10 | 7.810 | **0.310**±.19 |
| $do(A=a)$ | **0.130** | 0.187±.03 | 20.44 | **1.197**±.15 | 3.872 | **0.836**±.08 | 6.110 | **0.347**±.09 |
| $do(D=d)$ | **0.030** | 0.067±.00 | 20.14 | **0.627**±.18 | 0.560 | **0.435**±.05 | 22.01 | **2.280**±.37 |
| $do(\mathtt{rand})$ | 1.330 | **0.150**±.02 | 24.14 | **0.730**±.22 | 1.166 | **0.510**±.05 | 7.010 | **0.640**±.17 |
| | *(Soft CF fine-tuning Xia et al. (2024))* | | | | | | | |
| INTERVENTION | $|\Delta_{\mathrm{AUC}}|$ (%) ↓ | | $|\Delta_{\mathrm{AUC}}|$ (%) ↓ | | $\Delta_{\mathrm{MAE}}$ (yr) ↓ | | $|\Delta_{\mathrm{AUC}}|$ (%) ↓ | |
| $do(S=s)$ | **0.070** | 0.173±.02 | 2.040 | **0.583**±.15 | - | 0.333±.06 | 0.090 | **0.023**±.05 |
| $do(R=r)$ | **0.130** | 0.180±.01 | 3.360 | **0.050**±.07 | - | 0.394±.10 | **0.110** | 0.310±.19 |
| $do(D=d)$ | **0.030** | 0.067±.00 | 1.540 | **0.627**±.18 | - | 0.435±.05 | 3.690 | **2.280**±.37 |

Table 9: Measuring counterfactual *reversibility* and *composition* properties (Monteiro et al., 2023). The baseline results are reproduced exactly according to Ribeiro et al. (2023). We use an Euler ODE solver or dropi5 (1e-5 tol). A single reverse cycle was used for computing *reversibility*. Three randomly seeded subsets of 1000 test samples were used, and the results were averaged. Similar initial performance was observed for FLOW, thus improving OT further is expected to boost performance.

| METHOD | NFE | MIMIC Chest X-ray (192×192) | |
| | | COMPOSITION MAE (px) ↓ | REVERSIBILITY MAE (px) ↓ |
| --- | --- | --- | --- |
| HVAE (Ribeiro et al., 2023) | N/A | 3.09543 ± 0.0536 | 2.89816 ± 0.4153 |
| OT-FLOW (Ours) | 50 | 2.70379 ± 0.1151 | 20.6228 ± 0.6492 |
| | 250 | 0.55289 ± 0.0280 | 1.22693 ± 0.1630 |
| | dopri5 | **0.18352** ± 0.0276 | **0.49485** ± 0.1549 |

Table 10: Ablation study of our conditional OT coupling on counterfactual *effectiveness*. The baseline is the standard OT coupling (NAIVE) described in Section 5, which violates the Markovian requirement. The results show our approach boosts counterfactual effectiveness, especially for disease and race interventions. The most notable improvements are highlighted using shaded blue areas.

| | | MIMIC Chest X-ray (192×192) | | | |
| INTERVENTION | OT Coupling | SEX ($S$) AUC (%) ↑ | RACE ($R$) AUC (%) ↑ | AGE ($A$) MAE (yr) ↓ | DISEASE ($D$) AUC (%) ↑ |
| --- | --- | --- | --- | --- | --- |
| N/A (Observed data) | N/A | 99.63 | 95.34 | 6.197 | 94.41 |
| $do(S = s)$ | NAIVE | 99.42 | 93.85 | 6.602 | 94.47 |
| | Ours | 99.45±.02 | 94.76±.15 | 6.529±.06 | 94.39±.05 |
| $do(R = r)$ | NAIVE | 99.56 | 88.03 | 6.482 | 94.24 |
| | Ours | 99.45±.06 | 95.29±.73 | 6.590±.10 | 94.10±.19 |
| $do(A = a)$ | NAIVE | 99.53 | 93.95 | 7.723 | 93.88 |
| | Ours | 99.44±.03 | 94.14±.16 | 7.032±.07 | 94.76±.09 |
| $do(D = d)$ | NAIVE | 99.58 | 93.66 | 6.630 | 88.35 |
| | Ours | 99.53±.01 | 94.71±.19 | 6.632±.05 | 92.13±.37 |
| $do(\mathtt{rand})$ | NAIVE | 99.55 | 92.75 | 6.844 | 92.62 |
| | Ours | 99.48±.02 | 94.61±.23 | 6.706±.04 | 93.77±.17 |

Table 11: Ablation study of Batch-OT on counterfactual *effectiveness*. We observe different performance trade-offs for different models; e.g., the OT-FLOW performs better for race interventions while performing worse than FLOW for disease, suggesting possible non-Markovian interactions. We note that a batch size of 64 was used for training due to our current GPU memory constraints. Larger performance differences are expected with larger batch sizes (Klein et al., 2025), or by adopting more accurate and scalable OT approximations (Mousavi-Hosseini et al., 2025).

| | | **MIMIC Chest X-ray** (192×192) | | | |
|---|---|---|---|---|---|
| | | SEX $(S)$ | RACE $(R)$ | AGE $(A)$ | DISEASE $(D)$ |
| INTERVENTION | OT | AUC (%) ↑ | AUC (%) ↑ | MAE (yr) ↓ | AUC (%) ↑ |
| N/A (Observed data) | N/A | 99.63 | 95.34 | 6.197 | 94.41 |
| $do(S = s)$ | N | 99.58 | 94.64 | 6.495 | 94.72 |
| | Y | 99.45±.02 | 94.76±.15 | 6.529±.06 | 94.39±.05 |
| $do(R = r)$ | N | 99.56 | 93.09 | 6.460 | 94.59 |
| | Y | 99.45±.06 | 95.29±.73 | 6.590±.10 | 94.10±.19 |
| $do(A = a)$ | N | 99.54 | 94.40 | 6.855 | 95.23 |
| | Y | 99.44±.03 | 94.14±.16 | 7.032±.07 | 94.76±.09 |
| $do(D = d)$ | N | 99.56 | 94.26 | 6.589 | 92.95 |
| | Y | 99.53±.01 | 94.71±.19 | 6.632±.05 | 92.13±.37 |
| $do(\mathtt{rand})$ | N | 99.56 | 94.17 | 6.596 | 94.48 |
| | Y | 99.48±.02 | 94.61±.23 | 6.706±.04 | 93.77±.17 |

Table 12: Ablation study of AGE $(A)$ bin widths on counterfactual *effectiveness*. We observe minimal differences in performance across different bin sizes. This is encouraging as it demonstrates our OT coupling strategy can still work well for continuous parent variables.

| | | **MIMIC Chest X-ray** (192×192) | | | |
|---|---|---|---|---|---|
| | | SEX $(S)$ | RACE $(R)$ | AGE $(A)$ | DISEASE $(D)$ |
| INTERVENTION | Bin | AUC (%) ↑ | AUC (%) ↑ | MAE (yr) ↓ | AUC (%) ↑ |
| N/A (Observed data) | N/A | 99.63 | 95.34 | 6.197 | 94.41 |
| $do(S = s)$ | 2 | 99.48±.10 | 94.49±.34 | 6.580±.02 | 94.37±.21 |
| | 3 | 99.45±.02 | 94.76±.15 | 6.529±.06 | 94.39±.05 |
| | 5 | 99.41±.02 | 94.85±.16 | 6.600±.04 | 94.44±.05 |
| $do(R = r)$ | 2 | 99.50±.05 | 94.49±1.1 | 6.605±.05 | 94.07±.18 |
| | 3 | 99.45±.06 | 95.29±.73 | 6.590±.10 | 94.10±.19 |
| | 5 | 99.43±.02 | 94.97±.33 | 6.625±.02 | 94.03±.09 |
| $do(A = a)$ | 2 | 99.50±.03 | 94.06±.09 | 7.137±.06 | 94.67±.35 |
| | 3 | 99.44±.03 | 94.14±.16 | 7.032±.07 | 94.76±.09 |
| | 5 | 99.43±.06 | 93.89±.14 | 7.159±.05 | 94.62±.02 |
| $do(D = d)$ | 2 | 99.56±.02 | 94.44±.11 | 6.675±.89 | 91.72±.54 |
| | 3 | 99.53±.01 | 94.71±.19 | 6.632±.05 | 92.13±.37 |
| | 5 | 99.52±.02 | 94.70±.23 | 6.676±.03 | 91.66±.02 |
| $do(\mathtt{rand})$ | 2 | 99.49±.03 | 94.35±.48 | 6.755±.03 | 93.77±.24 |
| | 3 | 99.48±.02 | 94.61±.23 | 6.706±.04 | 93.77±.17 |
| | 5 | 99.46±.03 | 94.53±.19 | 6.760±.03 | 93.66±.15 |

## F.4 Qualitative Results

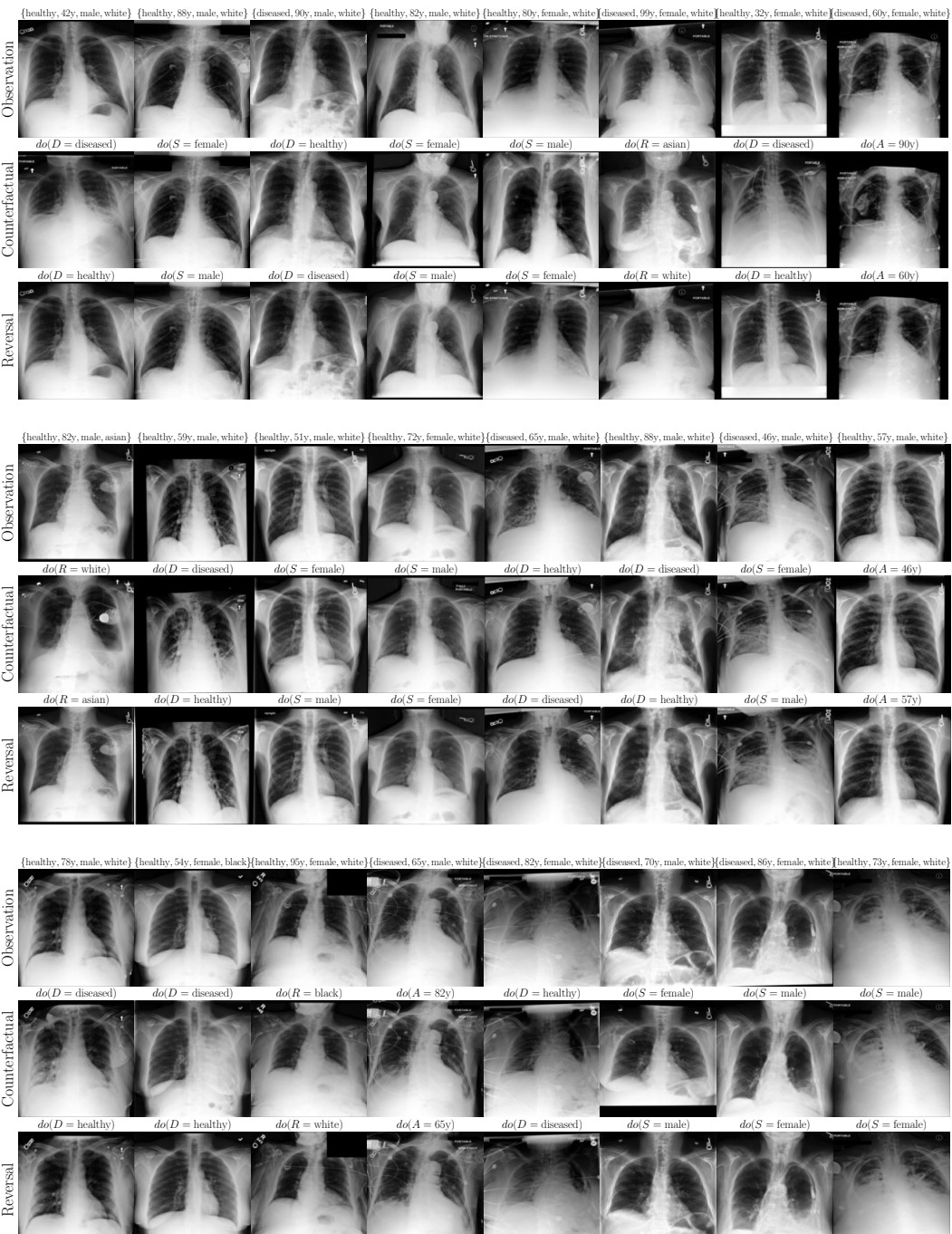

Figure 12: Qualitative counterfactual inference and *reversibility* results (500 ODE Euler solver steps) using our Markovian OT-FLOW model. Despite the small model size, we observe faithful, reasonably identity-preserving interventions, as well as impressive counterfactual reversibility. Importantly, no costly counterfactual fine-tuning (Ribeiro et al., 2023), or classifier(-free) strategies were required.

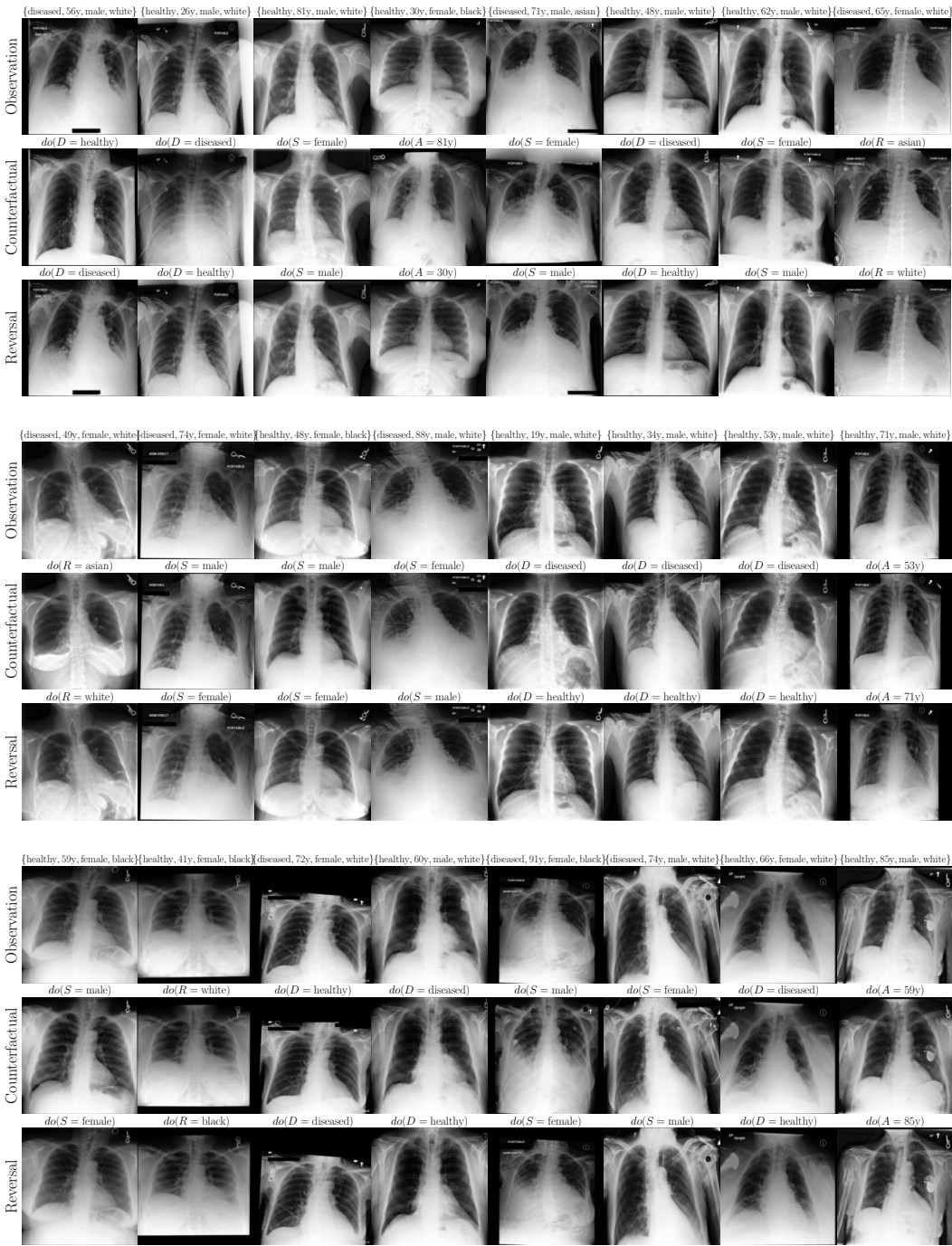

Figure 13: Qualitative counterfactual inference and *reversibility* results (500 ODE Euler solver steps) using our Markovian OT-FLOW model. Despite the small model size, we observe faithful, reasonably identity-preserving interventions, as well as impressive counterfactual reversibility. Importantly, no costly counterfactual fine-tuning (Ribeiro et al., 2023), or classifier(-free) strategies were required.

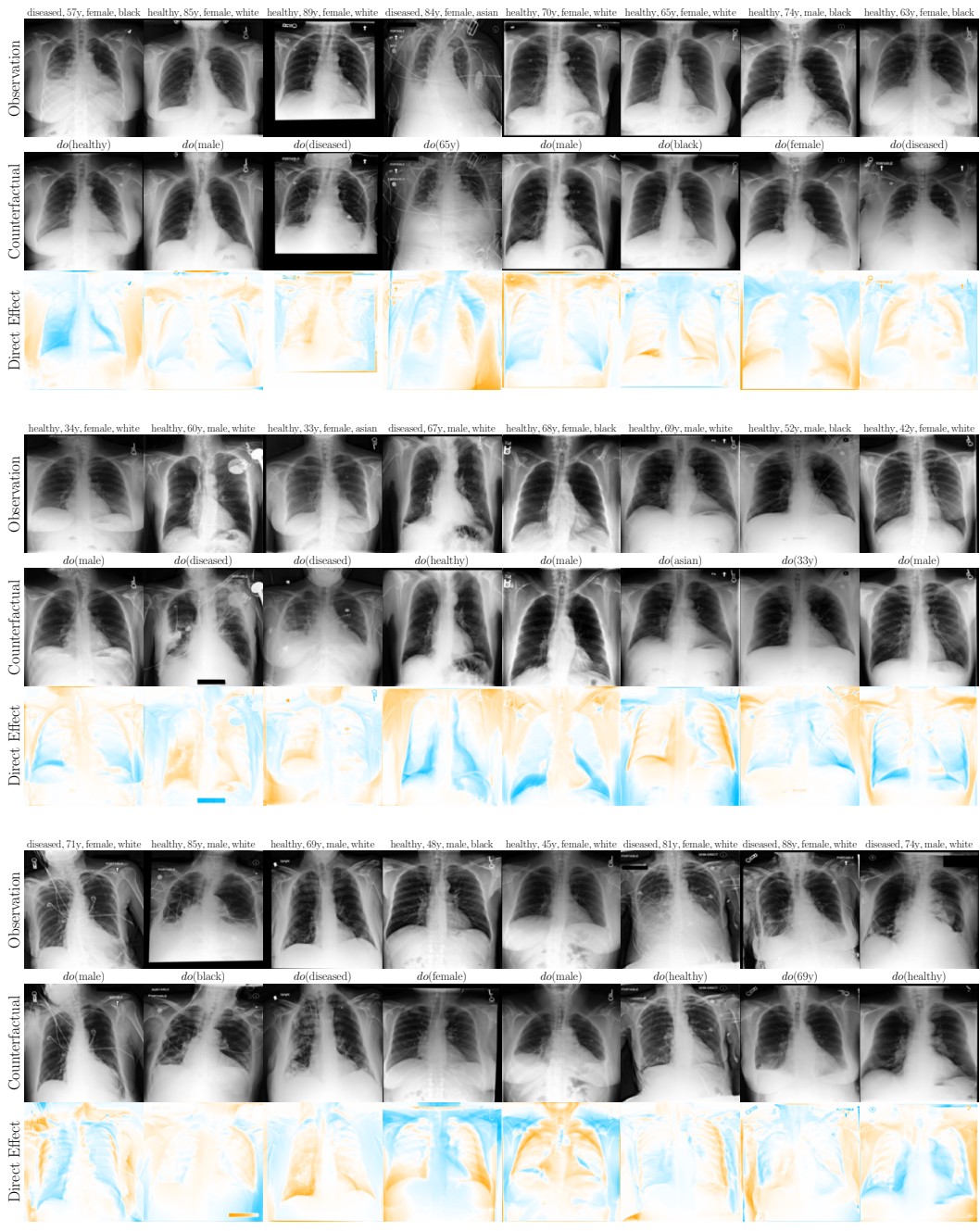

Figure 14: Extra qualitative counterfactual inference results with direct effect maps (500 ODE Euler solver steps). We observe fairly good identity-preservation overall, but counterfactual *effectiveness* appears to be somewhat prioritised by the model. We anticipate that identity preservation can be further improved using inference time guidance strategies without having to retrain the model.

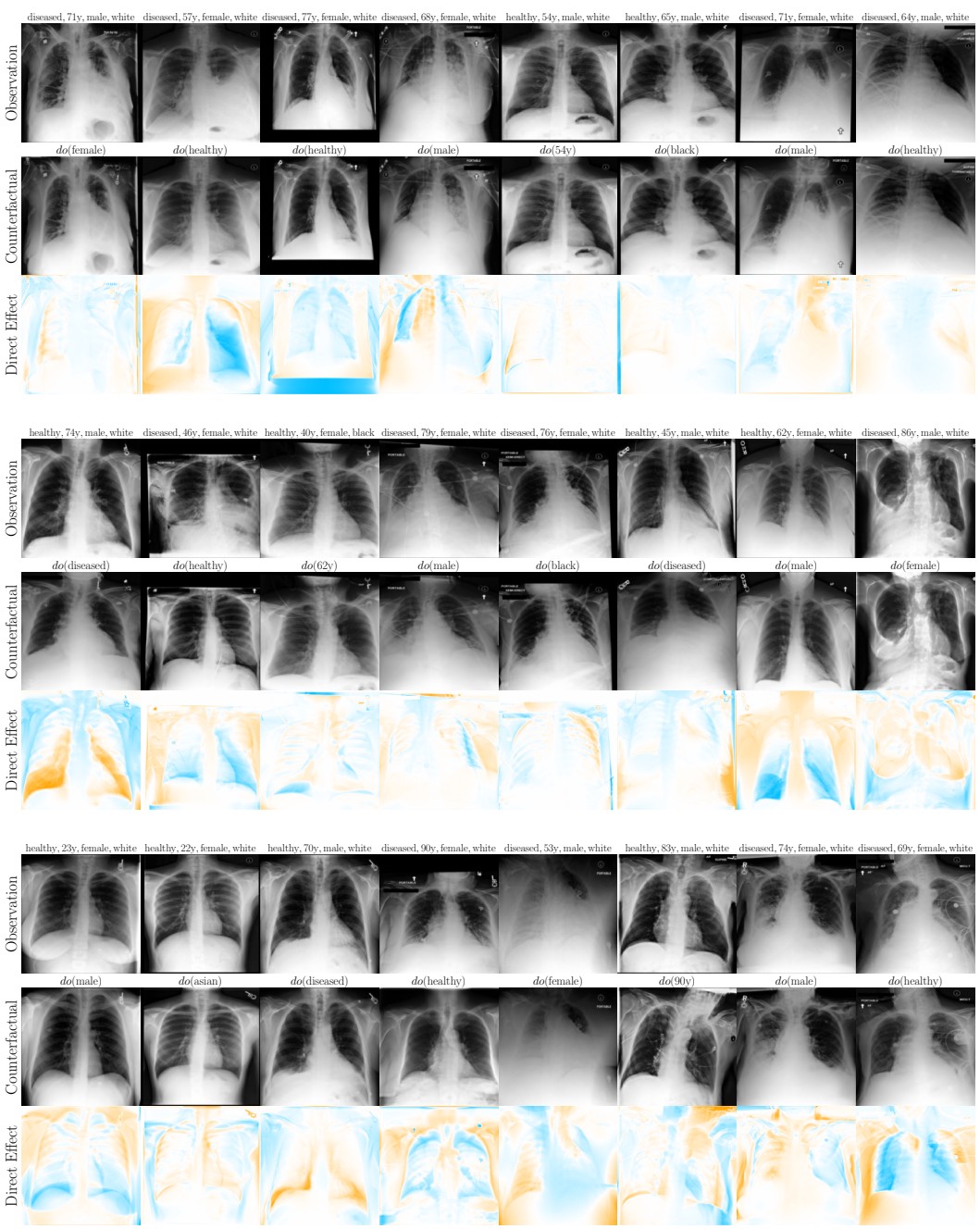

Figure 15: Qualitative counterfactual inference results with only 50 ODE Euler solver steps. We observe decent results considering the small number of ODE solving steps used (classical diffusion models would use, e.g. 1000). However, we find that identity preservation is more challenging in this regime, and spurious associations (e.g. background, artefacts (Pérez-García et al., 2025)) tend to be more prevalent. We expect inference-time classifier(-free) guidance to further improve the results.

