# OpenReview forum: "Counterfactual Identifiability via Dynamic Optimal Transport"
_NeurIPS.cc/2025/Conference — NeurIPS 2025 poster_

### Official Review · Reviewer_ugrj · 2025-06-28

**Clarity:** 3
**Significance:** 4
**Originality:** 4
**Rating:** 5
**Confidence:** 3

**Summary:**

The authors consider counterfactuals of an outcome with respect to its parent set, assumed causally sufficient. Under strict monotonicity and regularity conditions referenced in the manuscript, the authors prove nonparametric identifiability from observational data using dynamic optimal transport. They construct an estimator and show that it provides consistent results in experiments (simulated and real world) where nonparametric baselines without rigorous identifiability proofs appear to be inconsistent.

**Questions:**

The authors' results apply to counterfactuals regarding atomic interventions and atomic conditions on evidences. Do the authors expect similar results to apply under soft interventions or distributional constraints on evidences?

**Ethical Concerns:**

["NO or VERY MINOR ethics concerns only"]

**Final Justification:**

Reviews have been largely positive with concerns raised by Reviewer LANz. Primarily, the reviewer is concerned as to the significance of an extension to multivariate outcomes. In my opinion this is very much a significant extension.

Even forgoing counterfactual analysis and focusing on population-level treatment effects, I understand multidimensional outcomes in the instrument variables setup to significantly affect identifiability---in linear models we lose identifiability of the average treatment affect even with two outcomes. To my mind, the jump from single variable to multivariable outcomes is as significant as the jump from discrete/bounded outcomes to continuous unbounded ones.

For these reasons, I will be retaining my score.

**Limitations:**

Yes.

**Paper Formatting Concerns:**

None.

**Quality:**

4

**Strengths And Weaknesses:**

To the best of my knowledge, the authors are correct in saying that nonparametric counterfactual identification under monotonicity with multidimensional treatments has not been proven elsewhere. This result is therefore original and significant. The authors communicate their approach very well.

There are a few points which were not entirely clear to me:
- The authors should expound upon the baselines and other estimators without consistency proofs: are these clearly incompatible the approach proposed by the authors?
- While it is clear strict monotonically makes their approach easier to follow, where is it shown that bijectivity is insufficient alone?

I disagree with the authors in the second paragraph of their introduction that point-identifiability is necessary for valid causal inference. Bounding, hypothesis testing and structure discovery are all valuable approaches in causal inference, even when a causal model is only partially identified. The authors make clear, however, that consistent counterfactual estimation requires point identification.

---

> ### Author Rebuttal · Authors · 2025-07-30
>
> We appreciate the reviewer's positive assessment and for recognising that our main theoretical result is "**original and significant**" and "**has not been proven elsewhere**". We devoted substantial effort to presenting all our results clearly, and we are pleased that this was recognised. We address all the reviewer's comments below.
>
> > ### **Q1: The authors should expound upon the baselines and other estimators without consistency proofs: are these clearly incompatible [with] the approach proposed by the authors?**
>
> We appreciate the opportunity to elaborate on this. Baselines and other estimators without consistency proofs cannot safely be relied upon to support causal claims. In other words, a lack of identifiability, full or partial, implies the impossibility of making a precise causal claim. This means that the answer can be completely unrelated to the ground truth, and providing such guarantees is what causal methods are designed to do.
>
> For example, the $\mathrm{Naive}$ OT coupling flow baseline can model the observational distribution well, but the inferred counterfactuals are inconsistent, as the model violates Markovianity (c.f. **Figure 1**). Nasr-Esfahany et al. (2023) reported that their spline flow-based baseline also **failed** in the Markovian case, only succeeding when the backdoor criterion was applicable. In contrast, we demonstrate that our method produces consistent counterfactual inferences in both the Markovian and Frontdoor criterion settings (c.f. **Table 1**). We’ll clarify this further in the final version.
>
> > ### **Q2: While it is clear strict monotonically makes their approach easier to follow, where is it shown that bijectivity is insufficient alone?**
>
> Thank you for pointing this out. We will make this clearer in the final version.
>
> In short, a vector-valued function can be bijective but not monotone. As a simple example, consider the negative identity function:
>
> $T: \mathbb{R}^d \times \mathbb{R}^k \to \mathbb{R}^d, \quad T(u;\mathbf{pa}) = -u$,
>
> which is bijective in $u$, but not vector-monotone:
>
> $ \langle T(u_1;\mathbf{pa}) - T(u_2;\mathbf{pa}), u_1 - u_2 \rangle = (-u_1 + u_2)^\top(u_1 - u_2) = -\lVert u_1 - u_2 \rVert ^2_2 < 0$,
>
> as it violates the condition
>
> $ \langle T(u_1;\mathbf{pa}) - T(u_2;\mathbf{pa}), u_1 - u_2 \rangle \geq 0$.
>
> Another example is the $T^{(3)}$ mechanism of **Eq. (29)** in **Appendix A.1**, which is bijective but not monotone; thus, its counterfactual estimates are shown not to be consistent.  In **Appendix A.1**, we provide a gentle example showing that monotonicity is necessary for consistent counterfactual inferences in the Markovian setting.
>
> > ### **Q3: Bounding, hypothesis testing, and structure discovery are all valuable approaches in causal inference, even when a causal model is only partially identified. The authors make clear, however, that consistent counterfactual estimation requires point identification.**
>
>
> We certainly agree that partial identification methods, such as Manski-style bounds (Manski 2003), constitute valuable causal inference approaches. Our intention was not to discount such efforts, but rather to stress that identification, partial or otherwise, is essential for valid counterfactual inference and is a prerequisite for making precise causal claims from observational data. Unfortunately, this fact remains underappreciated in ongoing empirical work on high-dimensional counterfactual inference, posing an operational risk which we highlight and seek to address.
>
> To avoid misunderstanding, we will add the following to the text:
>
> *“In settings where full identification is impossible, researchers can still draw valuable causal conclusions through bounding, hypothesis testing, or structure discovery. In this work, however, we concentrate on scenarios that require point-identification to obtain consistent counterfactual estimates from observational data alone.”*
>
> > ### **Q4: The authors' results apply to counterfactuals regarding atomic interventions and atomic conditions on evidences. Do the authors expect similar results to apply under soft interventions or distributional constraints on evidences?**
>
> Thank you for the insightful question. While further theoretical work is needed to accommodate *arbitrary* soft interventions, as they can alter the underlying causal mechanism and potentially violate (vector) monotonicity, the current method supports a suitably well-defined subset of soft interventions so long as they preserve those properties.
>
> Similarly, distributional constraints on evidence (e.g. $ X  \sim Q(X) $) are more general than atomic observations $X = x$, and may not yield point-identifiable counterfactuals without additional assumptions. We see this as fertile ground for future research and expect optimal transport to play a valuable role in the analysis. We will highlight this further in our Discussion section.
>
> **References**
>
> Manski, C.F., 2003. Partial identification of probability distributions. New York, NY: Springer New York.

---

> > ### Author Response · Authors · 2025-08-07
> >
> > Dear Reviewer,
> >
> > The discussion period ends in less than 2 days (8th August AoE).
> > If you have any remaining questions or concerns, could you please let us know? This would be much appreciated.
> >
> > Thank you for your time,
> > The authors

---

> > > ### Comment · Reviewer_ugrj · 2025-08-08
> > >
> > > Reviews have been largely positive with concerns raised by `Reviewer LANz`. Primarily, the reviewer is concerned as to the significance of an extension to multivariate outcomes $V$. In my opinion this is very much a significant extension.
> > >
> > > Even forgoing counterfactual analysis and focusing on population-level treatment effects, I understand multidimensional outcomes in the instrument variables setup to significantly affect identifiability---in linear models we lose identifiability of the average treatment affect even with two outcomes. To my mind, the jump from single variable to multivariable outcomes is as significant as the jump from discrete/bounded outcomes to continuous unbounded ones - this is no trivial extension.
> > >
> > > For these reasons, I will be retaining my score.

---

### Official Review · Reviewer_LANz · 2025-06-28

**Clarity:** 4
**Significance:** 3
**Originality:** 2
**Rating:** 5
**Confidence:** 3

**Summary:**

The work studies counterfactual identifiability from observational data in models with no unbserved confounders and high-dimensional variables. The proposed approach is is a dynamic optimal transport flow mechanism strictly monotone in the target variable. On the practical side, the authors describe a specialized Batch OT coupling approach suitable for Markovian model, i.e., exogeneity is satisfied. Extensive experiments are performed on real data to validate the the paper's claims.

**Questions:**

Please see weaknesses from above.

Additionally,

(Q1) Can the results from this work be easily extended to more complex counterfactuals, e.g., nested counterfactuals?
(Q2) Can you comment on the potential connection/overlap of the result from Proposition 4.3 and counterfactual identifiability results under *noise monotonicity* (Lu et.al. Triantafyllou et.al.).

Lu, Chaochao, et al. "Sample-efficient reinforcement learning via counterfactual-based data augmentation." arXiv preprint arXiv:2012.09092 (2020).
Triantafyllou, Stelios, et al. "Agent-specific effects: A causal effect propagation analysis in multi-agent MDPs." ICML, 2024

**Ethical Concerns:**

["NO or VERY MINOR ethics concerns only"]

**Final Justification:**

Concerns about novelty were relieved.

**Limitations:**

Mentioning more clearly assumptions in conclusion and how they limit the pracictability of the work would be beneficial for readers.

**Paper Formatting Concerns:**

All good

**Quality:**

4

**Strengths And Weaknesses:**

Strengths

- The paper combines ideas from several fields and provides a rigorous theoretical analysis. I did not manage to follow all ideas and results presented, but regardless I recognize that this paper is well-written and nicely structured. Necessary background is well-covered given the space constraints. The authors have also made a very meticulous job in covering related work.
- Experimental results are the main strength of this paper. Very cleverly executed and convincing in validating theoretical claims.

Weaknesses

- (Contributions) It feels that most results are derived or similar to prior work. I am familiar with the work from Nasr-Esfahany et al. (2023) which I understand does similar work for models with one-dimensional variables. What are the challenges going from the results from this work to the multi-dimensional problem? Can the authors highlight which contributions are novel and to what extent?
- (Assumptions) To my understanding, the proposed approach does the following assumptions: exogeneity, dim(endogenous) = dim(exogenous), and exogenous distribution is uniform. It seems that this setting is too restrictive. Could the authors comment on this?

---

> ### Author Rebuttal · Authors · 2025-07-30
>
> We thank the reviewer for their positive assessment, noting our integration of ideas across fields, rigorous theoretical claims, clear structure, and thorough coverage of related work. We also appreciate their view that the experiments are a core strength: "**very cleverly executed and convincing in validating theoretical claims**". Please find our responses to all the reviewer's comments below.
>
> > ### **Q1: I am familiar with the work from Nasr-Esfahany et al. (2023) which I understand does similar work for models with one-dimensional variables. What are the challenges going from the results from this work to the multi-dimensional problem? Can the authors highlight which contributions are novel and to what extent?**
>
> We thank the reviewer for their feedback and for the opportunity to clarify our contributions.
>
> As pointed out by Nasr-Esfahany et al. (2023) in their **Section 5.1**:
>
> *“It is not clear how to generalize the monotonicity condition to BGMs with multidimensional $V$, which is a known issue in Markovian causal structures (Nasr-Esfahany & Kiciman, 2023).”*
>
> This generalisation to $d > 1$ in Markovian causal structures was indeed an open problem, with close connections to both the (non-linear)-ICA (Hyvärinen and Pajunen 1999) and disentanglement (Locatello et al. 2019) literature. In multi-dimensional settings, one’s model is subject to indeterminacies, partly caused by symmetries of the prior $P_U$, which are unresolvable without further constraints.
>
> To the best of our knowledge, we are the first to identify the constraints needed to provably solve this problem. We develop a new theoretical foundation for counterfactual identification of multi-dimensional variables by connecting dynamic Optimal Transport (OT) and causality. Building on this foundation, we introduced a  Markovian OT coupling flow which enables the estimation of high-dimensional, unique and monotone causal mechanisms. We demonstrate that our method enables consistent counterfactual inferences of high-dimensional variables ($d > 1$), given only observational data.
>
> Establishing counterfactual identification of high-dimensional Markovian SCMs is both timely and important for the responsible use of causal AI, as many recent works rely on such assumptions to support causal claims (Pawlowski et al. 2020; Sanchez and Tsaftaris 2021; Sánchez-Martin et al. 2022; Monteiro et al. 2023; Ribeiro et al. 2023; Y. Chen, L. Tian, et al. 2024).
>
> Although not the main focus of our analysis, in **Appendix A.4** we also extend counterfactual identifiability results from observational data to non-Markovian multi-dimensional ($d > 1$) settings when the following common criteria apply: **(i)** Instrumental Variable (IV), **(ii)** Backdoor Criterion (BC), and **(iii)** Frontdoor Criterion (FC). We prove counterfactual identifiability when the FC applies under similar conditions to the BC setup in Nasr-Esfahany et al. (2023), and expand the instrumental variable results to $d > 1$. These results show our method is widely applicable and useful for all these common causal structures.
>
> To conclude, in the context of the most closely related work by Nasr-Esfahany et al. (2023), we can argue that our contribution represents a breakthrough in causal modelling, with significant implications for practical applications involving counterfactual image generation.
>
> > ### **Q2: To my understanding, the proposed approach does the following assumptions: exogeneity, dim(endogenous) = dim(exogenous), and exogenous distribution is uniform. It seems that this setting is too restrictive. Could the authors comment on this?**
>
> We thank the reviewer for raising this point, and invite them to refer to the dedicated section on *Assumptions & Plausibility* in **Appendix A.5**. In addition, we provide a more succinct response below.
>
> We would like to point out that, aside from the uniform prior, these assumptions are commonplace in the literature (Pawlowski et al. 2020; Monteiro et al. 2023; Ribeiro et al. 2023; Nasr-Esfahany et al. 2023). What sets our work apart is that we provide counterfactual identifiability guarantees for the multi-dimensional case, a set of results that was previously missing for the Markovian, Instrumental Variable and Frontdoor criterion cases. These results are of vital importance to ensure that any causal claims derived from our models are properly supported.
>
> We only require that the prior $P_U$ be a uniform measure for our most strict result. Under different priors, a counterfactual equivalence relation is obtained, in the spirit of many existing identifiability results (Khemakhem et al. 2021; Nasr-Esfahany et al. 2023; Hyvärinen et al. 2024). Thus, relaxing the uniformity constraint on the prior is permissible in exchange for less strict guarantees.
>
> We close with Pearl and Bareinboim (2022):
>
> *"Assumptions are self‑destructive in their honesty. The more explicit the assumption, the more
> criticism it invites. [...] Researchers therefore prefer to declare 'threats' in public and make assumptions in private."*
>
> Our work counters this practice: we intend to make our assumptions and constraints explicit so they can be better understood, challenged, and ultimately relaxed in light of new evidence or insight.
>
> > ### **Q3: Can the results from this work be easily extended to more complex counterfactuals, e.g., nested counterfactuals?**
>
> In **Appendix A.4.3**, we prove counterfactual identifiability for ($d>1$)-dimensional variables when the Fontdoor Criterion (FC) applies. In the FC setting, we already compute nested or cross-world counterfactuals of the sort found in mediation analysis, as described below.
>
> For example, consider the graph: $X \to M \to Y$, with unobserved confounder $Z$, such that:  $X \leftarrow Z \to Y$.
> To compute a cross-world counterfactual, we nest two operations.
>
> Given observed evidence $\\{M = m, X = x\\}$, we compute the counterfactual mediator $M_{x'}$ had $X$ been $x'$:
>
> $M_{x'} \mid \\{M = m, X = x \\}$.
>
> In structural equation form, using abduction-action-prediction, we compute:
>
> $m' = f_M(x', f_M^{-1}(x, m))$.
>
> We then compute the counterfactual outcome (nested) given the counterfactual mediator:
>
> $Y_{x',M_{x'}} \mid \\{ Y = y, M = m, X = x \\} $,
>
> which in functional form is given by the nested operation:
>
> $y' = f_Y(m', f_Y^{-1}(m, y)) = f_Y(f_M(x', f_M^{-1}(x, m)), f_Y^{-1}(m, y))$
>
> So, yes, our approach already works for complex counterfactuals.
>
> > ### **Q4: Can you comment on the potential connection/overlap of the result from Proposition 4.3 and counterfactual identifiability results under noise monotonicity (Lu et.al. Triantafyllou et.al.).**
>
> We thank the reviewer for the pointers. We will add a reference to Triantafyllou et al. (2024).
>
> Definition 4.2 for noise monotonicity in Triantafyllou et al. (2024), as first specified by Pearl (2009) for binary SCMs, specifies a total order $\leq$ on the domain of $V_i$, which naturally applies to scalars ($d=1$), but not vectors ($d>1$). In the absence of a meaningful natural order, readers have thus far only considered $V_i$ as a scalar quantity (Nasr-Esfahany et al. 2023), to the best of our knowledge.
>
> For the $d=1$ case in **Proposition 4.3** (c.f. **Appendix A.2**), our result is practically equivalent to noise monotonicity, in line with Theorem 5.1 in Nasr-Esfahany et al. (2023).
>
> For the $d>1$ case in **Proposition 4.3** (c.f. **Appendix A.2**), we provide a new result by identifying a generalised notion of monotonicity for vectors and its relation to Optimal Transport (OT). We find that since the OT map is the gradient of a convex function, it is a monotone operator, and thus the counterfactual transport map must be ‘vector-monotone’ in $x$; a desired property of counterfactual functions. This is not our main result, but it represents an important motivating step for our study of OT maps as causal mechanisms.
>
> **References**
>
> Triantafyllou, Stelios, et al. "Agent-specific effects: A causal effect propagation analysis in multi-agent MDPs." ICML, 2024
>
> Lu, Chaochao, et al. "Sample-efficient reinforcement learning via counterfactual-based data augmentation." arXiv preprint arXiv:2012.09092 (2020).
>
> Pearl, J. and Bareinboim, E., 2022. External validity: From do-calculus to transportability across populations. In Probabilistic and causal inference: The works of Judea Pearl (pp. 451-482).

---

> > ### Author Response · Authors · 2025-08-05
> >
> > Dear Reviewer LANz,
> >
> > Thank you again for your positive and insightful feedback.
> >
> > In our responses, we clarified our contributions and the broad applicability of our method relative to prior work. We hope this addresses your concerns.
> >
> > If anything remains unclear, a brief comment before August 8 (AoE) would be much appreciated, and we are happy to discuss further.
> >
> > Kind regards,
> > The Authors

---

> > > ### Comment · Reviewer_LANz · 2025-08-07
> > >
> > > Thank you a lot for your nice and detailed answer. After carefully reading your answer to Q1 and going through the paper again, my concern about the novelty of this work is relieved. I would suggest the authors to add a similar discussion to the paper, explicitly highlighting the challenges they faced compared to prior work. I raise my score.

---

> > > > ### Author Response · Authors · 2025-08-07
> > > >
> > > > Dear Reviewer,
> > > >
> > > > Thank you for taking the time to reread the paper and for raising your score.
> > > > We’re glad our rebuttal resolved your concerns and appreciate the encouraging feedback.
> > > > As suggested, we’ll add a brief discussion like the one in **Q1** to better frame our contributions relative to prior work.
> > > >
> > > > Thanks again,
> > > > The authors

---

### Official Review · Reviewer_5cpq · 2025-07-01

**Clarity:** 2
**Significance:** 2
**Originality:** 3
**Rating:** 3
**Confidence:** 2

**Summary:**

This work provides theoretical guarantees for counterfactual identifiability in structural causal models (SCMs). The main result of the paper is an extension of Theorem 5.1 of Nasr-Esfahany et al. (2023) to the multivariate setting. The authors also provide empirical evaluations of their methodology.

Main assumptions:
* A SCM with independent noise variables, that is, for all $i \in \{1, \dots, n\}$
	$$X_i = f_i(X_{pa(i)}, U_i) \qquad U_i \perp X_{pa(i)}$$
where $U_i$ and $X_i$ are assumed to be $d$-dimensional random vectors.
* For all $i \in \{1, \dots, n\}$ the functions $f_i$ are bijective with respect to $U_i$.
* Knowledge of the observational distribution of the SCM and its induced graph is assumed.

Goal:
The goal is counterfactual identifiability, that is, the authors want to provide conditions under which a counterfactual query $Q$ is identifiable.

Under the aforementioned assumptions and in the univariate case, that is $d=1$, and assuming strict monotonicity of $f_i$ Nasr-Esfahany et al. (2023) establish counterfactual identifiability. The goal of the present paper is to extend this result to the multivariate case, that is, $d>1$.

**Questions:**

* Do you require strong monotonicity of the $f_i$'s? If not, how are your assumptions distinguished from strong monotonicity?

* The proof of Proposition A.8 is not quite clear to me. In the proposition, it says that there is a function $g$ for every combination of $u$ and $pa$. In general these functions can be different. It is therefore unclear to me how the authors arrive at equation (60) as $g_{u_2}$ and $g^{-1}_{u_1}$ seem to not necessarily be equal.

* If I understand Theorem 5.1 of Nasr-Esfahany et al. (2023) correctly, their results extends to the multivariate case when there is one noise variable for each $X_i$. In this context, it would be interesting to know what the significance is of modelling multiple noise variables for a single observed variable? Are there specific applications you have in mind? For example, intuitively, why would multiple noise variables be advantageous for the X-Ray experiment?

**Ethical Concerns:**

["NO or VERY MINOR ethics concerns only"]

**Final Justification:**

I would like to thank the authors for the productive discussion.

I acknowledge their commitment to revising the manuscript. However, the scope of the necessary changes seem significant, and I therefore believe the paper is not yet ready for publication without addressing two key areas.

$\textbf{1. Framing of the Contribution: }$ In my opinion the paper's central claim needs to be framed more accurately. My understanding is that Theorem 5.1 of Nasr-Esfahany et al. (2023) already handles the multidimensional case. The key contribution of this work, therefore, appears to be the generalization from using a single noise variable for each $X_i$ to using a multi-dimensional noise vector. This is an important distinction, and the justification for this specific extension is not yet sufficiently rigorous, as discussed in my previous point (Q7). The authors should clarify this point and ensure their claims precisely reflect their contribution.

$\textbf{2. Mathematical Clarity: }$ Regarding mathematical clarity, I appreciate the authors' commitment to addressing the points I raised. However, I believe the required changes are substantial and go beyond minor edits. The presentation currently lacks precision in several key places, which hinders a reader's ability to verify the technical contributions. Because significant parts of the argument may need to be rewritten for clarity, I think a thorough revision cycle would be more appropriate to ensure the paper is clear and verifiable.

Given the authors' constructive engagement, I will raise my score to 3.

**Limitations:**

It would be nice to have a discussion about what it means that $X_i$ and $U_i$ are assumed to have the same dimension $d$, and about the implications of the bounded support assumption for the noise $U_i$.

**Quality:**

2

**Strengths And Weaknesses:**

Quality:
I am not familiar enough with the optimal transport literature to verify that the proofs are correct. However, it is sometimes not clear to me what assumptions are made to get the stated conclusion. Specifically, for Proposition 4.5 and 4.11. For example, it is not clear to me what “satisfying a proposition” (line 181, 228) means. I am also unsure about the correctness of Proposition A.8. I go into more detail about the proof of Proposition A.8 in the section "Questions".
I like the empirical evaluations; they are clear and seem practically relevant.

Clarity:
In my opinion clarity is a significant issue with this paper. Especially in the theoretical part, the paper is missing mathematical clarity.
It is often unclear what assumptions are made. For example, it is unclear to me what precise assumptions are made in Proposition 4.5 and Proposition 4.11.  Do we assume monotonicity of $f_i$? It would also be good to see a discussion about the assumptions and what they mean intuitively. Below is a list of sections where I believe there are issues with clarity:
* The term "counterfactual query" is used in Definition 4.2 but never defined mathematically.
* Please only number the equations that you are actually referencing.
* It seems like the counterfactual dynamic OT map $T^* $ is used in the main text but not defined. Maybe Equation (77) in the Appendix defines $T^*$? If so, please state this explicitly in the main text.
* Sometimes "$\coloneqq$" is used for definitions and sometimes not. Please be consistent in your use of notation.
* Definition 4.2 is a definition and a statement in one. Please use the definition environment for definitions only.
* $T$ seems to be a function $T: \mathbb{R}^{d} \times \mathbb{R}^k \to \mathbb{R}^d$ (line 187) and $T: \mathbb{R}^{d} \times \mathbb{R}^{d} \times \mathbb{R}^k \to \mathbb{R}^d$ (for example line 181). If you abuse notation, please indicate this clearly.
* It also seems like $T$ is defined outside an environment and inside. Specifically, in line 187, 193 and 227. Are these all the same function?
* There is no such thing as “satisfying a proposition” (line 181, 228 for example). Please state clearly what assumption you make.
* Sometimes you use $U$ and $X$ and sometimes $\boldsymbol{U}$ and $\boldsymbol{X}$. It seems like $U$ and $X$ were never defined. It is unclear to me whether both notations mean the same thing.
* There is a “motivating example” in the Appendix under “Proofs: Counterfactual Identifiability”. This seems misplaced.
* In Equation (9) on the left hand side $x$ seems to be fixed while on the right hand side this is not the case.
* “… absolutely continuous with densities strictly positive on a bounded, open, convex domain w.r.t. the Lebesgue measure”. It probably should be “… absolutely continuous w.r.t. the Lebesgue measure and with densities strictly positive on a bounded, open, convex domain.”
* Propositions in the Appendix do not have the same numbers as the corresponding Propositions in the main text.
* In line 612 the authors require “smooth respective densities” which is an assumption that does not appear in the propositions of the main text.
* Line 726: Does the fact that the “velocity field is Lipschitz continuous” follow from a previously made assumption? Because it sounds like it is something that is assumed on the spot. If so please include all such assumptions in the proposition.

Significance:
I would have liked to see more discussion about the relevance of multivariate noise vectors for counterfactual identifiability. It seems to me that Theorem 5.1 of Nasr-Esfahany et al. (2023) trivially extends to the multivariate case if there is one noise variable for each $X_i$, that is, $X_i = f_i(X_{pa(i)}, U_i)$. For which problems is it important to model multiple noise variables per observed variable?

Originality:
This is difficult for me to answer as I do not fully understand the assumption made in this paper. However, it seems that the proof of Theorem 5.1 of Nasr-Esfahany et al. (2023) can be extended to the multivariate setting by assuming strong monotonicity of the $f_i$ (i.e. the generalisation of monotonicity to the multivariate setting) and then use Brenier’s Theorem to get the unique, monotone transport map that pushes $U$ to $X$. $T$ is then the replacement of the inverse of the CDF in the proof of Theorem 5.1 of Nasr-Esfahany et al. (2023). The theoretical results seem to mirror this idea. However, I cannot find the strong monotonicity assumption for the $f_i$. Could the authors clarify this point?

---

> ### Author Rebuttal · Authors · 2025-07-30
>
> We sincerely appreciate the reviewer’s time in preparing their feedback. We are pleased that the empirical evaluations were found to be **"clear and practically relevant**". Although we respectfully disagree that clarity is a significant issue, as corroborated by the other reviewers, we commit to incorporating all the suggested changes to further improve clarity. We consider these to be minor and straightforward to address. Please find below our detailed responses to each concern.
>
> > ### **Q1: It is unclear to me what precise assumptions are made in Proposition 4.5 and Proposition 4.11. Do we assume monotonicity of f_i? It would also be good to see a discussion about the assumptions and what they mean intuitively.**
>
> For an intuitive discussion about the assumptions, we invite the reviewer to refer to the dedicated *Assumptions & Plausibility* section in **Appendix A.5**, which appears to have been missed.
>
> To clarify, we treat 'vector-monotonicity' (c.f. **Eq. (36)**) as a maintained hypothesis about the true data-generating function when applying the model to data. However, it was not obvious that the counterfactual transport map $T^*$ would be strictly vector-monotone in $x$, and we do not assume it a priori in the theory; we prove it is true given only the standard assumptions outlined below.
>
> The assumptions made to prove **Proposition 4.5** are stated at the beginning of the proposition, precisely:
> 1. Equal dimension $\operatorname{dim}(X) = \operatorname{dim}(U)$;
> 2. Markovian setting $U \perp\\!\\!\\!\perp PA$;
> 3. $P_U$ and $P_{X|\mathbf{PA}}$ are absolutely continuous w.r.t. the Lebesgue measure, with strictly positive and bounded densities on a bounded, open, convex domain.
>
> Item 3 above corresponds to the assumptions needed for Caffarelli (1992)’s regularity theorem, so the result applies directly, ensuring that the OT map is bijective. For a helpful summary of Caffarelli (1992)’s results, the reviewer may refer to Theorem 12.50 in Villani (2008). In combination, our assumptions ensure that $P_U$ and $P_{X|\mathbf{PA}}$ admit well-behaved densities and guarantee the existence and uniqueness of a dynamic OT map (Brenier 1991; Caffarelli 1992).
>
> The assumptions made to prove **Proposition 4.11** are stated at the beginning of the proposition, namely:
>
> 1. Assumptions 1, 2 and 3 as listed above for Proposition 4.5;
> 2. $P_U$ is the continuous uniform measure on $[0,1]^d$;
> 3. $T$ is the time-1 dynamic OT map that pushes $P_U$ forward to $P_{X|\mathbf{PA}}$ described in Proposition 4.5.
>
> We then prove that the counterfactual transport map $T^*$ must be strictly vector-monotone in $x$. To the best of our knowledge, this is a new result, and it extends the classical monotone-quantile notion to multivariate settings ($d > 1$), thereby guaranteeing counterfactual identifiability from observational data.
>
> > ### **Q2: $T$ seems to be a function $T : \mathbb{R}^d \times \mathbb{R}^k \to \mathbb{R}^d$ (line 187) and $T : \mathbb{R}^d \times \mathbb{R}^d \times \mathbb{R}^k \to \mathbb{R}^d$ (for example line 181). If you abuse notation, please indicate this clearly.**
>
> Thank you for raising this. Firstly, we presume the reviewer meant to state that the map on **line 181** is $T : \mathbb{R}^k \times \mathbb{R}^k \times \mathbb{R}^d \to \mathbb{R}^d$  rather than $T : \mathbb{R}^d \times \mathbb{R}^d \times \mathbb{R}^k \to \mathbb{R}^d$. To clarify, we intended to redefine $T$ on **line 187** but recognise that this can be made clearer by avoiding an overload in notation with the map on **line 181**, for example.
>
> To resolve this ambiguity, we will define the counterfactual transport map using $T^* : \mathbb{R}^k \times \mathbb{R}^k \times \mathbb{R}^d \to \mathbb{R}^d$ before its first use in **Eq. (8)**, also ensuring consistency with its use in **Proposition 4.11**.
>
> > ### **Q3: Sometimes you use $X$ and $U$ and sometimes $\boldsymbol{X}$ and $\boldsymbol{U}$. It seems like $X$ and $U$ were never defined. It is unclear to me whether both notations mean the same thing.**
>
> As per **Definition 3.1**, we use $\mathbf{X}$ and $\mathbf{U}$ to denote the set of all the endogenous and exogenous variables in the SCM, respectively. Without loss of generality, we then define $X$ and $U$ on line 142, as relating only to the $i^{\text{th}}$ causal mechanism in the SCM.
>
> > ### **Q4: In line 612 the authors require “smooth respective densities” which is an assumption that does not appear in the propositions of the main text.**
>
> Thank you for spotting this. We inadvertently stated this in the proof, but it is not necessary to obtain this result, so we will omit it in the final version. Please refer to our response to query **Q1** above for an intuitive discussion of the assumptions required to obtain the result.
>
>
> > ### **Q5: Does the fact that the “velocity field is Lipschitz continuous” follow from a previously made assumption? Because it sounds like it is something that is assumed on the spot. If so please include all such assumptions in the proposition.**
>
> Thank you for the suggestion. We had treated Lipschitz continuity as a standard standing assumption in the Neural ODE literature. As in R. T. Chen et al. (2018) and Kidger et al. (2020), this condition holds when the velocity field is parameterised by a neural network with finite weights and Lipschitz activations (e.g. ReLU, Leaky-ReLU, tanh, softplus etc). Other smooth activations with bounded derivatives, such as GELU and SiLU, have finite global Lipschitz constants and can be scaled to 1-Lipschitz without changing qualitative behaviour.
>
> To avoid ambiguity, we will state the above explicitly and explain that it follows immediately from our network parameterisation.
>
> > ### **Q6: The proof of Proposition A.8 is not quite clear to me. In the proposition, it says that there is a function g for every combination of u and pa. In general these functions can be different. It is therefore unclear to me how the authors arrive at equation (60) as $g_{u_2}$ and $g_{u_1}^{-1}$ seem to not necessarily be equal.**
>
> Thank you, we are happy to clarify. Our **Proposition A.8** result mirrors Nasr-Esfahany et al. (2023)’s Proposition 6.2, but here it is extended to an optimal transport context.
>
> First, we prove that two equivalent transport maps $T^{(1)} \sim_{\mathcal{L}_3} T^{(2)}$, in the sense of **Definition 4.7**, produce the same counterfactuals. Transport equivalence implies that there exists an invertible transition map $g$ between the exogenous variables of the two transport maps $T^{(1)}, T^{(2)}$. That is, we have that:
>
> $u^{(1)} = g(u^{(2)}), \quad \text{and} \quad u^{(2)} = g^{-1}(u^{(1)})$.
>
> Then, by the equivalence condition in **Definition 4.7**, the equation in question (i.e. **Eq. (60)**) follows directly by substitution:
>
> $T^{(1)}(g(u^{(2)}); \mathbf{pa}^\*) = T^{(2)}(g^{-1}(u^{(1)}); \mathbf{pa}^\*)$.
>
>
> > ### **Q7: If I understand Theorem 5.1 of Nasr-Esfahany et al. (2023) correctly, their results extends to the multivariate case when there is one noise variable for each x_i. In this context, it would be interesting to know what the significance is of modelling multiple noise variables for a single observed variable?**
>
> As pointed out by Nasr-Esfahany et al. (2023) in their Section 5.1:
>
> *“It is not clear how to generalize the monotonicity condition to BGMs with multidimensional $V$, which is a known issue in Markovain causal structures (Nasr-Esfahany & Kiciman, 2023).”*
>
> Our contributions include precisely this missing generalisation to $d > 1$ for the Markovian, Instrumental Variable and Frontdoor Criterion settings. This unlocks counterfactual identifiability for multi-dimensional variables, such as images.
>
> The Markovian setting in particular was an important open problem with close connections to both the (non-linear)-ICA (Hyvärinen and Pajunen 1999) and disentanglement (Locatello et al. 2019) literature, rendering our theoretical results broadly relevant and widely applicable. In the context of the most closely related work by Nasr-Esfahany et al. (2023), we can argue that our contribution represents a breakthrough in causal modelling, with significant implications for practical applications involving counterfactual image generation.
>
> There seems to be a misunderstanding, as we never use multiple noise variables per observed variable. In all cases, we make the equal dimension assumption that $\operatorname{dim}(X) = \operatorname{dim}(U)$, for any multi-dimensional variable $X$ within an SCM $\mathfrak{C}$ (see line 142). For an intuitive discussion about this assumption, please refer to the *Assumptions & Plausibility* section in **Appendix A.5**, and the response below.
>
> > ### **Q8: It would be nice to have a discussion about what it means that $X_i$ and $U_i$ are assumed to have the same dimension $d$, and about the implications of the bounded support assumption for the noise $U_i$.**
>
> The equal-dimension assumption $\operatorname{dim}(X) = \operatorname{dim}(U)$ is commonplace in flow-based generative mode, as it ensures that a bijective transport map is mathematically possible. When this assumption does not hold for the true data-generating process, a bijective model can still be used by, for instance, augmenting the lower-dimensional variable with dummy coordinates.
>
> Lastly, $P_U$ must be absolutely continuous w.r.t. the Lebesgue measure, with a strictly positive and bounded density on a bounded, open, convex domain for Caffarelli (1992)’s regularity theorem to apply. The implication of this is that the OT map is bijective provided that the target measure $P_{X|\mathbf{PA}}$ meets the same criteria. We will make this clearer in the final version.
>
> **References**
>
> Villani, C., 2008. Optimal transport: old and new (Vol. 338, pp. 129-131). Berlin: springer.
>
> Kidger, P., Morrill, J., Foster, J. and Lyons, T., 2020. Neural controlled differential equations for irregular time series. Advances in neural information processing systems, 33, pp.6696-6707.

---

> > ### Author Response · Authors · 2025-08-05
> >
> > Dear Reviewer 5cpq,
> >
> > Thank you again for taking the time to review our paper.
> >
> > We hope our detailed responses have clarified our contributions and the assumptions underlying the new theoretical results.
> >
> > If anything remains unclear, a brief comment before August 8 (AoE) would be appreciated, and we will respond quickly.
> >
> > Kind regards,
> > The Authors

---

> > ### Comment · Reviewer_5cpq · 2025-08-05
> >
> > I thank the authors for their response. While I appreciate the clarifications, several key points remain unclear, and I note that some of my original remarks were not addressed. I will now elaborate on my outstanding concerns regarding the points that were discussed.
> >
> > $\textbf{Q1:}$
> > The concept of 'vector-monotonicity' (c.f. Equation (36)) appears to be a central assumption. For the paper to be self-contained and for the results to be verifiable, this concept must be formally defined in the main text and its implications clearly discussed. I was unable to find the mentioned discussion of this assumption in Appendix A.5.
> >
> > $\textbf{Q2:}$
> > The proposed redefinition of $T$ resolves my concern on this point. I thank the authors for this helpful clarification.
> >
> > $\textbf{Q3:}$
> > I remain confused regarding the dimensionality of variables within the SCM framework. By convention, and as reflected in Definition 3.1, a structural equation defines a scalar random variable (e.g., $U_i$, $X_i$). The authors' formulation (line 142), however, seems to imply that the structural equation for a single node $X_i$ outputs a random vector of dimension $d>1$.
> >
> > $\textbf{Q4:}$
> > The authors answer resolves my concern.
> >
> > $\textbf{Q5:}$
> > The authors answer resolves my concern.
> >
> > $\textbf{Q6:}$
> > I follow the authors' reasoning but remain unconvinced by the counterargument. The existence quantifier $\forall u, \text{pa} \, \exists g$ implies that the function $g$ may depend on the specific values of $\text{pa}$ and $u$. To make this dependence explicit, let us denote the function as $g_{\text{pa}, u}$. From Equation (58) and Definition A.7, it follows that $u^{(1)} = g_{\text{pa}, u^{(1)}}(u^{(2)})$. Consequently, we can write:
> > $T^{(1)}(u^{(1)}; \text{pa}^\star) = T^{(1)}(g_{\text{pa}, u^{(1)}}(u^{(2)}); \text{pa}^\star) = T^{(2)}((g_{\text{pa}^\star, u^{(1)}})^{-1}(g_{\text{pa}, u^{(1)}}(u^{(2)})); \text{pa}^\star)$.
> > My core question remains: why does the composition $(g_{pa^\star, u^{(1)}})^{-1} \circ g_{pa, u^{(1)}} $ simplify to the identity? Since the function $g$ is parameterized by the parent configuration, and we are considering two different configurations ($\text{pa}$ and $\text{pa}^\star$), this step seems to not be self-evident.
> >
> > $\textbf{Q7:}$
> > My understanding of the proposed framework is that $X = f(\text{PA}(X), U)$, where $X$ and $U$ are random vectors of dimension $d>1$ (which is one structural assignment in the SCM). Could the authors clarify the advantage of this formulation---where each component $X_i$ depends on the entire exogenous vector $U$ (i.e., $X_i = f_i(\text{PA}(X), U)$)? What is gained compared to a model with component-wise exogenous variables, $X_i = f_i(\text{PA}(X), U_i)$, for which Theorem 5.1 of Nasr-Esfahany et al. (2023) seems to apply (as we can identify each $f_i$ separately)?
> >
> > $\textbf{Q8:}$
> > The authors state that assuming $X$ and $U$ are random vectors of the same dimension is a commonplace assumption in flow-based generative models. While this may be true, this is a non-standard assumption in the SCM literature. The same concern applies to the assumption of a bounded support for $U$. It is essential to justify these assumptions within the theoretical framework of SCMs, rather than simply importing conventions from a different field. What is the causal interpretation or theoretical motivation for these constraints within the context of Structural Causal Models?

---

> ### Author Response · Authors · 2025-08-06
>
> We appreciate the reviewer's time and feedback. We've done our best to address each point in detail below, given the limited available space.
>
> **Q1:** We thank the reviewer for the helpful suggestion. We have added the following definition of $d>1$ monotonicity in the main text (based on **Eq. (36)**), *before* **Proposition 4.3** where it is first invoked.
>
> **Definition** (Monotone Operator in $u$)**.** Fix $\mathbf{pa} \in \mathbb{R}^k$. A mapping $f : \mathbb{R}^k \times \mathbb{R}^d \to \mathbb{R}^d$ is monotone in $u$ if
>
> $\langle f(\mathbf{pa},u_1)-f(\mathbf{pa},u_2), u_1-u_2\rangle \geq 0 \quad \text{for all} \\, u_1, u_2 \in \mathbb{R}^d$.
>
> We also now include the following **implications** paragraph *after* **Proposition 4.3** based on the material previously in **Remarks A.2** & **A.4**.
>
> "Monotonicity of the counterfactual transport map $T^*$ w.r.t. $x$ ensures rank preservation: individuals with higher factual outcomes retain higher counterfactual outcomes (no rank inversions), which supports *fairness* and consistent counterfactual inferences (c.f. **App A.1**). As we will show, the Optimal Transport (OT) specialisation of this map generalises quantile and rank functions for $d>1$, ensuring multivariate rank preservation. Intuitively, mass is pushed along gradients of a convex potential; the gradient structure prevents "folding'', and its monotonicity enforces directional consistency so that points that start close remain close under the map, thereby preserving order and preventing overlaps."
>
> **Q3:** In **Definition 3.1**, we deliberately do not specify the variables as scalars explicitly, to allow for multivariate ($d>1$) generalisations of $X_{i}, U_{i}$. We then clarify our intentions on **line 107**, stating that we focus on counterfactual identification for multi-dimensional variables $X_i \in \mathbf{X}$. Without loss of generality, on **line 142** we reiterate our intent and drop the $i$ subscript on $X, U$ to avoid notational clutter. We hope this clarifies our intention, and we are happy to consider suggestions for improvements.
>
> **Q6:** We appreciate the reviewer's observation and identify that the placement of the existence quantifier in **Eq. (57)** for the function $g$ as the cause for the confusion. To address this, we have amended it to correctly state that the existence quantifier precedes the universal ones, i.e. it now reads: $\exists g, \forall \\, u, \mathbf{pa}$, etc.
>
> This aligns with **Definition 6.1** in Nasr-Esfahany et al. (2023), where a single, invertible function $g$ is required to ensure equivalence across all values of $u$ and $\mathbf{pa}$. There is no dependence of $g$ on $\mathbf{pa}$, nor is there a need for a different $g$ for each value of $u$.
>
> Importantly, this amendment is purely notational and does not affect any of the theoretical results. We have revised the text accordingly to clarify this point and avoid further ambiguity.
>
> **Q7:** Thank you for the question; we are happy to clarify. The reviewer is correct, and our Markovian SCM may have $n$ such structural assignments. In our analysis, we focus on the $i^{\text{th}}$ mechanism in the Markovian SCM without loss of generality.
>
> *"Could the authors clarify the advantage of this [our] formulation?"*
>
> We appreciate the reviewer's subtle point. Beyond being the natural parameterisation for multi-dimensional variables, our formulation lets global factors (e.g. image brightness/contrast) act coherently across pixels. From an Independence of Cause and Mechanism (ICM) perspective, the mechanisms stay fixed while global shifts are absorbed by $P_U$. In contrast, the component-wise model $X_i = f_i(\operatorname{PA}_{X}, U_i)$ can only transmit dependence through parent links, so mimicking global shifts requires intricate, coordinated retuning of many $f_i$, undermining modularity.
>
> As pointed out by Nasr-Esfahany et al. (2023) in their Section 5.1:
>
> *"It is not clear how to generalize the monotonicity condition to BGMs with multidimensional $V$, which is a known issue in Markovian causal structures (Nasr-Esfahany & Kiciman, 2023)."*
>
> To the best of our knowledge, we are the first to identify the constraints needed to provably solve this problem.
>
> **Q8:** Thank you for your comment. We point out that assuming $X$ and $U$ are random vectors of equal dimension is commonplace in prior work using SCMs, see e.g. Pawlowski et al. (2020)'s invertible/explicit mechanism, Nasr-Esfahany et al. (2023)'s BC result for $d>1$, and (Sanchez and Tsaftaris 2021, Rasal et al. 2025) using diffusion. Assuming $P_U$ has bounded support is also not uncommon in the literature, see e.g. **Definition 3** in Xia et al. (2021), and their subsequent line of work. We hope this addresses the reviewer's concern, as such assumptions are not uniquely ours.
>
> Rasal et al. Diffusion Counterfactual Generation with Semantic Abduction. ICML 2025.
> Xia et al. The causal-neural connection: Expressiveness, learnability, and inference. NeurIPS 2021.

---

> ### Author Response · Authors · 2025-08-06
>
> Please find below our responses to the remaining (minor) remarks in the initial review; we were not able to address these directly in our initial rebuttal due to space constraints. If there are any other points we have missed, please do let us know, and we will be more than happy to clarify.
>
> **Q9-10:** The term "counterfactual query" is used in Definition 4.2 but never defined mathematically. Definition 4.2 is a definition and a statement in one. Please use the definition environment for definitions only.
>
> Thank you for your feedback. I appreciate your point about separating the definition and the statement. Our definition is designed to be consistent with Pearl (2009)'s Definition 3.2.3, with theirs applying to any query $Q$. Given this established treatment, we chose to structure ours similarly, as counterfactual queries are subsumed therein. I hope this clarifies the reasoning behind our approach, but we are open to further adjustments if needed. For example, we can first define $Q$ to be any computable quantity of a model as in Pearl (2009)'s Definition 3.2.3, then proceed with our Definition 4.2.
>
> **Q11:** $T$ is defined outside an environment and inside. Specifically, in line 187, 193 and 227. Are these all the same function?
>
> Yes, these are all the same function. In line with our proposal to **Q2**, we will omit the definition on **line 187** to avoid redundancy. If agreed by the reviewer, we can also avoid redefining $T$ on **line 227** by writing: *"Using the time-1 dynamic OT map $T$ defined in Proposition 4.5, which pushes $P_U$ forward to $P_{X|\mathbf{PA}}$ ..."*. We would be happy to consider alternative phrasings and suggestions.
>
> **Q12:** There is no such thing as “satisfying a proposition” (line 181, 228 for example). Please state clearly what assumption you make.
>
> Thank you for the feedback. We will rephrase **line 181** to read: *"Monotonicity of the counterfactual transport map $T^\*$ w.r.t. $x$ (c.f. Proposition 4.3) guarantees that..."*.
>
> As per **Q11**, we will also adjust **line 228** to read: *"Using the time-1 dynamic OT map $T$ defined in Proposition 4.5, which pushes $P_U$ forward to $P_{X|\mathbf{PA}}$ ..."*. We trust these changes address the reviewer's concern. If a restatement of the assumptions made in **Proposition 4.5** would be more appropriate, please let us know and we will append them to the sentence above.
>
> **Q13:** There is a “motivating example” in the Appendix under “Proofs: Counterfactual Identifiability”. This seems misplaced.
>
> Thank you for pointing this out. We agree and will move the motivating example to a new, different section directly above the “Proofs: Counterfactual Identifiability” section in the Appendix.
>
> **Q14:** In Equation (9) $x$ on the left hand side seems to be fixed while on the right hand side this is not the case.
>
> Thank you for spotting this. We can fix $x$ on the right-hand side, which results in a Dirac point mass $\delta_{T^{\*}(\mathbf{pa}^*, \mathbf{pa}, x)}$, for a deterministic counterfactual map $T^\*$. We will amend the paper to reflect this change.
>
> **Q15:** Propositions in the Appendix do not have the same numbers as the corresponding Propositions in the main text.
>
> Thank you for raising this. We will revise the Appendix so that any Proposition restated there carries the same number as in the main text (e.g., “Proposition 4.3 (restated)”), and we’ll update all cross-references.

---

> > ### Comment · Reviewer_5cpq · 2025-08-07
> >
> > I would like to thank the authors for the productive discussion.
> >
> > I acknowledge their commitment to revising the manuscript. However, the scope of the necessary changes seem significant, and I therefore believe the paper is not yet ready for publication without addressing two key areas.
> >
> > $\textbf{1. Framing of the Contribution: }$
> > In my opinion the paper's central claim needs to be framed more accurately. My understanding is that Theorem 5.1 of Nasr-Esfahany et al. (2023) already handles the multidimensional case. The key contribution of this work, therefore, appears to be the generalization from using a single noise variable for each $X_i$ to using a multi-dimensional noise vector.
> > This is an important distinction, and the justification for this specific extension is not yet sufficiently rigorous, as discussed in my previous point (Q7). The authors should clarify this point and ensure their claims precisely reflect their contribution.
> >
> > $\textbf{2. Mathematical Clarity: }$
> > Regarding mathematical clarity, I appreciate the authors' commitment to addressing the points I raised. However, I believe the required changes are substantial and go beyond minor edits. The presentation currently lacks precision in several key places, which hinders a reader's ability to verify the technical contributions. Because significant parts of the argument may need to be rewritten for clarity, I think a thorough revision cycle would be more appropriate to ensure the paper is clear and verifiable.
> >
> > Given the authors' constructive engagement, I will raise my score to 3.

---

> ### Author Response · Authors · 2025-08-07
>
> Dear Reviewer,
>
>
> Thank you again for the time and effort you have dedicated to our manuscript.
>
> We respectfully disagree with your two central criticisms. Below are our final remarks for your and others' consideration.
>
> **1. Scope relative to Nasr-Esfahany et al. (2023)**
>
> Nasr-Esfahany et al. themselves note in Section 5.1:
>
> *“It is not clear how to generalize the monotonicity condition to BGMs with multidimensional
> $V$, which is a known issue in Markovian causal structures (Nasr-Esfahany & Kiciman, 2023).”*
>
> Applying their $d=1$ result component-wise introduces three key limitations:
>
> 1. **Impose an arbitrary coordinate order.** Scalars have a natural order; high-dimensional variables such as images do not;
> 2. **Force coordinate-wise strong monotonicity**, restricting models to pixel-wise autoregressive functions;
> 3. **Global effects** (e.g., image brightness/contrast) must be expressed through an intricate coordination of pixel-wise mechanisms, undermining mechanism modularity.
>
> Our new approach represents the natural extension to $(d>1)$-dimensional variables, and avoids all three issues.
>
> **2. Mathematical Clarity**
>
> We consider the original exposition clear, and other reviewers echoed that judgment:
>
> - "clearly written, with minimal typographical errors"
> - "well-organized"
> - "Section 4 presents a series of theoretical results in a clear and structured manner"
> - "well-written and nicely structured"
> - "Necessary background is well-covered given the space constraints"
> - "authors have also made a very meticulous job in covering related work"
> - "authors communicate their approach very well"
>
> In our view, the suggested edits are clarifications rather than substantive changes, as they leave every theoretical result and conclusion intact. All of your suggestions are nonetheless appreciated and have already been incorporated.
>
> Thank you again for your time and for helping strengthen our paper.
>
>
> Best regards,
> The authors

---

### Official Review · Reviewer_hTdz · 2025-07-04

**Clarity:** 4
**Significance:** 3
**Originality:** 3
**Rating:** 5
**Confidence:** 3

**Summary:**

Counterfactual identifiability ($\mathcal{L}_3$) from observational data is both important and challenging. According to Pearl's Causal Ladder, it is generally impossible without (often strong) assumptions. Previous methods for counterfactual identifiability often require model assumptions such as linearity, bijectivity, Lipschitz continuity, or the Markovian property. They also rely on data assumptions such as faithfulness, atomic interventions, interventional data, or paired data (e.g., from randomized controlled trials or counterfactual datasets) obtained under different interventional distributions.

This paper, primarily extending the work of Nasr-Esfahany et al. (2023), builds theoretical guarantees based on bijectivity and monotonicity assumptions to extend identifiability from one-dimensional endogenous variables $X_i$ to $d$-dimensional endogenous variables $X_i$. The goal is to infer counterfactual queries of $X_i$ given its parents $\textbf{pa}_i$.

The theortical contributions of the paper follow this roadmap:

1. Proposition 4.3 shows that if the mechanism is monotone in $u$, then the counterfactual transport map $T$ is monotone in $x$, linking the data-generating process assumption to a learnable property of the function.

2. Proposition 4.5 shows that, under several assumptions, monotonicity leads to a unique mapping $T$ determined by $P_U$ and $P_{X|\textbf{PA}}^\mathfrak{C}$.

3. Proposition 4.8 attempts to prove that two transport maps are counterfactually equivalent if there exists an invertible map $g$.

4. Proposition 4.10 shows the existence of a monotone, bijective map $g^*$ that minimizes the quadratic transport loss.

5. Proposition 4.11, building on the previous propositions, establishes counterfactual identifiability for Markovian Structural Causal Models (SCMs).

Furthermore, the paper proposes a practical matching method using dynamic optimal transport (OT) flow and demonstrates its effectiveness on both synthetic datasets and a real-world X-ray dataset.

**Questions:**

1. In Proposition 4.8, two transport maps $T^{(1)}$ and $T^{(2)}$ are equivalent if there exists an invertible transition map $g$, and in Proposition 4.10, two prior distributions can be mapped by a unique, bijective, and monotone transport map $g^\*$ that minimizes the quadratic transport cost. But in Proposition 4.11, we need to identify/learn the $g$ between the two priors: the uniformly distributed prior and the unknown true prior. We cannot directly learn the exogenous prior from data; thus, minimizing the quadratic transport cost might lead to a $g$ that is not the true $g^\*$ we want.

2. What is the assumption on the interventions: soft intervention or hard intervention? It seems that the paper requires the intervention to preserve the monotonicity of the interventional SCM. Is that too restrictive?

3. In the synthetic experiment, what is $Z$? (line 287)

4. Intuitively, in section 6.1, given a point observation X=x, how possible could we learn the whole ellipse? Could you elaborate more on the experiment and connect it to the theoretical assumptions and guarantees?

**Ethical Concerns:**

["NO or VERY MINOR ethics concerns only"]

**Final Justification:**

Counterfactual identifiability from observational data is both important and challenging. By extending the analysis from 1-dimensional to a $d$-dimensional endogenous variable $X_i$, the authors take this challenge one step further. I recommend accept for this paper due to its theoretical contribution. Notice that I didn't check the proofs in the appendix.

**Limitations:**

this paper discusses the limitation briefly in the conclusion section.

**Paper Formatting Concerns:**

N.A.

**Quality:**

3

**Strengths And Weaknesses:**

**Strengths**

1. The extension of counterfactual identifiability in Bijective Causal Models from one-dimensional to $d$-dimensional endogenous variables is a significant and meaningful contribution.

2. The paper is clearly written, with minimal typographical errors.

3. The paper is well-organized. In particular, Section 4 presents a series of theoretical results in a clear and structured manner.

**Weaknesses**
1. The strong assumptions and setting (e.g., known DAG, bijective and monotonic SCM) might limit the practical usefulness of the method. Could the authors elaborate on potential application scenarios where these assumptions are reasonable?

2. My main concern for the synthetic dataset is about the selection of $P(U).$ If a more complicated $P(U)$ is used in the data generating process, would the generation result be affected?
3. Due to the restrictive assumptions made in this paper, additional experiments might be beneficial. For example, how does the proposed method perform when certain assumptions are violated? Specifically, without the monotonicity assumption or without bijectiveness, to what extent is performance affected?

**Minor Comments**

1. In Definition 4.1 (Markovian SCM.), the period inside the parentheses should be removed.

2. On page 3, line 106, what does $P_X^{\mathfrak{C}_{x'}|X=x}$ mean? Particular, how is an SCM conditioned on $X = x$ interpreted? Clarification would be helpful.

---

> ### Author Rebuttal · Authors · 2025-07-30
>
> We thank the reviewer for their positive feedback. We are pleased that our counterfactual identifiability result was deemed **"a significant and meaningful contribution"**. We also appreciate the recognition of our paper's clarity, minimal typographical issues, and well-organised structure, particularly the presentation of theoretical results.
>
> > ### **Q1: The strong assumptions (e.g., known DAG, bijective and monotonic SCM) might limit the practical usefulness of the method. Could the authors elaborate on potential application scenarios where these assumptions are reasonable?**
>
> In **Section 6.2**, we demonstrate the practical utility of the method on a high-dimensional medical imaging problem, outperforming existing works by a large margin.
>
> **Known DAG:** Many causal relationships are known in the medical domain, thanks to a long history of results from randomised controlled trials and meta-analyses. This positions healthcare as the ideal application area for causal modelling, as one can tap into existing medical knowledge to help build causal graphs.
>
> **Bijectivity:** Bijective causal mechanisms (Pawlowski et al. 2020, Nasr-Esfahany et al. 2023) subsume many model classes with known identifiability results, e.g., (non)-linear additive noise models (Shimizu et al. 2006; Hoyer et al. 2008; Peters et al. 2014), post-nonlinear models (K. Zhang and Hyvärinen 2009), and location-scale models (Immer et al. 2023). As such, they represent an attractive model class with practical instantiations using modern flows.
>
> **Monotone Map:** Learning high-dimensional monotone maps is challenging. However, this is now provably possible in practice thanks to recent advances (Pooladian et al. 2023; Tong et al. 2024). When the model is misspecified and/or the assumptions are violated, guarantees may not hold. This is not unique to our approach, but holds for all identification results.
>
> Many recent works already rely on such assumptions (Pawlowski et al. 2020; Sanchez and Tsaftaris 2021; Monteiro et al. 2023; Ribeiro et al. 2023), but their causal claims remain speculative without identification, representing an operational risk. Thus, our work represents an important advance for the responsible use of causal AI.
>
> We also provide theoretical results for non-Markovian settings (**Section 4.3**), when 3 common criteria apply: Backdoor Criterion, Instrument Variable and the Frontdoor Criterion, rendering our approach widely applicable.
>
> > ### **Q2: My main concern for the synthetic dataset is about the selection of P(U). If a more complicated P(U) is used in the data-generating process, would the generation result be affected?**
>
> Intuitively, a more complex data-generating process (DGP) is expected to be harder to model.
>
> To test this hypothesis, we created new datasets with multimodal $P_U$. We use an indicator $s_i \sim \text{Bern}(0.5)$ to induce a non-linear, bimodal shift $\mu_i \in \\{-2, 2\\}$ into $U$'s mechanism. We also use a categorical shift, with $\mu_i \in \\{-2, 0, 2\\}$, to generate another multimodal setting. More details on all new experiments will be added to the paper.
>
> We confirm that more complex $P_U$'s can affect generation, but importantly, the relative rank of our models remains consistent. Naturally, we expect larger models to reduce performance gaps for complicated $P_U$'s.
>
> |DGP $P_U$|EBM|Flow|OT-EBM|OT-Flow|
> |-|-|-|-|-|
> |Original|2.32±.01|2.30±.02|0.93±.02|0.76±.01|
> |Bimodal|4.95±.01|4.95±.08|1.72±.05|1.40±.05|
> |Multimodal|5.11±.03|5.01±.03|1.73±.09|1.49±.06|
>
> > ### **Q3: How does the proposed method perform when certain assumptions are violated? Specifically, without the monotonicity assumption or without bijectiveness, to what extent is performance affected?**
>
> To thoroughly address this, we ran a new set of experiments.
>
> In addition to measuring counterfactual error, we inspect the **Composition ($\downarrow$)** and **Reversibility ($\downarrow$)** counterfactual soundness axioms, to reveal performance disparities under assumption violations.
>
> We use the Mean Average Percentage Error (MAPE) metric, and 50/250 steps for ODE/SDE solving.
>
> **Experiment 1: Monotonicity violation**
>
> We use 20 cycles for composition and reversibility measurements; models which are not vector-monotone exhibit significantly inferior counterfactual soundness.
>
> |Model|Monotone|Composition|Reversibility|CF Error|
> |-|-|-|-|-|
> |Nasr-Esfahany et al. (2023)|N|-|-|607|
> |EBM|N|27.86±.369|39.15±.349|2.319±.006|
> |Flow|N|27.99±.096|39.39±.229|2.295±.020|
> |OT-EBM|Y|0.659±.020|1.056±.072 |0.925±.019|
> |OT-Flow|Y|**0.618**±.046|**1.016**±.090 |**0.763**±.015|
>
> **Experiment 2: Bijectivity violation**
>
> We use 3 strategies to stress-test bijectivity violations: **(i)** stochastic *abduction* by solving an SDE instead of an ODE; **(ii)** stochastic *prediction* by solving an SDE; **(iii)** classifier-free guidance (CFG). One cycle was used for composition and reversibility.
>
> We observe that bijectivity violations affect reversibility and composition significantly, as it becomes harder to 'undo' interventions.
>
> |Model|Bijective|Composition|Reversibility|CF Error|
> |-|-|-|-|-|
> |SDE Abduction|N|2.950±.013|4.173±.019|3.064±.011|
> |SDE Prediction|N|3.006±.015|4.276±.019|3.126±.007|
> |CFG (w=1.05)|N|0.032±.004|0.487±.010|2.174±.070|
> |CFG (w=1.15)|N|0.032±.004|1.499±.013|6.081±.069|
> |CFG (w=1.25)|N|0.032±.004|2.598±.024|9.809±.078|
> |OT-Flow|Y|0.032±.004|**0.056**±.009|**0.763**±.015|
>
> **Experiment 3: Markovianity violation**
>
> We observe that composition and reversibility are reasonable for the Naive OT-Flow version, as the map remains bijective and monotone. However, the counterfactual error is significantly higher for the Naive version, as the violation leads to inconsistent counterfactuals.
>
> |OT-Flow|Markovian|Composition|Reversibility|CF Error|
> |-|-|-|-|-|
> |Naive|N|0.063±.005|0.346±.012|47.01±.016|
> |Ours|Y|**0.032**±.004|**0.056**±.009 |**0.763**±.015|
>
> > ### **Q4: In Proposition 4.11, we need to identify/learn the g between the uniform prior and the true prior. We cannot directly learn the exogenous prior from data; thus, minimizing the transport cost might lead to a g that is not the true g we want.**
>
> Thank you for the thoughtful question. To clarify, learning the prior transition map $g$ explicitly is not strictly necessary.
>
> Let the true generator be $x = f(u;\operatorname{pa})$, $u \sim P_U$. We train a single OT-Flow $T(u;\operatorname{pa})$, $u \sim U(0,1)^d$, to match the output law.
>
> Although there exists a unique OT solution for $T$, we make no claims that it factorises exactly into $f \circ g$, for any choice of $P_U$. There could be a map with a lower transport cost that only implicitly encodes $g$.
>
> To illustrate this simply, consider a rubber-cube representing the uniform prior $U(0,1)^d$.
>
> 1. A warp $g$ stretches the cube so that the embedded points follow the true prior $P_U$;
>
> 2. A map $f$ then moulds the cube into data space.
>
> A single map $T$ could enable steps 1 & 2 in unison if this reduces transport cost. In analogous terms, a path A-to-C may be closer than a path A-to-B-to-C. We will make this point clearer in the final version.
>
> > ### **Q5: What is the assumption on the interventions: soft intervention or hard intervention? It seems that the paper requires the intervention to preserve the monotonicity of the interventional SCM. Is that too restrictive?**
>
> We focus on hard interventions as they enable many useful real-world applications. This is exemplified by our experiments in Section 6.2, and ongoing work in: (i) evaluating the impact of interventions (Kusner et al. 36 2017; Tsirtsis and Rodriguez 2023), (ii) understanding cause-effect relationships (Karimi et al. 2020; Budhathoki et al. 2022), (iii) generating targeted synthetic data (Pitis et al. 2022; Roschewitz et al. 2024), and (iv) quantifying disease severity (Mehta et al. 2025).
>
> While further theoretical work is needed to accommodate *arbitrary* soft interventions, as they can alter the underlying mechanism, the current method supports a suitably defined subset of soft interventions that preserve vector-monotonicity.
>
> > ### **Q6: In the synthetic experiment, what is Z? (line 287)**
>
> We thank the reviewer, as their question helped us find a typo. The random variable $Z$ is part of the data-generating process, a confounder satisfying the Backdoor Criterion (BC) w.r.t. $\operatorname{PA} \to X$. We include it to reproduce Nasr-Esfahany et al. (2023)’s BC result. In the text, we inadvertently state that we randomised $Z$ to induce a Markovian setting, but what we did was randomise $\operatorname{PA}$, ensuring $U  \perp\\!\\!\\!\perp \operatorname{PA}$. We will amend this in the final version.
>
> > ### **Q7: Intuitively, in section 6.1, given a point observation $X=x$, how possible could we learn the whole ellipse? Could you elaborate more on the experiment and connect it to the theoretical assumptions and guarantees?**
>
> A point observation of $X = x$ fixes the location but underdetermines the shape of the ellipse; thus, infinitely many ellipses can be drawn through that point. By further observing the angle $\operatorname{PA} = \operatorname{pa}$, it is possible to infer the semi-major and -minor parameters of the ellipse via causal *abduction*. The whole ellipse for an observed $(\operatorname{pa}, x)$ can then be estimated by answering counterfactual queries for all angles in $(0,2\pi)$.
>
> We can provably learn the correct model by using our proposed Markovian OT-Flow coupling. In contrast, the Naive coupling violates Markovianity, entangling $U$ with $\operatorname{PA}$, and yielding inconsistent counterfactuals. This constructed scenario is crucial for verifying our theoretical claims. It shows that when our assumptions and constraints hold (i.e. Markovian/FC setting, monotone OT-Flow map), counterfactual identification in $d>1$ from observational data is possible.
>
> Mehta, R. CF-Seg: Counterfactuals meet Segmentation. arXiv preprint arXiv:2506.16213.

---

> > ### Author Response · Authors · 2025-08-05
> >
> > Dear Reviewer hTdz,
> >
> >
> > Thank you again for your time and thoughtful feedback.
> >
> > We hope our detailed responses and additional experiments on **Q2: more complicated priors** and **Q3: assumption violation stress-testing** address your concerns.
> >
> > If anything remains unclear, please share a brief note before the discussion period closes on August 8 (AoE), and we will clarify promptly.
> >
> > Kind regards,
> > The Authors

---

> > ### Comment · Reviewer_hTdz · 2025-08-08
> >
> > I thank the authors for their response. I acknowledge their efforts with extra experiments, explaination regarding my concerns about the eclipse experiments. I will adjust my score accordingly.

---

> > > ### Author Response · Authors · 2025-08-08
> > >
> > > Dear Reviewer,
> > >
> > > Thank you for taking the time to re-evaluate our work and for adjusting your score.
> > > We are pleased that our rebuttal helped address your concerns.
> > > The extra experiments you suggested have certainly strengthen our work.
> > >
> > > Much appreciated,
> > > The authors

---

### Note · Authors · 2025-08-11

As stated by Reviewer ugrj, the reviews have been largely positive:

- Reviewer hTdz considers our work **"a significant and meaningful contribution"**;
- Reviewer 5cpq found our empirical evaluations **"clear and practically relevant"**;
- Reviewer LANz found our experiments to be **"very cleverly executed and convincing in validating theoretical claims"**;
- Reviewer ugrj stated that our theoretical result is **"original and significant"** and **"has not been proven elsewhere"**.

The pre-rebuttal concerns raised can be categorised as:

**(i) Effect of assumption violations and the choice of prior** [Reviewer hTdz].
To thoroughly address this concern, we conducted new experiments.
- We generated new datasets with multimodal priors and analysed their effect on counterfactual inferences. Our findings confirm that more complicated priors are harder to model, but crucially, the relative rank of the models remains consistent; our Markovian OT-Flow comes out on top.
- We stress-tested 3 types of assumption violation: (i) monotonicity; (ii) bijectivity, and (iii) Markovianity. We find that: (i) models which are not vector-monotone exhibit significantly inferior counterfactual soundness; (ii) models which are not bijective cannot reverse interventions well; (iii) the naive OT flow, which violates Markovianity, produces inconsistent counterfactuals.

***Reviewer hTdz acknowledged the extra experiments and agreed to adjust their score.***

**(ii) Significance and implications of new theoretical results** [Reviewers LANz, hTdz, 5cpq].

In our responses, we first point to Section 5.1 in Nasr-Esfahany et al. (2023):

*“It is not clear how to generalize the monotonicity condition to BGMs with multidimensional $V$, which is a known issue in Markovain causal structures (Nasr-Esfahany & Kiciman, 2023).”*

We then clarify that our contributions include precisely this missing generalisation to $d>1$ for the Markovian, Instrumental Variable and Frontdoor Criterion settings, establishing a new theoretical foundation for multivariate counterfactual identification via OT.

***Reviewer LANz's concern was relieved, and they raised their score.***

***Reviewer ugrj emphasised that our identification results for multivariate outcomes are "very much a significant and non-trivial extension".***

**(iii) Mathematical clarity** [Reviewer 5cpq].

All the reviewer's suggestions, which we see as minor, were incorporated. Please see our final rebuttal remarks for details.

---

### Decision · Program_Chairs · 2025-09-17

**Decision:**

Accept (poster)

**Comment:**

This paper provides the counterfactual identifiability results, generalising the result of Nasr-Esfahany et al. (2023) to the multi-dimensional setting, using dynamic Optimal Transport (OT) approach. The contribution is a non-trivial advancement of the counterfactual inference for high-dimensional data such as images.

A majority of reviews have been largely positive, although there is a disagreement in terms of significance, framing of the contribution, and mathematical clarity. Reviewer `5cpq` questioned the significance of the contribution, citing that "Theorem 5.1 of Nasr-Esfahany et al. (2023) already handles the multidimensional case." and argued that the scope of the necessary changes seems too significant for the paper to be ready for publication. On the other hand, Reviewers `hTdz`, `LANz`, `ugrj` seem to agree that the contribution is significant enough for the paper to be published. Reviewer ugrj has specifically mentioned that "In [his] opinion this is very much a significant extension [of Nasr-Esfahany et al. (2023)]."

After reading the paper, I concur with the majority of the reviewers that the contribution of this paper is substantial enough to warrant its publication at NeurIPS as a poster. This decision is based on the argument that Theorem 5.1 of Nasr-Esfahany et al. can already handles the multidimensional case. Nevertheless, this doesn't significantly diminish the contribution of this paper.